

# Topological superconductors on superstring worldsheets

**Justin Kaidi[1], Julio Parra-Martinez[1,2] and Yuji Tachikawa[3],**
**with a mathematical appendix by Arun Debray [4]**

**1** Mani L. Bhaumik Institute for Theoretical Physics, Department of Physics and Astronomy,
University of California, Los Angeles, CA 90095, USA
**2** Center of Mathematical Sciences and Applications,
Harvard University, Cambridge, MA 02138, USA
**3** Kavli Institute for the Physics and Mathematics of the Universe (WPI),
University of Tokyo, Kashiwa, Chiba 277-8583, Japan
**4** Department of Mathematics, University of Texas at Austin, TX 78712, USA

## Abstract

We point out that different choices of Gliozzi-Scherk-Olive (GSO) projections in superstring theory can be conveniently understood by the inclusion of fermionic invertible phases, or equivalently topological superconductors, on the worldsheet. This allows us to find that the unoriented Type 0 string theory with $\Omega^2 = (-1)^f$ admits different GSO projections parameterized by $n \mod 8$, depending on the number of Kitaev chains on the worldsheet. The presence of $n$ boundary Majorana fermions then leads to the classification of D-branes by $KO^n(X) \oplus KO^{-n}(X)$ in these theories, which we also confirm by the study of the D-brane boundary states. Finally, we show that there is no essentially new GSO projection for the Type I worldsheet theory by studying the relevant bordism group, which classifies corresponding invertible phases. In two appendixes the relevant bordism group is computed in two ways.

This paper provides the details for the results announced in the letter [1].



# 1 Introduction and summary

## 1.1 Generalities

In perturbative formulations of superstring theories, one treats the 2d worldsheets of strings as 2d quantum field theories with fermions. The most common treatment is the one due to Neveu-Schwarz [2] and Ramond [3], often called the NSR formalism. There, one starts with ten bosonic fields $X^{\mu=0,\dots,9}$ and ten left- and right-moving fermionic fields $\psi^{\mu=0,\dots,9}$ and $\tilde{\psi}^{\mu=0,\dots,9}$. To remove the closed-string tachyon and at the same time obtain spacetime spinors, one must perform a crucial step called the Gliozzi-Scherk-Olive (GSO) projection [4, 5]. The two Type II superstring theories, Type IIA and Type IIB, arise due to a difference in the specifics

of this projection.

It was pointed out in [6, 7] that the GSO projection can be interpreted as a sum over the possible spin structures on the worldsheet, with different consistent GSO projections corresponding to different ways of assigning complex phases to spin structures, in a manner consistent with cutting and gluing of the worldsheet. This point of view makes manifest the all-genus consistency of known GSO projections, which is not evident in the one-loop analysis often presented to beginners of string theory e.g. in [8]. It does not, however, tell us whether we have found all possible GSO projections. Indeed, possible consistent GSO projections for unoriented superstring theories have not been studied systematically in the past.

One approach to the enumeration of all consistent GSO projections comes from a rather unexpected place, namely from the study of symmetry-protected topological (SPT) and invertible phases of matter in condensed matter physics.[1] For the purposes of this paper, an invertible phase for a symmetry $G$ can be defined as a system which has a one-dimensional gapped vacuum on any closed spatial manifold with a background field for $G$. By taking the infrared limit, one can then isolate a quantum system whose entire Hilbert space on any closed spatial manifold with any background field is one-dimensional and contains the vacuum only, with its partition function simply a complex phase. We note that for an internal symmetry, the background field is simply a non-dynamical gauge field for the symmetry. On the other hand, for fermion number symmetry the background field is the spin structure, while for time-reversal symmetry the background field is the "un-oriented-ness" of the spacetime.

An invertible phase provides a method for assigning complex phases to spin and other structures on a manifold, in a manner consistent with cutting and gluing. Conversely, any such assignment corresponds to an invertible phase. Therefore, if one can classify invertible phases, one can classify all consistent GSO projections. Invertible phases with fermion number symmetry and some additional discrete symmetries are usually called topological superconductors in the condensed matter literature. Therefore, the classification of GSO projections is equivalent to the classification of topological superconductors in $(1 + 1)$ dimensions.

The classification of invertible phases has been an important topic of recent research in theoretical condensed matter physics. As will be reviewed below, a general answer in terms of bordism groups has been obtained, see e.g. [9–11]. The upshot is that with this result, we can now carry out the classification of possible GSO projections on a given worldsheet, once the structure on said worldsheet is specified.

One important feature of a nontrivial invertible phase in $d$ dimensions for a symmetry $G$ is that, if it is put on a spacetime with boundary, the $(d-1)$-dimensional boundary theory necessarily hosts nontrivial degrees of freedom. In particular, if $G$ is unbroken on the boundary, the $G$-symmetric boundary theory carries a corresponding $G$-anomaly. One familiar case is that of the chiral anomaly of a fermion in spacetime dimension $2n$, which is captured by a Chern-Simons term in $2n + 1$ dimensions. This is known as anomaly inflow [12, 13]. The current understanding is that all anomalies[2] in $(d-1)$ dimensions, both local and global, can

---

[1]In the literature of high-energy physics, the terms SPT and invertible phase are often used interchangeably. In condensed matter physics they have subtly different connotations. In this paper we stick to the terminology of invertible phases, which are more directly relevant for our purposes.

[2]Some qualifications need to be added to this blanket statement. First, this framework is mostly about the anomalies of partition functions, and therefore does not immediately describe the conformal anomaly. Second, anomalies of supersymmetry are less well understood, and it is not clear whether they can be described by bulk invertible phases. That said, neither is conclusively outside of this framework. As for the first, the anomaly described by the Kitaev chain is about the impossibility of quantizing a single Majorana fermion, which is also not directly about the phase of the partition function. As for the second, the supersymmetry anomaly recently found in [14, 15], which is a superpartner of the anomaly in R-symmetry, was first found in the context of AdS/CFT [16]. We also note that the shortening anomaly of [17] is related to the fact that the scalar target space of the holographic supergravity dual is often not Kähler. All this suggests that these anomalies might also be described in a suitable generalization of the current framework. It would be interesting to work this out.

be characterized in terms of invertible phases in $d$ dimensions.

Applying this observation to the worldsheets of superstrings, we conclude that different invertible phases, i.e. different GSO projections, will require different boundary conditions on the edges of worldsheets. Since the boundaries of worldsheets describe the D-branes to which strings attach, this means that the properties of D-branes reflect the choice of GSO projection. The discussions up to this point can be summarized schematically as follows:

$$
\begin{array}{cccc}
 & \text{boundary anomaly} & : & \text{bulk invertible phase} \\
\sim & \text{properties of D-branes} & : & \text{choice of GSO projection.}
\end{array}
\tag{1.1}
$$

## 1.2 GSO projections and K-theory classification of D-branes

Let us now be more concrete. First, we recall the classification of invertible phases in terms of bordism groups.[3] Let $X$ denote collectively the structure on the spacetime, appropriate for the systems we would like to classify. For example, $X$ consists of a spin structure and a $G$ gauge field for systems with fermion number symmetry and an internal symmetry $G$. We define the $X$-bordism group $\Omega_d^X$ in dimension $d$ to be

$$
\Omega_d^X := \{d\text{-dimensional manifolds with } X \text{ structure}\}/\sim,
\tag{1.2}
$$

where the equivalence relation is introduced so that $M_d \sim M_d'$ if and only if there exists $N_{d+1}$ with the structure $X$ such that $\partial N_{d+1}$ has $M_d$ as the incoming boundary and $M_d'$ as the outgoing boundary. The group structure is given by the disjoint union. Then, the topological invertible phases in spacetime dimension $d$ are classified by [9–11]

$$
\mho_X^d := \mathrm{Hom}(\Omega_d^X, U(1)).
\tag{1.3}
$$

This simply means that the topological invertible phase for an element $\alpha \in \mho_X^d$ assigns the partition function $\alpha(M_d) \in U(1)$ in such a way that it only depends on the bordism class $[M_d] \in \Omega_d^X$.[4]

In this paper we will encounter the following structures on the worldsheet: spin structure for oriented Type 0 strings, $\mathrm{Spin} \times \mathbb{Z}_2$ structure for oriented Type II strings, $\mathrm{Pin}^\pm$ structure for two types of unoriented Type 0 strings, and "double pin" or DPin structure for Type I strings. We remark here that originally the II, I, 0 in Type II, I, 0 strings referred to the number of supersymmetries in ten dimensions. Contrary to this usage, in this paper we refer to any NSR strings with independent GSO projections on left- and right-moving spin structures as Type II, and to any NSR strings with diagonal GSO projections as Type 0. Type I strings will then be defined as Type II NSR strings on unoriented worldsheets.

The relevant dual bordism groups are listed in Table 1. There we have used a slightly more general notation, with $\Omega_d^X(Y)$ representing the bordism group of $d$-dimensional manifolds with $X$ structure equipped with a map to $Y$. Then for example a structure $X'$ consisting of spin structure and a $\mathbb{Z}_2$ gauge field can equivalently be thought of as having $(X, Y) = (\mathrm{spin\ structure}, B\mathbb{Z}_2)$,

---

[3]For other recent applications of bordism groups to high-energy theory, see [18–26].

[4]More precisely, $\mho_X^d$ as defined here classifies invertible phases whose partition functions do not depend continuously on the background fields. Therefore it includes e.g. the 2d theta term $\int \theta F/(2\pi)$, which only depends on topological data, but it does not include e.g. the 3d gravitational Chern-Simons term, which does depend continuously on the metric. The latter is accounted for by considering $(D\Omega^X)^{d+1}$ instead, where $D$ denotes the Anderson dual. This group classifies the deformation classes of invertible phases which can depend continuously on the background fields. Since we take the deformation classes, $(D\Omega^X)^{d+1}$ does not include the theta term, which can be continuously varied. The torsion parts of both groups coincide: $\mathrm{Tors}\,\mathrm{Hom}(\Omega_d^X, U(1)) = \mathrm{Tors}(D\Omega^X)^{d+1}$, since torsion invertible phases are discrete and cannot depend continuously on the background data. Both $\mathrm{Hom}(\Omega_d^X, U(1))$ and $(D\Omega^X)^{d+1}$ are generalized cohomology theories, but their common torsion part is not. This unfortunately makes the torsion part less mathematically natural. For all the cases of interest to us in this paper, all bordism groups are finite, so these subtle differences can and will be ignored.

Table 1: Dual bordism groups relevant to our analysis. The first four columns are classic [9, 27]. The last column is new.

| $d$ | $\mho^d_{\mathrm{Spin}}(pt)$ | $\mho^d_{\mathrm{Spin}}(B\mathbb{Z}_2)$ | $\mho^d_{\mathrm{Pin}^-}(pt)$ | $\mho^d_{\mathrm{Pin}^+}(pt)$ | $\mho^d_{\mathrm{DPin}}(pt)$ |
|---|---|---|---|---|---|
| 2 | $\mathbb{Z}_2$ | $\mathbb{Z}_2^2$ | $\mathbb{Z}_8$ | $\mathbb{Z}_2$ | $\mathbb{Z}_2^2$ |
| 3 | 0 | $\mathbb{Z}_8$ | 0 | $\mathbb{Z}_2$ | $\mathbb{Z}_8$ |

where $B\mathbb{Z}_2$ is the classifying space of $\mathbb{Z}_2$ gauge fields, and therefore $\Omega_d^{X'} = \Omega_d^{\mathrm{Spin}}(B\mathbb{Z}_2)$. Similarly, letting $pt$ stands for a point, we have $\Omega_d^X = \Omega_d^X(pt)$.

Let us begin by discussing the group $\mho^2_{\mathrm{Spin}}(pt) = \mathbb{Z}_2$. The nontrivial element is mathematically known as the Arf invariant, and assigns to a surface $\Sigma$ with a choice of spin structure $\sigma$ a sign $(-1)^{\mathrm{Arf}(\Sigma,\sigma)}$. A spin structure is called even or odd depending on whether this sign is $+1$ or $-1$. There are many mathematical and physical definitions of the Arf invariant, one of which is as the number modulo two of zero modes of the Dirac operator on the surface $\Sigma$ with spin structure $\sigma$ [28]. From this definition, we see easily that $(-1)^{\mathrm{Arf}(T^2,\sigma)}$ is $-1$ if and only if the spin structure $\sigma$ is periodic along both cycles of the torus $T^2$. There is also a combinatorial definition [29, 30], which we will recall below.

In the continuum quantum field theory language, the Arf invariant may be written in terms of the partition function of a mass $m$ Majorana fermion on $(\Sigma, \sigma)$ as $Z_{\mathrm{ferm}}(m \gg 0; \Sigma, \sigma)/Z_{\mathrm{ferm}}(m \ll 0; \Sigma, \sigma)$. In general, the infinite-mass limit of a fermion partition function is known as an $\eta$-invariant in the mathematics literature, meaning that the Arf invariant is an example of an $\eta$-invariant. There is also a discretized Hamiltonian version of this massive Majorana fermion defined on a spin chain — this is known as the Kitaev chain [31]. In both of these descriptions, it is easy to argue that one needs a single Majorana fermion on the $(0+1)$d boundary of the $(1+1)$d system hosting the Arf invariant theory.

A single Majorana fermion cannot be consistently quantized, since two Majorana fermions act on a two-dimensional Hilbert space irreducibly. This means that, assuming that this Hilbert space is the tensor product of two copies of the Hilbert space for a single fermion, the single-fermion Hilbert space would need to have dimension $\sqrt{2}$. This is one manifestation of the anomaly of the boundary theory, and will turn out to explain the difference by a factor of $\sqrt{2}$ between the tensions of D9-branes in Type IIA and Type IIB theories, originally found in [32].

Without the Arf invariant on the worldsheet, as will be the case for Type IIB strings, the endpoints of open strings will naturally couple to unitary bundles. Consideration of tachyon condensation motivates one to introduce an equivalence relation on unitary bundles, leading to the statement that stable D-branes on $X$ are classified by complex K-theory $K^0(X)$ [33]. On the other hand, in the presence of the Arf invariant the boundary of the worldsheet needs unitary bundles together with an additional Majorana fermion, or equivalently with an action of the complex Clifford algebra $\mathrm{Cl}(1, \mathbb{C})$. Unitary bundles with an action of $\mathrm{Cl}(1, \mathbb{C})$, under a suitable equivalence relation implementing tachyon condensation, are classified by $K^1(X)$, thus reproducing the known classification of the D-branes in the Type IIA theory [33, 34].

We next discuss the effects of including the topological superconductor corresponding to the group $\mho^2_{\mathrm{Pin}^-}(pt) = \mathbb{Z}_8$ on worldsheets. The worldsheets can now be nonorientable and are equipped with a Pin$^-$ structure. This structure will be seen to be compatible with the Type 0 string, but not with the Type I string. The generator of the group $\mathbb{Z}_8$ of time-reversal invariant topological superconductors is again the Kitaev chain, but now with the added assumption of time-reversal invariance. The invertible phase corresponding to $n$ modulo 8 is simply $n$ copies of the Kitaev chain, and has $n$ time-reversal symmetric Majorana fermions on the boundary. Physically, the reason that we need only consider $n$ modulo 8 is that one can introduce a four-

fermi interaction to the $n = 8$ theory which gives rise to a theory with unique ground state [35]. In continuum field theory language, the effective action describing the basic non-trivial phase is the Arf-Brown-Kervaire (ABK) invariant, to be discussed below.

Roughly speaking, with $n$ copies of the time-reversal-symmetric Kitaev chain on the worldsheet, open string endpoints can now couple to orthogonal bundles with an action of $n$ time-reversal invariant Majorana fermions, or equivalently with an action of the real Clifford algebra $\text{Cl}(n, \mathbb{R})$. Tachyon condensation then leads to the classification of D-branes by $KO^n(X)$. As we will see, a more careful analysis reveals that the classification is in fact by $KO^n(X) \oplus KO^{-n}(X)$.

In the case of Type I strings, the natural way to specify the worldsheet fermions is to consider chiral fermions on the orientation double cover of the worldsheet. This leads to a structure which we call "double pin" structure, since it will be shown to contain both $\text{Pin}^\pm$ as subgroups. We will find by a standard algebraic topology computation that any invertible phase one can add on the worldsheet is either the Arf invariant associated to the orientation double cover, or a continuum version of the Haldane chain. We will find that the Arf invariant on the double cover can be removed by performing a spacetime parity transformation along one direction, meaning that it does not give rise to physically distinct theories. On the other hand, the invertible phase corresponding to the low energy limit of the $S = 1$ Haldane chain [36, 37], whose partition function counts the number of $\mathbb{RP}^2$ modulo 2, gives rise to the difference between $O9^\pm$-orientifold planes, thus differentiating between Type I and $\tilde{\text{I}}$ worldsheet theories.

We note in passing that Ryu and Takayanagi have pointed out in [38, 39] that the periodic table [40, 41] of free topological superconductors and topological insulators, based on K-theory and KO-theory, can be naturally realized by considering D-branes in string theory. In those works the topological superconductors were realized on brane worldvolumes, whereas here we consider the topological superconductors on string worldsheets.

## Organization

The aim of this paper is to give details on the results presented thus far in the Introduction. The rest of this paper is organized as follows.

In Sec. 2, we begin by reviewing the necessary preliminary material concerning topological superconductors with several variants of spin structure. In Sec. 3, we study the effects of worldsheet invertible phases on massless closed string and D-brane spectra. In the next two sections, we give a more detailed analysis of the classification of D-branes. This is done from two different perspectives: in Sec. 4 we study D-branes via boundary fermions and tachyon condensation, while in Sec. 5 we utilize the boundary state formalism. The final section Sec. 6 is devoted to the algebraic-topological study of the possible invertible phases on the Type I worldsheet.

Additional background information and details of calculations are given in the appendices. In Appendix A, we briefly review the NSR formulation of superstring theory. In Appendix B, we provide a short review of the boundary state formalism necessary for calculations in Sec. 5. The results of this appendix are also utilized in Appendix C, in which we discuss the issue of tadpole cancellation for some of the Type 0 theories discussed in this paper. The final three appendices are more mathematical. In Appendix D we reobtain many of our results for Arf and ABK invariants by means of index theory. Much of this appendix is due to E. Witten [42]. In Appendix E we provide the technical details of the algebraic-topological computation used in Sec. 6, which uses the Atiyah-Hirzebruch spectral sequence. In Appendix F, written by Arun Debray, we explain another computation of the same bordism group via the Adams spectral sequence.

The results of this paper were already announced in a short letter [1] by the same authors.

The authors also note that when said work was nearing completion, they were informed[5] that E. Witten has an unpublished work with large overlap with theirs; two seminars he gave can be found in [43, 44], and are highly recommended for their clarity in presentation.

# 2 The (1+1)d topological superconductors

The main topic of this paper is the addition of fermionic invertible phases to oriented and unoriented string worldsheets. In this section we begin by reviewing some basic facts about fermions and (s)pin structures (see [6, 45] for more detailed reviews), as well as about the known invertible phases for Spin, Spin $\times \mathbb{Z}_2$, Pin$^-$, and Pin$^+$ structures [9]. Also of importance to us will be "double pin" or DPin structure, though we postpone a discussion of this to Section 6.

Many of the results that we obtain via combinatoric methods in this section can also be obtained via index theory, i.e. by studying the properties of free fermions. For completeness, we discuss this approach in Appendix D.

## 2.1 Oriented invertible phases

On an oriented $d$-manifold $M$ the structure group of the tangent bundle $TM$ is SO($d$). In order to consider fermions on $M$, we need to lift SO($d$) to its double cover Spin($d$) as specified by the short exact sequence

$$0 \to \mathbb{Z}_2 \to \text{Spin}(d) \to \text{SO}(d) \to 0. \tag{2.1}$$

There might be an obstruction to doing so, which is captured by the second Stiefel-Whitney class of $TM$, i.e. $w_2(TM) \in H^2(M, \mathbb{Z}_2)$. If this class is trivial, we say that the manifold admits a spin structure. Such a spin structure is generically not unique. Given a spin structure, we can obtain another one by twisting by an element of $H^1(M, \mathbb{Z}_2)$.

In this paper, we focus on two-dimensional manifolds $\Sigma$, which we take to be the worldsheet of a string. Any orientable two-manifold admits a spin structure since $w_2(T\Sigma) = w_1^2(T\Sigma)$ mod 2. For notational simplicity, from now on we write $w_i := w_i(T\Sigma)$.

### 2.1.1 Invertible phase for Spin

Our primary interest is in fermionic invertible phases, i.e. phases which depend on a choice of spin structure $\sigma$ on $\Sigma$. In the absence of any symmetry besides fermion number $(-1)^{\text{f}}$, the group capturing such phases is $\mho_{\text{Spin}}^2(pt) = \mathbb{Z}_2$. The effective action for the corresponding fermionic invertible phase can be written in terms of the Arf invariant [9],

$$e^{2i\pi S_{\text{eff}}(\Sigma, \sigma)} = (-1)^{\text{Arf}(\Sigma, \sigma)}, \tag{2.2}$$

where $\text{Arf}(\Sigma, \sigma)$ is defined modulo 2. For simplicity, we will often leave the dependence on $\sigma$ implicit.

As discussed in the Introduction, this phase is a continuum version of the Kitaev chain [31]. In the continuum field theory language, this corresponds to the definition [45]

$$(-1)^{\text{Arf}(\Sigma, \sigma)} := \frac{Z_{\text{ferm}}(m \gg 0; \Sigma, \sigma)}{Z_{\text{ferm}}(m \ll 0; \Sigma, \sigma)}, \tag{2.3}$$

where $Z_{\text{ferm}}(m; \Sigma, \sigma)$ is the partition function of a free massive Majorana fermion of mass $m$. To see that the right-hand side is $\pm 1$, we note that the non-zero eigenvalues $E$ of the Dirac

---

[5]The authors thank Kantaro Ohmori and Matthew Heydeman for this information.

operator $D$ comes in pairs $\pm E$, since $\Gamma := \gamma^1\gamma^2$ is globally well-defined on an oriented spin surface and $D\Gamma = -\Gamma D$. Therefore,

$$\frac{Z_{\text{ferm}}(+m)}{Z_{\text{ferm}}(-m)} = \prod_{E=0}\left(\frac{iE+m}{iE-m}\right)\prod_{E>0}\frac{(iE+m)(-iE+m)}{(iE-m)(-iE-m)} = (-1)^{\text{index }D}. \tag{2.4}$$

The Arf invariant can also be defined combinatorially. To do so, given a spin structure $\sigma$ on $\Sigma$, we define $\tilde{q}(a) \in \mathbb{Z}_2$ for each $\mathbb{Z}_2$-valued 1-cocycle $a$ on $\Sigma$ by taking a non-self-intersecting 1-cycle $A$ Poincaré dual to it and declaring

$$\tilde{q}(a) = \begin{cases} 0 & \text{if the spin structure around } A \text{ is NS,} \\ 1 & \text{if the spin structure around } A \text{ is R .} \end{cases} \tag{2.5}$$

This function $\tilde{q}(a)$ is known as a quadratic form and satisfies

$$\tilde{q}(a+b) - \tilde{q}(a) - \tilde{q}(b) = \int_\Sigma a \cup b. \tag{2.6}$$

There is a one-to-one correspondence between such quadratic forms and spin structures [30].

The Arf invariant can be defined in terms of this quadratic form as follows,

$$(-1)^{\text{Arf}(\Sigma,\sigma)} := \frac{1}{\sqrt{|H^1(\Sigma,\mathbb{Z}_2)|}} \sum_{a \in H^1(\Sigma,\mathbb{Z}_2)} (-1)^{\tilde{q}(a)}. \tag{2.7}$$

To see that the right-hand side is $\pm 1$, we consider its square:

$$\begin{aligned}
\text{RHS}^2 &= \frac{1}{|H^1(\Sigma,\mathbb{Z}_2)|} \sum_{a,b \in H^1(\Sigma,\mathbb{Z}_2)} (-1)^{\tilde{q}(a)+\tilde{q}(b)} \\
&= \frac{1}{|H^1(\Sigma,\mathbb{Z}_2)|} \sum_{a,b \in H^1(\Sigma,\mathbb{Z}_2)} (-1)^{\tilde{q}(a+b)+\int_\Sigma a \cup b} \\
&= \frac{1}{|H^1(\Sigma,\mathbb{Z}_2)|} \sum_{a,c \in H^1(\Sigma,\mathbb{Z}_2)} (-1)^{\tilde{q}(c)+\int_\Sigma a \cup c},
\end{aligned} \tag{2.8}$$

where we have defined $c = a + b \in H^1(\Sigma,\mathbb{Z}_2)$ and used that $\int_\Sigma a \cup a = 0$ for an orientable manifold. When $c = 0$, the summand is 1 and the sum contributes a factor of $|H^1(\Sigma,\mathbb{Z}_2)|$. When $c \neq 0$ there are equally many $\int_\Sigma a \cup c = 0,1$ contributions by assumption of a non-degenerate intersection pairing, and hence these contributions cancel out. Thus we find that $\text{RHS}^2 = 1$.

We now focus on the torus $T^2$, which admits four spin structures. We begin by listing all elements of $H^1(T^2,\mathbb{Z}_2)$, which is an order four group containing $\{0, a, b, a+b\}$. Here $a$ and $b$ are mod 2 Poincaré duals of the $A$- and $B$-cycles of the torus, respectively. Then using formula (2.7), we have

$$\begin{aligned}
(-1)^{\text{Arf}(T^2)} &= \frac{1}{\sqrt{4}}\left(1 + e^{i\pi\tilde{q}(a)} + e^{i\pi\tilde{q}(b)} + e^{i\pi\tilde{q}(a+b)}\right) \\
&= \frac{1}{2}\left(1 + e^{i\pi\tilde{q}(a)} + e^{i\pi\tilde{q}(b)} - e^{i\pi\tilde{q}(a)}e^{i\pi\tilde{q}(b)}\right),
\end{aligned} \tag{2.9}$$

where we have made use of (2.6) and noted that $\int a \cup b = 1$ for the two 1-cycles of the torus. Depending on the spin structure, one has $(\tilde{q}(a),\tilde{q}(b)) \in \{(0,0),(0,1),(1,0),(1,1)\}$, for which

we find that $(-1)^{\mathrm{Arf}(T^2)}$ assigns

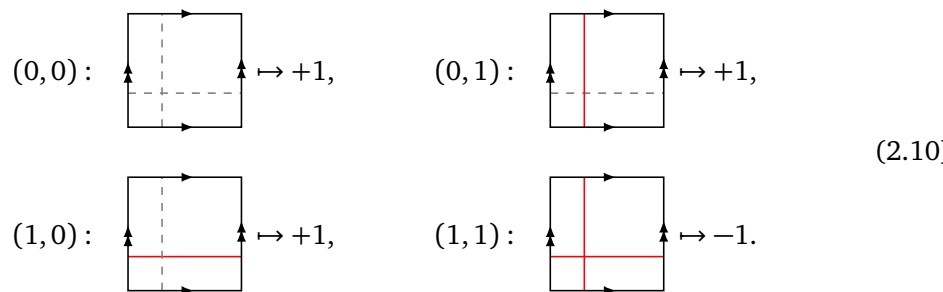

$$(2.10)$$

Here, we have represented the spin structure by lines on the torus — a grey dashed line means that fermions are anti-periodic i.e. NS in the normal direction, whereas solid red lines means that fermions are periodic i.e. R in the normal direction. From the perspective of canonical quantization the red lines can be interpreted as insertions of $(-1)^{\mathrm{f}}$ symmetry defects/operators.

On a manifold with boundary, the bulk invertible phase requires the presence of an odd number of Majorana fermions on each boundary. As reviewed in Section 4.1, there is no canonical way to quantize an odd-dimensional Clifford algebra. This can be thought of as an anomaly of the boundary system, which is compensated by the presence of the bulk invertible phase.

### 2.1.2 Invertible phase for $\mathrm{Spin} \times \mathbb{Z}_2$

We will also need to consider $\mathrm{Spin} \times \mathbb{Z}_2$-structures on $\Sigma$. These are given by a choice of spin structure $\sigma$ and a $\mathbb{Z}_2$ bundle with background gauge field $a \in H^1(\Sigma, \mathbb{Z}_2)$. As mentioned above, $H^1(\Sigma, \mathbb{Z}_2)$ acts on the space of spin structures, so the choice of $(\sigma, a)$ is equivalent to a choice of two separate spin structures $(\sigma_L, \sigma_R) := (\sigma, \sigma + a)$, where we take $\sigma_L$ and $\sigma_R$ to be the left- and right-moving spin structures. The corresponding invertible phases are classified by $\mho^2_{\mathrm{Spin}}(B\mathbb{Z}_2) = \mathbb{Z}_2^2$, which is generated by the separate Arf invariants for $\sigma_L$ and $\sigma_R$:

$$(-1)^{\mathrm{Arf}(\Sigma, \sigma_L)}, \quad (-1)^{\mathrm{Arf}(\Sigma, \sigma_R)}. \tag{2.11}$$

The discussion of each of these phases is identical to that in the previous section.

## 2.2 Unoriented invertible phases

We would now like to discuss invertible phases which can be formulated on unoriented manifolds. They can be thought of as phases protected by the action of time-reversal $\mathsf{T}$.

### 2.2.1 Invertible phase for orientation

Before considering fermionic phases protected by $\mathsf{T}$, let us discuss bosonic phases protected by $\mathsf{T}$. The structure group of the tangent bundle of an unoriented $d$-manifold $M$ is $O(d)$, which cannot be reduced to $SO(d)$. The obstruction to doing so is given by the first Stiefel-Whitney class of the tangent bundle $w_1 \in H^1(M, \mathbb{Z}_2)$. For a 1-cycle $C$, we have

$$\oint_C w_1 = \begin{cases} 0 & \text{if going around } C \text{ preserves orientation,} \\ 1 & \text{if going around } C \text{ reverses orientation.} \end{cases} \tag{2.12}$$

Bosonic unoriented phases on the worldsheet are classified by $\mho^2_O(pt) = \mathbb{Z}_2$, the generator of which is the low-energy limit of the $S = 1$ Haldane chain [36, 37]. In the continuum field

theory language, the effective action for this phase is[6]

$$e^{2\pi i S_{\text{eff}}(\Sigma)} = (-1)^{\int_\Sigma w_1^2}. \tag{2.13}$$

The generator of the bordism group $\Omega_2^O(pt)$ is the projective plane $\mathbb{RP}^2$, i.e. $e^{2\pi i S_{\text{eff}}(\mathbb{RP}^2)} = -1$. One can easily calculate the value of the action on any other manifold by counting the number of constituent $\mathbb{RP}^2$ appearing in its connected sum decomposition (mod 2). For instance, the Möbius strip $M_2$ is a connected sum of the disc and the projective plane, i.e. $M_2 \cong D_2 \# \mathbb{RP}^2$. Hence

$$e^{2\pi i S_{\text{eff}}(M_2)} = -1. \tag{2.14}$$

On the other hand the Klein Bottle $K_2$ is a connected sum of two copies of the projective plane, i.e. $K_2 \cong \mathbb{RP}^2 \# \mathbb{RP}^2$, and

$$e^{2\pi i S_{\text{eff}}(K_2)} = 1. \tag{2.15}$$

When considered on a manifold with boundary, this phase captures the time-reversal anomaly of the $(0+1)$d boundary theory. This anomaly can be carried by a bosonic Kramers doublet on the boundary, i.e. we have $\mathsf{T}^2 = -1$ instead of $\mathsf{T}^2 = +1$.

### 2.2.2 Pin structures

In order to describe fermionic invertible phases protected by $\mathsf{T}$, we need to briefly review how to put fermions on an unoriented manifold. There exist two different lifts of the $O(d)$ bundle, known as $\text{Pin}^\pm(d)$, which fit into the short exact sequence

$$0 \to \mathbb{Z}_2 \to \text{Pin}^\pm(d) \to O(d) \to 0. \tag{2.16}$$

Both $\text{Pin}^\pm(d)$ contain $\text{Spin}(d)$ as their component connected to the identity, and their difference lies in how time-reversal $\mathsf{T}$ and spatial reflection $\mathsf{R}$ lift,

$$
\begin{aligned}
\text{Pin}^+: &\quad \mathsf{T}^2 = (-1)^f, &\quad \mathsf{R}^2 = 1. \\
\text{Pin}^-: &\quad \mathsf{T}^2 = 1, &\quad \mathsf{R}^2 = (-1)^f.
\end{aligned}
\tag{2.17}
$$

The corresponding obstruction classes are

$$\text{Pin}^+: \quad w_2, \qquad\qquad \text{Pin}^-: \quad w_2 + w_1^2. \tag{2.18}$$

Every two-manifold $\Sigma$ has $w_2 + w_1^2 = 0 \mod 2$, and hence admits a $\text{Pin}^-$ structure. However, the same is not true for $\text{Pin}^+$. For instance, the real projective plane $\mathbb{RP}^2$ has $w_2 \neq 0$, and so it does not admit a $\text{Pin}^+$ structure.

The action of $\text{Pin}^\pm$ on fermions $\Psi$ can be given in terms of gamma matrices. In particular, reflection of the $i$-th coordinate acts on $\Psi$ by the gamma matrix $\gamma_i$. The reflection squared is trivial in $O(d)$, and therefore its lift when applied to a fermion is $\pm 1$. Then we have

$$\{\gamma_i, \gamma_j\} = \pm 2\eta_{ij}, \tag{2.19}$$

for $\text{Pin}^\pm$ structure, respectively, where we temporarily use the Lorentzian signature and the metric $\eta_{ij}$ is mostly plus. This explains why $\mathsf{T}$ squares to $(-1)^f$ when $\mathsf{R}$ squares to 1 and vice versa, as written in (2.17). The fermion fields $\Psi$ transforming in this manner are sometimes called "pinors."

---

[6]This is also occasionally written as $(-1)^{\chi(\Sigma)}$, where $\chi(\Sigma)$ is the Euler characteristic of $\Sigma$. The reason for this is that $w_1^2$ is equal to the mod-two reduction of the Euler class $e$, which satisfies $\int_\Sigma e = \chi(\Sigma)$. We prefer writing this phase in terms of Stiefel-Whitney classes in order to make bordism invariance manifest.

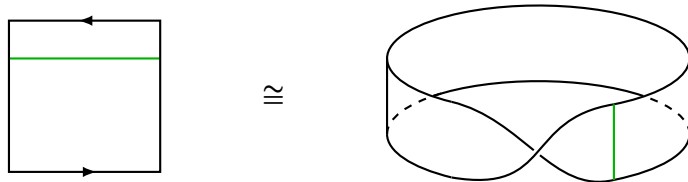

Figure 1: Möbius strip with an orientation-reversing line (in green).

In our study of unoriented string amplitudes, the behavior of pinors on the boundary of the Möbius strip with $Pin^{\pm}$ structure will be particularly important. Recall that on a circle, one can consistently define both anti-periodic and periodic fermions, i.e. fermions in the NS and R sectors. In contrast, the choice of NS or R on the boundary of the Möbius strip is fixed by the choice of $Pin^{\pm}$. We may see this as follows. Note that the Möbius strip can be constructed by taking a strip and gluing its ends together along an orientation-reversing line (Fig. 1) — upon crossing this line, we pick up an action of $\gamma_i$ on fermions. Traversing the boundary of the Möbius strip involves crossing this line twice, and this picks up an action of $\gamma_i^2 = \pm 1$ on fermions. Thus we conclude that boundary fermions on the $Pin^+$ Möbius strip are in the R sector, while those on the $Pin^-$ Möbius strip are in the NS sector.

### 2.2.3 Invertible phase for $Pin^-$ structure

We now study fermionic invertible phases protected by $T$ such that $T^2 = +1$. Such phases are classified by $\mho^2_{Pin^-}(pt) = \mathbb{Z}_8$, which is generated by the Arf-Brown-Kervaire (ABK) invariant [46]. For recent work on this invertible phase, see e.g. [47–49]. The ABK invariant can be thought of as the effective action of the Kitaev chain protected by time-reversal. In the continuum version, we have:

$$e^{2\pi i S_{\text{eff}}(\Sigma, \sigma)} = e^{\pi i \text{ABK}(\Sigma, \sigma)/4} = \frac{Z_{\text{ferm}}(m \gg 0; \Sigma, \sigma)}{Z_{\text{ferm}}(m \ll 0; \Sigma, \sigma)}. \tag{2.20}$$

Here $\sigma$ represents a choice of $Pin^-$ structure, which we will often omit from the argument of ABK for brevity. Alternatively, the ABK invariant can be defined combinatorially as

$$e^{\pi i \text{ABK}(\Sigma)/4} = \frac{1}{\sqrt{|H^1(\Sigma, \mathbb{Z}_2)|}} \sum_{a \in H^1(\Sigma, \mathbb{Z}_2)} e^{\pi i q(a)/2}, \tag{2.21}$$

where $q$ is a quadratic enhancement $q : H^1(\Sigma, \mathbb{Z}_2) \to \mathbb{Z}_4$ satisfying

$$q(a+b) - q(a) - q(b) = 2 \int_\Sigma a \cup b. \tag{2.22}$$

One may think of this $q$ as a sort of doubling of the spin structure quadratic form $\tilde{q} : H^1(\Sigma, \mathbb{Z}_2) \to \mathbb{Z}_2$ introduced earlier. Indeed, if the worldsheet is orientable then $q = 2\tilde{q} \pmod 4$. In that case one finds $\text{ABK}(\Sigma) = 4\,\text{Arf}(\Sigma) \pmod 8$, and (2.20) reduces to (2.2) as expected.

To see that the right-hand side of (2.21) is an eighth root of unity, again we consider its

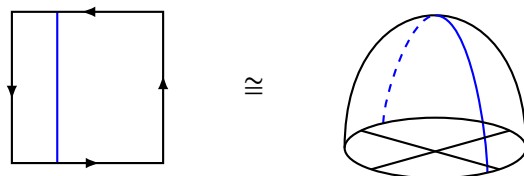

Figure 2: Projective plane with the single generator $Z$ of its first homology class (in blue). To the right this is drawn as a sphere with a crosscap.

square:

$$
\begin{aligned}
\mathrm{RHS}^2 &= \frac{1}{|H^1(\Sigma, \mathbb{Z}_2)|} \sum_{a,b \in H^1(\Sigma, \mathbb{Z}_2)} i^{q(a)+q(b)} \\
&= \frac{1}{|H^1(\Sigma, \mathbb{Z}_2)|} \sum_{a,b \in H^1(\Sigma, \mathbb{Z}_2)} i^{q(a+b)} (-1)^{\int_\Sigma a \cup b} \\
&= \frac{1}{|H^1(\Sigma, \mathbb{Z}_2)|} \sum_{a,c \in H^1(\Sigma, \mathbb{Z}_2)} i^{q(c)} (-1)^{\int_\Sigma (a \cup c + a \cup a)} \\
&= \frac{1}{|H^1(\Sigma, \mathbb{Z}_2)|} \sum_{a,c \in H^1(\Sigma, \mathbb{Z}_2)} i^{q(c)} (-1)^{\int_\Sigma (a \cup c + a \cup w_1)} \\
&= i^{q(w_1)},
\end{aligned}
\tag{2.23}
$$

where we used $\int a \cup a = \int a \cup w_1$.

We will need some basic values for the ABK invariant. We begin by reviewing the case of $\mathbb{RP}^2$, for which there is a single generator $Z$ of $H_1(\mathbb{RP}^2, \mathbb{Z}_2)$, as depicted in Fig. 2. Its Poincaré dual, $z$, is an unoriented cocycle with $\int_{\mathbb{RP}^2} z \cup z = 1$. Then using (2.22) and $q(0) = 0$, we conclude that $2q(z) + 2 = 0 \mod 4$ and hence $q(z) = 1$ or $3$. These label the two distinct $\mathrm{Pin}^-$ structures on $\mathbb{RP}^2$. The corresponding ABK invariants are easily calculated,

$$
\begin{aligned}
1: \quad & e^{\pi i \mathrm{ABK}(\mathbb{RP}^2)/4} = \frac{1}{\sqrt{2}} \left(1 + e^{i\pi/2}\right) = e^{i\pi/4}, \\
3: \quad & e^{\pi i \mathrm{ABK}(\mathbb{RP}^2)/4} = \frac{1}{\sqrt{2}} \left(1 + e^{i3\pi/2}\right) = e^{-i\pi/4}.
\end{aligned}
\tag{2.24}
$$

We will also need the value of the ABK invariant on the Klein bottle. Because $K_2$ is a connected sum of two copies of $\mathbb{RP}^2$, it admits four choices of quadratic enhancement. Taking $z_1$ and $z_2$ to be the basis naturally adapted to the connected sum gives $(q(z_1), q(z_2)) \in \{(1,1), (1,3), (3,1), (3,3)\}$. However, it will behoove us to switch to the more familiar basis of $H_1(K_2, \mathbb{Z}_2)$, for which $a := z_1 + z_2$ is Poincaré dual to the orientation-preserving $A$-cycle while $b := z_2$ is dual to the orientation-reversing $B$-cycle. The quadratic enhancements are then $(q(a), q(b)) \in \{(0,1), (0,3), (2,1), (2,3)\}$. For each of these $\mathrm{Pin}^-$ structures we compute the following values for the ABK invariant,

$$
\begin{aligned}
(0,1): \quad & e^{\pi i \mathrm{ABK}(K_2)/4} = \frac{1}{2}(1 + 1 + i - i) = 1, \\
(0,3): \quad & e^{\pi i \mathrm{ABK}(K_2)/4} = \frac{1}{2}(1 + 1 - i + i) = 1, \\
(2,1): \quad & e^{\pi i \mathrm{ABK}(K_2)/4} = \frac{1}{2}(1 - 1 + i + i) = i, \\
(2,3): \quad & e^{\pi i \mathrm{ABK}(K_2)/4} = \frac{1}{2}(1 - 1 - i - i) = -i.
\end{aligned}
\tag{2.25}
$$

It is useful to note that

$$e^{\pi i (4\mathrm{ABK}(\Sigma))/4} = (-1)^{\int w_1^2},  \tag{2.26}$$

namely that four copies of the ABK theory is the same as the nontrivial bosonic unoriented phase discussed in Section 2.2.1.

The boundary theory of $n$ copies of the ABK theory is detailed in Section 4.1. When $n = 4$, we have a system of four Majorana fermions $\chi_i$. We can then introduce a T-invariant interaction term $\chi_1 \chi_2 \chi_3 \chi_4$. The ground state with this interaction is bosonic, two-fold degenerate, and has the projective anomaly $T^2 = -1$, thus manifesting the equality (2.26) on the boundary. Similarly, when $n = 8$, we can consider a suitable quartic interaction under which the ground state is unique [35].

### 2.2.4 Invertible phase for $\mathrm{Pin}^+$ structure

We now study fermionic invertible phases protected by T such that $T^2 = (-1)^f$. Such phases are classified by $\mho^2_{\mathrm{Pin}^+}(pt) = \mathbb{Z}_2$, which is generated by the Arf invariant (2.7) on the orientation double cover $\hat{\Sigma}$ of the worldsheet, i.e. $(-1)^{\mathrm{Arf}(\hat{\Sigma})}$ [9]. This gives $\pm 1$ on the Klein bottle, depending on the spin structure of its double cover torus. Concretely, the Klein bottle admits four $\mathrm{Pin}^+$ structures, two of which have NS sector fermions along the oriented $A$-cycle of $K_2$, and two of which have R sector fermions along the $A$-cycle. For each of these $\mathrm{Pin}^+$ structures, we must determine the corresponding spin structure on the orientation double cover, i.e. the torus. To do so, let us consider for simplicity a rectangular torus with real coordinates $x, y$ satisfying $x \sim x + 2\pi$, $y \sim y + 2L$ for some $L$. The original Klein bottle is obtained by imposing the identification $(x, y) \sim (-x, y + L)$. The cycle in the $x$-direction is the oriented $A$-cycle on $K_2$, while the cycle in the $y$-direction is twice the unoriented $B$-cycle. From this we learn that the choice of $\mathrm{Pin}^+$ structure along the $B$-cycle is irrelevant on the double cover — the fermion is always periodic along the cycle in the $y$-direction. Thus we conclude that the four $\mathrm{Pin}^+$ structures are assigned phases $(-1)^{\mathrm{Arf}(\hat{\Sigma})} = 1, 1, -1, -1$.

## 3 GSO projections

In this section, we consider the addition of fermionic invertible phases to the worldsheet theories of various superstrings. This will allow us to enumerate all possible GSO projections. We will focus on oriented Type II and Type 0 theories, as well as unoriented Type 0 theories. The case of unoriented Type II (i.e. Type I) strings will be treated in Section 6. In our discussion we will work in the NSR formalism, which is briefly reviewed in Appendix A. The properties of the low-lying states in this formalism are summarized in Table 2 for convenience.

### 3.1 Oriented strings

#### 3.1.1 Type 0

Let us begin by studying oriented Type 0 superstrings, which differ from Type II strings in that there is a single spin structure for both left- and right-movers [6,50]. The SPT phases which can be consistently realized on the Type 0 worldsheets are classified by $\mho^2_{\mathrm{Spin}}(pt) = \mathbb{Z}_2$. As we have seen above, the effective action for the non-trivial phase can be written in terms of the Arf invariant (2.2). Depending on whether or not one allows for this non-trivial phase on the worldsheet, one expects to arrive at two different Type 0 theories, with torus partition functions

$$Z^{(n)} = \frac{1}{2} \sum_{\sigma} (-1)^{n\,\mathrm{Arf}(T^2, \sigma)} Z[\sigma]\overline{Z}[\sigma]\,, \qquad n = 0, 1\,,  \tag{3.1}$$

Table 2: Fermion parity and representations of the low-lying states of the closed NSR superstring.

| State | $(-1)^f$ | $(-1)^{f_L}$ | $(-1)^{f_R}$ | Little group rep. |
|---|---|---|---|---|
| $(NS-, NS-)$ | $+1$ | $-1$ | $-1$ | $\mathbf{1}$ |
| $(NS+, NS+)$ | $+1$ | $+1$ | $+1$ | $\mathbf{8}_v \otimes \mathbf{8}_v = \mathbf{1} \oplus \mathbf{28} \oplus \mathbf{35}$ |
| $(R+, R+)$ | $+1$ | $+1$ | $+1$ | $\mathbf{8}_s \otimes \mathbf{8}_s = \mathbf{1} \oplus \mathbf{28} \oplus \mathbf{35}_-$ |
| $(R-, R-)$ | $+1$ | $-1$ | $-1$ | $\mathbf{8}_c \otimes \mathbf{8}_c = \mathbf{1} \oplus \mathbf{28} \oplus \mathbf{35}_+$ |
| $(R+, R-)$ | $-1$ | $+1$ | $-1$ | $\mathbf{8}_s \otimes \mathbf{8}_c = \mathbf{8}_v \oplus \mathbf{56}$ |
| $(R-, R+)$ | $-1$ | $-1$ | $+1$ | $\mathbf{8}_c \otimes \mathbf{8}_s = \mathbf{8}_v \oplus \mathbf{56}$ |
| $(NS+, R+)$ | $+1$ | $+1$ | $+1$ | $\mathbf{8}_v \otimes \mathbf{8}_s = \mathbf{8}_c \oplus \mathbf{56}_s$ |
| $(NS+, R-)$ | $-1$ | $+1$ | $-1$ | $\mathbf{8}_v \otimes \mathbf{8}_c = \mathbf{8}_s \oplus \mathbf{56}_c$ |

where $Z[\sigma], \overline{Z}[\sigma]$ are the standard left- and right-moving worldsheet torus partition functions with spin structure $\sigma$. The two theories obtained in this way are Type 0B ($n = 0$) and Type 0A ($n = 1$),

$$
\begin{aligned}
0A: \quad Z^{(1)} &= \frac{1}{2} \left( |Z[\sigma_{00}]|^2 + |Z[\sigma_{01}]|^2 + |Z[\sigma_{10}]|^2 - |Z[\sigma_{11}]|^2 \right), \\
0B: \quad Z^{(0)} &= \frac{1}{2} \left( |Z[\sigma_{00}]|^2 + |Z[\sigma_{01}]|^2 + |Z[\sigma_{10}]|^2 + |Z[\sigma_{11}]|^2 \right),
\end{aligned}
\tag{3.2}
$$

and their massless RR states are found to be

$$
\begin{aligned}
0A: \quad &|0\rangle_{RR}^{a\dot{b}}, |0\rangle_{RR}^{\dot{a}b} \in (\mathbf{8}_s \otimes \mathbf{8}_c) \oplus (\mathbf{8}_c \otimes \mathbf{8}_s) = (2 \cdot \mathbf{8}_v) \oplus (2 \cdot \mathbf{56}), \\
0B: \quad &|0\rangle_{RR}^{ab}, |0\rangle_{RR}^{\dot{a}\dot{b}} \in (\mathbf{8}_s \otimes \mathbf{8}_s) \oplus (\mathbf{8}_c \otimes \mathbf{8}_c) = (2 \cdot \mathbf{1}) \oplus (2 \cdot \mathbf{28}) \oplus \mathbf{70}.
\end{aligned}
\tag{3.3}
$$

The massless RR sector for Type 0A contains two 1-forms and two 3-form fields, while that of Type 0B contains two scalars, two 2-forms, and a 4-form with no self-duality constraint. This is exactly double the RR content of the corresponding Type II theories. This leads one to expect a doubled brane spectrum, where for each $p$ we have both D$p$ and D$p'$ branes. This will be discussed further in Section 3.3.

### 3.1.2 Type II

We now proceed to discuss the more familiar Type II superstrings. A characteristic feature of these strings is that they have separate spin structures for left- and right-movers, which means that the worldsheet is endowed with a Spin $\times \mathbb{Z}_2$ structure. The anomalies and invertible phases on such worldsheets are captured by $\mho^3_{\mathrm{Spin}}(B\mathbb{Z}_2)$ and $\mho^2_{\mathrm{Spin}}(B\mathbb{Z}_2)$, respectively. These groups are listed in Table 1. As discussed in [51, 52], the fact that $\mho^3_{\mathrm{Spin}}(B\mathbb{Z}_2) = \mathbb{Z}_8$ implies that the number of physical pairs of left- and right-movers needs to be a multiple of eight to have a non-anomalous GSO projection. This is indeed the case if the physical string lives in ten dimensions.

On the other hand, the SPT phases on the worldsheet are classified by $\mho^2_{\mathrm{Spin}}(B\mathbb{Z}_2) = (\mathbb{Z}_2)^2$. The two $\mathbb{Z}_2$ can be interpreted as separate left- and right-moving fermionic invertible phases, as discussed in Section 2.1.2. In other words, the partition functions for these phases are given by

$$
(-1)^{n_L \mathrm{Arf}(\Sigma, \sigma_L) + n_R \mathrm{Arf}(\Sigma, \sigma_R)}, \qquad n_{L,R} = 0, 1,
\tag{3.4}
$$

where $\sigma_{L(R)}$ is the left(right)-moving spin structure on the worldsheet $\Sigma$. The corresponding torus partition functions for these theories are given by

$$Z^{(n_L, n_R)} = \frac{1}{4}\left(\sum_{\sigma_L}(-1)^{n_L\,\mathrm{Arf}(T^2, \sigma_L)}Z[\sigma_L]\right) \times \left(\sum_{\sigma_R}(-1)^{n_R\,\mathrm{Arf}(T^2, \sigma_R)}\overline{Z}[\sigma_R]\right). \quad (3.5)$$

For instance, two cases are

$$Z^{(0,0)} = \frac{1}{4}\left(Z[\sigma_{00}] + Z[\sigma_{01}] + Z[\sigma_{10}] + Z[\sigma_{11}]\right)\left(\overline{Z}[\sigma_{00}] + \overline{Z}[\sigma_{01}] + \overline{Z}[\sigma_{10}] + \overline{Z}[\sigma_{11}]\right),$$

$$Z^{(0,1)} = \frac{1}{4}\left(Z[\sigma_{00}] + Z[\sigma_{01}] + Z[\sigma_{10}] + Z[\sigma_{11}]\right)\left(\overline{Z}[\sigma_{00}] + \overline{Z}[\sigma_{01}] + \overline{Z}[\sigma_{10}] - \overline{Z}[\sigma_{11}]\right).$$
$$(3.6)$$

While there are seemingly four distinct SPT phases, there are in fact only two physically distinct Type II theories. To see this, recall the continuum definition of the Arf invariant:

$$(-1)^{\mathrm{Arf}(\Sigma, \sigma)} = \frac{Z_{\mathrm{ferm}}(m \gg 0; \Sigma, \sigma)}{Z_{\mathrm{ferm}}(m \ll 0; \Sigma, \sigma)}, \quad (3.7)$$

where $Z_{\mathrm{ferm}}(m; \Sigma, \sigma)$ is the partition function of a free massive Majorana fermion of mass $m$. We note this formula holds at finite mass as well,

$$(-1)^{\mathrm{Arf}(\Sigma, \sigma)} = \frac{Z_{\mathrm{ferm}}(+m; \Sigma, \sigma)}{Z_{\mathrm{ferm}}(-m; \Sigma, \sigma)}, \quad (3.8)$$

which was already used in (2.4). In other words, upon flipping the sign of a mass term, $m \to -m$, one generates a factor of $(-1)^{\mathrm{Arf}(\Sigma, \sigma)}$ in the partition function. Note that such a flip of the mass term can be performed by $(\psi, \tilde{\psi}) \to (\psi, -\tilde{\psi})$. Taking the limit $m \to 0$, we find that a Majorana-Weyl fermion $\tilde{\psi}$ has an anomaly under $\tilde{\psi} \to -\tilde{\psi}$, and generates $(-1)^{\mathrm{Arf}(\Sigma, \sigma_R)}$. This in particular means that the parity transformation along a single spacetime direction, say $(\psi^{\mu=9}, \tilde{\psi}^{\mu=9}) \to (-\psi^9, -\tilde{\psi}^9)$, produces $(-1)^{\mathrm{Arf}(\Sigma, \sigma_L)+\mathrm{Arf}(\Sigma, \sigma_R)}$, i.e. $n_L = n_R = 1$ in (3.4). Therefore, there are only essentially two distinct Type II GSO projections, with the others being related by spacetime parity transformation. The cases $(n_L, n_R) = (0, 0), (1, 1)$ are traditionally called Type IIB while the cases $(n_L, n_R) = (0, 1), (1, 0)$ are called Type IIA.

The reasoning above also explains why T-duality exchanges Type IIA/B. Recall that T-duality along a spacetime direction keeps $(\partial X, \psi)$ fixed and implements $(\bar{\partial} X, \tilde{\psi}) \to (-\bar{\partial} X, -\tilde{\psi})$. Then by the previous paragraph, this generates $(-1)^{\mathrm{Arf}(\Sigma, \sigma_R)}$, exchanging Type IIA/B. By the same arguments, the two Type 0 theories are also exchanged by T-duality.

### 3.1.3 Comments on the two points of view on the effect of invertible phases

There are two ways of understanding the gauging of a global symmetry in the presence of a non-trivial invertible phase. The point of view which we have taken so far is to take the tensor product of the original theory and the invertible phase, and to then gauge the relevant symmetry. The Hilbert space of the invertible phase is one-dimensional, and therefore this changes the way the global symmetry acts on the states of the original theory. For example, in the case of Type 0 strings, we used the projectors

$$P_{0A} = \frac{1}{2}(1 + (-1)^f|_{0A}), \qquad P_{0B} = \frac{1}{2}(1 + (-1)^f|_{0B}) \quad (3.9)$$

and what produced the difference between the two was that on the RR sector, we had

$$(-1)^f|_{0A} = -(-1)^f|_{0B} \quad (3.10)$$

due to the presence of the Arf theory.

More traditionally, the action of $(-1)^f$ was fixed once and for all, for example to be equal to $(-1)^f|_{0B}$, and different GSO projections were said to correspond to different projectors. For example, in the RR sector, one would have written

$$P_{0A}^{RR} = \frac{1}{2}(1 - (-1)^f), \qquad P_{0B}^{RR} = \frac{1}{2}(1 + (-1)^f). \tag{3.11}$$

These two points of view clearly lead to the same results, and similar statements will be seen to hold for unoriented strings. Though we will briefly discuss this traditional viewpoint when we compare to the existing literature, we will mostly use the first point of view.

## 3.2 Unoriented strings

### 3.2.1 Orientation reversal on fermions and ground states

We now consider unoriented string theories. One way to obtain such theories is to gauge time-reversal symmetry $\mathsf{T}$ on the worldsheet. $\mathsf{T}$ is an antiunitary symmetry that acts on worldsheet fermions as

$$\mathsf{T}\psi(t,\sigma)\mathsf{T}^{-1} = \tilde{\psi}(-t,\sigma)\,, \qquad \mathsf{T}\tilde{\psi}(t,\sigma)\mathsf{T}^{-1} = \psi(-t,\sigma)\,, \tag{3.12}$$

with $\mathsf{T}^2 = 1$. In string theory, it is often more common to describe this in terms of worldsheet parity $\Omega$, which is a unitary symmetry whose action is given by

$$\Omega\psi(t,\sigma)\Omega^{-1} = -\tilde{\psi}(t, 2\pi - \sigma)\,, \qquad \Omega\tilde{\psi}(t,\sigma)\Omega^{-1} = \psi(t, 2\pi - \sigma)\,. \tag{3.13}$$

From this definition it is clear that $\Omega^2 = (-1)^f$. The ability to choose between $\mathsf{T}$ or $\Omega$ is a consequence of the CPT theorem. The fact that $\mathsf{T}^2 = 1$, or equivalently that $\Omega^2 = (-1)^f$, means that we are working with a Pin$^-$ structure on the worldsheet. In this case the action of $\Omega$ on the ground states in the NSNS and RR sectors can be taken to be

$$\Omega|0\rangle_{NSNS} = |0\rangle_{NSNS}\,, \qquad \Omega|0\rangle_R \otimes |\tilde{0}\rangle_R = \begin{cases} - |\tilde{0}\rangle_R \otimes |0\rangle_R & \text{for} \quad (R\pm, R\pm), \\ -i|\tilde{0}\rangle_R \otimes |0\rangle_R & \text{for} \quad (R\pm, R\mp). \end{cases} \tag{3.14}$$

One can also consider gauging $\Omega$ twisted by some $\mathbb{Z}_2$ symmetry. Here we consider $\Omega_f := \Omega(-1)^{f_L}$. We find

$$(\Omega_f)^2 = \Omega(-1)^{f_L}\Omega(-1)^{f_L} = \Omega(-1)^{f_L+f_R}\Omega = (-1)^f\Omega^2 = 1 \tag{3.15}$$

and hence gauging this operator gives Pin$^+$ structure on worldsheets. In this case parity acts on the NSNS and RR ground states as

$$\Omega_f|0\rangle_{NSNS} = -|0\rangle_{NSNS}\,, \qquad \Omega_f|0\rangle_R \otimes |\tilde{0}\rangle_R = \begin{cases} \mp |\tilde{0}\rangle_R \otimes |0\rangle_R & \text{for} \quad (R\pm, R\pm), \\ \mp i|\tilde{0}\rangle_R \otimes |0\rangle_R & \text{for} \quad (R\pm, R\mp). \end{cases} \tag{3.16}$$

Unlike for Type II strings where $\Omega$ is a symmetry of only Type IIB, for Type 0 strings $\Omega$ is a symmetry of both Type 0A and 0B. Hence we can obtain Pin$^-$ Type 0 theories by starting from either Type 0A or 0B and gauging $\Omega$. Likewise, one might expect that we can obtain Pin$^+$ Type 0 theories by starting from either Type 0A or 0B and gauging $\Omega_f$. However, it turns out that $\Omega_f$ cannot be consistently gauged in Type 0A, since it is incompatible with the Type 0A spin structure projection [53–55].

In the rest of this section we study the consistent unoriented Type 0 strings in more detail. We begin by analyzing the Pin$^-$ strings in Section 3.2.2, and then proceed to a discussion of the Pin$^+$ strings in Section 3.2.3. To the best of our knowledge, many of these theories have not been discussed in the literature — some preliminary works include [53–62]. For a condensed matter perspective, see e.g. [63, 64].

### 3.2.2   Pin⁻ **Type 0 Strings**

Let us begin by discussing Pin⁻ Type 0 strings. The group classifying the relevant invertible phases is $\mho^2_{\text{Pin}^-}(pt) = \mathbb{Z}_8$, which is generated by the ABK invariant. We are thus led to predict the existence of eight Pin⁻ theories, each distinguished by the presence of $n = 0, \ldots, 7$ copies of ABK on the worldsheet.

In unoriented theories, the presence of a non-trivial invertible phase manifests itself in the action of $\Omega$ on the different closed string ground states. In order to understand this action, we make use of the values of the ABK invariant on the Klein bottle $K_2$ obtained in Section 2.2.3. In particular, it was found there that the Klein bottle admits four Pin⁻ structures labelled by quadratic enhancements $(q(a), q(b)) \in \{(0,1),(0,3),(2,1),(2,3)\}$ with respective values $e^{i\pi\text{ABK}(K_2)/4} = 1, 1, i, -i$. Recall that the first entry $a$ corresponds to the orientation-preserving cycle on $K_2$, while the second entry $b$ corresponds to the orientation-reversing cycle.

We now want to interpret these results as the action of $\Omega$ and $(-1)^f$ on the closed string Hilbert space. This can be done as follows. We begin by cutting $K_2$ along the orientation-preserving cycle $A$ to obtain a cylinder with an insertion of an $\Omega$ symmetry line. Since $A$ is orientation-preserving, we know that $q = 2\tilde{q} \mod 4$, where $\tilde{q}(a)$ is the spin structure along $A$. Consequently, the first and second Pin⁻ structures correspond to NS structure along the $A$-cycle, while the third and fourth correspond to R structure along the $A$-cycle. We may then interpret the partition function for each Pin⁻ structure as the following traces on the torus. We have:

$$
\begin{aligned}
(0,1): &\quad \longleftrightarrow \quad \frac{1}{4}\text{Tr}_{\text{NSNS}}\left[\Omega\, e^{-2\pi l H_{\text{cl}}}\right]\\[2mm]
(0,3): &\quad \longleftrightarrow \quad \frac{1}{4}\text{Tr}_{\text{NSNS}}\left[\Omega(-1)^f\, e^{-2\pi l H_{\text{cl}}}\right]\\[2mm]
(2,1): &\quad \longleftrightarrow \quad \frac{1}{4}\text{Tr}_{\text{RR}}\left[\Omega\, e^{-2\pi l H_{\text{cl}}}\right]\\[2mm]
(2,3): &\quad \longleftrightarrow \quad \frac{1}{4}\text{Tr}_{\text{RR}}\left[\Omega(-1)^f\, e^{-2\pi l H_{\text{cl}}}\right]
\end{aligned}
\tag{3.17}
$$

where green lines represent orientation-reversal lines and red lines represent spin lines. In this way, the value of the ABK invariant on $K_2$ with Pin⁻ structure labeled by $(q(a), q(b))$ can be assigned to the action of $\Omega$ on the ground states with the appropriate cylinder spin structures. In particular, we conclude that $\Omega$ acts trivially on NS ground states, whereas it acts with an extra factor of $i$ on R ground states. This implies that the presence of $n$ copies of ABK changes the action of $\Omega$ on the RR sector ground states by a factor of $i^n$ relative to (3.14), giving

$$
\Omega|0\rangle_{\text{NSNS}} = |0\rangle_{\text{NSNS}}\,, \qquad \Omega|0\rangle_R \otimes |\tilde{0}\rangle_R = \begin{cases} -i^n\; |\tilde{0}\rangle_R \otimes |0\rangle_R & \text{for} \quad (\text{R}\pm, \text{R}\pm),\\ -i^{n+1}|\tilde{0}\rangle_R \otimes |0\rangle_R & \text{for} \quad (\text{R}\pm, \text{R}\mp). \end{cases}
\tag{3.18}
$$

Note that upon shifting $n \to n + 1$, the additional factor of $i$ changes the fermion-parity of the RR ground state, since $\Omega^2 = (-1)^f$. Since the Type 0A/B theories differ by a projection onto states of worldsheet fermion number $(-1)^f = \pm 1$, we see that theories with

even $n$ correspond to orientifolds of Type 0B, while theories with odd $n$ correspond to orientifolds of Type 0A. This is also supported by recalling that on oriented manifolds $\Sigma$, the ABK invariant reduces to $\text{ABK}(\Sigma) = 4\text{Arf}(\Sigma)$ (mod 8), and hence the partition function becomes $e^{in\pi\text{ABK}(\Sigma)/4} = (-1)^{n\,\text{Arf}(\Sigma)}$, which is precisely what distinguished the oriented Type 0A/B theories.

As far as the action of $\Omega$ on the vacuum (3.18) is concerned, theories differing by four copies of ABK are indistinguishable. The reason for this is that only data about the Klein bottle $K_2$ was used to obtain (3.18). However, the manifold that generates the bordism group $\Omega_2^{\text{Pin}^-}(pt)$ is the projective plane $\mathbb{RP}^2$, while $K_2$ is a connected sum of two copies thereof, i.e. $K_2 \cong \mathbb{RP}^2 \# \mathbb{RP}^2$. Consequently, the partition function on $K_2$ is insensitive to an additional sign that can arise on manifolds whose decompositions contain an odd number of copies of $\mathbb{RP}^2$. Indeed, four copies of the ABK theory is not trivial and gives partition function $e^{4\pi i\text{ABK}(\Sigma)/4} = (-1)^{\int_\Sigma w_1^2}$, as discussed in Section 2.2.3. As described in Section 2.2.1, unlike for the Klein bottle the Möbius strip amplitude *is* sensitive to this sign. This sign turns out to give precisely the difference between O9$^\pm$ orientifolds. More detail on this will be given in Section 5.1.

We may now study the closed string spectra of these theories. The action of $\Omega$ proposed in (3.18) does not project out the closed string tachyon in the NSNS sector, but has the following implications for the spectra of RR fields. For $n$ even, $\Omega$ projects out all $(\text{R}\pm, \text{R}\mp)$ states. The cases $n = 0 \mod 4$ and $n = 2 \mod 4$ differ by a sign in the action of $\Omega$, which projects out the symmetric or antisymmetric combinations of $(\text{R}\pm, \text{R}\pm)$ states. Then upon gauging $\Omega$ we obtain the following RR spectra,

$$
\begin{aligned}
n = 0,4: \quad & |0\rangle_{\text{RR}}^{[ab]} \in \mathbf{28} \subset \mathbf{8}_s \otimes \mathbf{8}_s, & & |0\rangle_{\text{RR}}^{[\dot{a}\dot{b}]} \in \mathbf{28} \subset \mathbf{8}_c \otimes \mathbf{8}_c, \\
n = 2,6: \quad & |0\rangle_{\text{RR}}^{(ab)} \in \mathbf{1} \oplus \mathbf{35}_- \subset \mathbf{8}_s \otimes \mathbf{8}_s, & & |0\rangle_{\text{RR}}^{(\dot{a}\dot{b})} \in \mathbf{1} \oplus \mathbf{35}_+ \subset \mathbf{8}_c \otimes \mathbf{8}_c.
\end{aligned}
\tag{3.19}
$$

In the $n = 0 \mod 4$ cases, only the two 2-forms survive the projection, while for $n = 2 \mod 4$ only the two scalars and the 4-form survive. These spectra of RR fields are indeed a projection of the Type 0B ones. For $n$ odd, the extra factor of $i$ in (3.18) projects out all the $(\text{R}\pm, \text{R}\pm)$ states, while the $(\text{R}\pm, \text{R}\mp)$ combinations

$$
\begin{aligned}
n = 1,5: \quad & \frac{1}{\sqrt{2}}(|0\rangle_{\text{RR}}^{a\dot{b}} + |0\rangle_{\text{RR}}^{\dot{b}a}) \in \mathbf{8}_v \oplus \mathbf{56} \subset (\mathbf{8}_s \otimes \mathbf{8}_c) \oplus (\mathbf{8}_c \otimes \mathbf{8}_s), \\
n = 3,7: \quad & \frac{1}{\sqrt{2}}(|0\rangle_{\text{RR}}^{a\dot{b}} - |0\rangle_{\text{RR}}^{\dot{b}a}) \in \mathbf{8}_v \oplus \mathbf{56} \subset (\mathbf{8}_s \otimes \mathbf{8}_c) \oplus (\mathbf{8}_c \otimes \mathbf{8}_s),
\end{aligned}
\tag{3.20}
$$

survive the projection. This leaves a single set of 1- and 3-form fields. These states are part of the Type 0A spectrum.

It is worth mentioning that because these theories possess neither spacetime fermions nor (anti-)self-dual form fields, they are all free of perturbative gravitational anomalies.

Finally, let us give a more traditional orientifold interpretation to the theories studied in this section. In perturbative string theory we often refer not only to left/right-moving worldsheet fermion number $(-1)^{f_L, f_R}$ but also to left/right-moving spacetime fermion number $(-1)^{F_L, F_R}$. We recall that $(-1)^F$ acts by $+1$ on the NS sector and by $-1$ on the R sector. We now consider $\Omega_F := \Omega(-1)^{F_L}$, which acts with an extra minus sign on the left-moving R sector. Above, we saw that gauging $\Omega$ with two copies of the ABK theory gives the same minus sign. This suggests the following identifications,

$$
\begin{aligned}
n &= 0,4: \quad (0B, \Omega) & n &= 1,5: \quad (0A, \Omega) \\
n &= 2,6: \quad (0B, \Omega_F) & n &= 3,7: \quad (0A, \Omega_F),
\end{aligned}
\tag{3.21}
$$

where the first element in parenthesis denotes the starting theory, and the second element denotes the operator being gauged. The difference between theories differing by 4 copies of

ABK is the action of $\Omega$ or $\Omega_F$ on Chan-Paton factors. This correspondence between the ABK viewpoint and the orientifold viewpoint will be discussed further in Section 5.1.

### 3.2.3 $\text{Pin}^+$ **Type 0 Strings**

We finally proceed to the case of $\text{Pin}^+$ Type 0 strings, which were studied in [65,66]. The group capturing potential invertible phases on the $\text{Pin}^+$ worldsheet is $\mho^2_{\text{Pin}^+}(pt) = \mathbb{Z}_2$. As reviewed in Section 2.2.4, the effective action for this invertible phase is given by the Arf invariant of the oriented double cover $\hat{\Sigma}$ of the worldsheet, i.e. $(-1)^{\text{Arf}(\hat{\Sigma})}$, whose generating manifold is the Klein bottle. The Klein bottle was seen to admit four $\text{Pin}^+$ structures, which we now label as $(0,1),(0,3),(2,1),(2,3)$ in analogy to the $\text{Pin}^-$ notation.[7] By examining the spin structure on the double cover torus, these were assigned respective phases $(-1)^{\text{Arf}(\hat{\Sigma})} = 1,1,-1,-1$.

We now proceed as in the $\text{Pin}^-$ case above. First, we recast the Klein bottle partition functions for the four $\text{Pin}^+$ structures in terms of traces on the torus. This gives

$$
\begin{aligned}
(0,1) &\leftrightarrow \frac{1}{4}\text{Tr}_{\text{NSNS}}\left[\Omega_f \, e^{-2\pi l H_{\text{cl}}}\right], & (0,3) &\leftrightarrow \frac{1}{4}\text{Tr}_{\text{NSNS}}\left[\Omega_f \, (-1)^f \, e^{-2\pi l H_{\text{cl}}}\right], \\
(2,1) &\leftrightarrow \frac{1}{4}\text{Tr}_{\text{RR}}\left[\Omega_f \, e^{-2\pi l H_{\text{cl}}}\right], & (2,3) &\leftrightarrow \frac{1}{4}\text{Tr}_{\text{RR}}\left[\Omega_f \, (-1)^f \, e^{-2\pi l H_{\text{cl}}}\right].
\end{aligned}
\tag{3.22}
$$

The $\text{Arf}(\hat{\Sigma})$ invertible phase assigns $-1$ to the Klein bottle with $\text{Pin}^+$ structure $(2,1)$ and $(2,3)$, and $+1$ to the other $\text{Pin}^+$ structures. This means that the presence of the non-trivial invertible phase changes the action of $\Omega_f$ on R sector ground states by a sign relative to (3.16), but does not change the action of $(-1)^f$.

With this information, we may turn towards the analysis of the massless closed string spectra of the theories. For the trivial phase, the orientifold projection keeps the symmetric combinations of (R−,R−) and antisymmetric contributions of (R+,R+) in the Type 0B spectra. In the non-trivial phase, one instead keeps the antisymmetric combinations of (R−,R−) and symmetric contributions of (R+,R+),

$$
\begin{aligned}
n = 0: & \quad |0\rangle^{[ab]}_{\text{RR}} \in \mathbf{28} \subset \mathbf{8}_s \otimes \mathbf{8}_s, & |0\rangle^{(\dot{a}\dot{b})}_{\text{RR}} \in \mathbf{1} \oplus \mathbf{35}_+ \subset \mathbf{8}_c \otimes \mathbf{8}_c, \\
n = 1: & \quad |0\rangle^{(ab)}_{\text{RR}} \in \mathbf{1} \oplus \mathbf{35}_- \subset \mathbf{8}_s \otimes \mathbf{8}_s, & |0\rangle^{[\dot{a}\dot{b}]}_{\text{RR}} \in \mathbf{28} \subset \mathbf{8}_c \otimes \mathbf{8}_c.
\end{aligned}
\tag{3.23}
$$

We note that these spectra are the same up to a spacetime parity transformation which exchanges the self-dual and anti-self-dual 4-forms. This observation can also be explained from the fact that the spacetime parity transformation generates $(-1)^{\text{Arf}(\hat{\Sigma})}$ on the worldsheet. Indeed, in the Type II case, the same operation generated $(-1)^{\text{Arf}(\Sigma,\sigma_L)+\text{Arf}(\Sigma,\sigma_R)}$ as we saw before, which is equal to $(-1)^{\text{Arf}(\hat{\Sigma})}$ when $\Sigma$ is oriented.

We note that the RR spectra are equivalent to that of Type IIB, and the theory has a gravitational anomaly from the anti-self-dual 4-form, with no fermions to cancel it. As we discuss briefly in Appendix C, consistency requires the theory to be coupled to fermionic open strings, giving a $U(32)$ gauge group [56,60].

## 3.3 Branes and K-theory

In the above analysis we identified two oriented Type II strings, two oriented Type 0 strings, and a number of unoriented Type 0 strings. In this subsection we discuss their spectra of stable branes. To do so, we begin by briefly reviewing the well-known K-theory classification of stable branes for oriented theories.

---

[7]We do this for notational convenience only. There is no correspondence between quadratic enhancements and $\text{Pin}^+$ structures in general.

Recall that oriented Type IIB on a spacetime $X$ has stable D-branes which are classified by the K-group $K(X)$ [33,34]. This is the group of pairs of vector bundles $(E,F)$ over $X$ subject to an equivalence relation $(E \oplus H, F \oplus H) \sim (E,F)$. More precisely, one should consider the reduced K-group $\widetilde{K}(X)$, for which the bundles $E$ and $F$ are required to have the same rank. Physically, the idea is to begin with a stack of equal numbers of D9- and $\overline{\text{D9}}$-branes, and then to consider annihilation amongst these stacks. When the vector bundles over these stacks are unequal this annihilation is not complete, and a residual brane of lower dimension is left over [32,33,67]. It is expected that all branes can be obtained in this way.

For Type IIA, stable branes are classified by the higher K-group $K^1(X) = \widetilde{K}(X \times S^1)$. One might entertain the possibility of allowing for even higher K-groups $\widetilde{K}^n(X)$ for $n > 1$. However, Bott periodicity states that for complex K-groups,

$$\widetilde{K}^n(X) = \widetilde{K}^{n+2}(X) \,. \tag{3.24}$$

Thus the only distinct complex K-groups are those mentioned above, and both are realized by string theories. The stable D$p$-branes are captured by the groups $\widetilde{K}^n(S^{9-p})$, as listed in the first two rows of Table 3.

As discussed in Section 3.1.1, the spectrum of massless RR fields in oriented Type 0A/B theories is precisely double that of Type IIA/B. As such, one expects the spectrum of branes in Type 0A/B to be doubled as well; the two branes of given worldvolume dimension $(p+1)$ are typically denoted as D$p$- and D$p'$-branes. It follows that the classification of stable branes is via two copies of the complex K-groups just described. In other words, because there now exist both D9- and D9$'$-branes, we must consider two separate pairs of vector bundles corresponding to D9-$\overline{\text{D9}}$ and D9$'$-$\overline{\text{D9}'}$ stacks. So the branes in the Type 0 theories are classified by

$$\widetilde{K}^n(X) \oplus \widetilde{K}^n(X) \cong \widetilde{K}^n(X) \oplus \widetilde{K}^{-n}(X) \,. \tag{3.25}$$

The equality above follows from the mod 2 periodicity of complex K-theory. As we now discuss, it is the latter form which generalizes to the unoriented case.

It has long been known that stable branes in unoriented Type I string theory are classified by real K-theory $\widetilde{KO}(X)$ [33,68]. Crucially, the reduced real K-groups have a mod 8 periodicity [69],

$$\widetilde{KO}^n(X) \cong \widetilde{KO}^{n+8}(X) \,. \tag{3.26}$$

It is thus natural to guess that the eight Pin$^-$ Type 0 strings labeled by $n$ mod 8 have stable branes captured by $\widetilde{KO}^n(X)$. More precisely, because one again expects a doubled spectrum from these Type 0 theories, the relevant group will be found to be

$$\widetilde{KO}^n(X) \oplus \widetilde{KO}^{-n}(X) \,. \tag{3.27}$$

In Sections 4 and 5, this group will be confirmed to classify the stable $p$-brane spectrum of the Pin$^-$ Type 0 theory with $n$ copies of ABK. Concretely, this spectrum is obtained by evaluating $KO^{\pm n}(X)$ on $X = S^{9-p}$, with the results listed in Table 3. The entries in this table can be obtained by noting that $\widetilde{KO}^n(S^k) = KO^{n-k}(pt)$, and then using the following values for real K-groups of points:

$$\begin{aligned} KO^0(pt) &= \mathbb{Z}, \quad KO^{-1}(pt) = \mathbb{Z}_2, \quad KO^{-2}(pt) = \mathbb{Z}_2, \quad KO^{-3}(pt) = 0, \\ KO^{-4}(pt) &= \mathbb{Z}, \quad KO^{-5}(pt) = 0, \quad KO^{-6}(pt) = 0, \quad KO^{-7}(pt) = 0. \end{aligned} \tag{3.28}$$

At this point we can check that the RR spectra we determined above agree with the non-torsion part of $\widetilde{KO}^n(S^{9-p}) \oplus \widetilde{KO}^{-n}(S^{9-p})$. The aim of the next two sections is to establish the agreement including the torsion parts.

For Pin$^+$ Type 0 strings, we saw in Section 3.2.3 that these theories have the same RR spectra as oriented Type IIB. As such, we expect to have the same classification via complex K-theory as in that case. Since the Pin$^+$ strings have less features not found previously than their Pin$^-$ counterparts, we will be very brief about them in what follows.

It is worth noting that whenever tadpole cancellation requires the addition of D9-branes, the question of stability of D$p$-branes must be revisited to account for the possibility of tachyonic modes of the strings stretched between the D$p$- and D9-branes. In this case, the K-theory classification outlined above may be modified, though we will not address these modifications.

Table 3: The ten K-groups capturing stable branes in the oriented Type II and (un)oriented Type 0 theories discussed above.

| | $-1$ | $0$ | $1$ | $2$ | $3$ | $4$ | $5$ | $6$ | $7$ | $8$ | $9$ |
|---|---|---|---|---|---|---|---|---|---|---|---|
| $\widetilde{K}$ | $\mathbb{Z}$ | $0$ | $\mathbb{Z}$ | $0$ | $\mathbb{Z}$ | $0$ | $\mathbb{Z}$ | $0$ | $\mathbb{Z}$ | $0$ | $\mathbb{Z}$ |
| $\widetilde{K}^1$ | $0$ | $\mathbb{Z}$ | $0$ | $\mathbb{Z}$ | $0$ | $\mathbb{Z}$ | $0$ | $\mathbb{Z}$ | $0$ | $\mathbb{Z}$ | $0$ |
| $\widetilde{KO}^0 \oplus \widetilde{KO}^{-0}$ | $2\mathbb{Z}_2$ | $2\mathbb{Z}_2$ | $2\mathbb{Z}$ | $0$ | $0$ | $0$ | $2\mathbb{Z}$ | $0$ | $2\mathbb{Z}_2$ | $2\mathbb{Z}_2$ | $2\mathbb{Z}$ |
| $\widetilde{KO}^1 \oplus \widetilde{KO}^{-1}$ | $\mathbb{Z}_2$ | $\mathbb{Z} \oplus \mathbb{Z}_2$ | $\mathbb{Z}_2$ | $\mathbb{Z}$ | $0$ | $\mathbb{Z}$ | $0$ | $\mathbb{Z} \oplus \mathbb{Z}_2$ | $\mathbb{Z}_2$ | $\mathbb{Z} \oplus \mathbb{Z}_2$ | $\mathbb{Z}_2$ |
| $\widetilde{KO}^2 \oplus \widetilde{KO}^{-2}$ | $2\mathbb{Z}$ | $0$ | $\mathbb{Z}_2$ | $\mathbb{Z}_2$ | $2\mathbb{Z}$ | $0$ | $\mathbb{Z}_2$ | $\mathbb{Z}_2$ | $2\mathbb{Z}$ | $0$ | $\mathbb{Z}_2$ |
| $\widetilde{KO}^3 \oplus \widetilde{KO}^{-3}$ | $0$ | $\mathbb{Z}$ | $0$ | $\mathbb{Z} \oplus \mathbb{Z}_2$ | $\mathbb{Z}_2$ | $\mathbb{Z} \oplus \mathbb{Z}_2$ | $\mathbb{Z}_2$ | $\mathbb{Z}$ | $0$ | $\mathbb{Z}$ | $0$ |
| $\widetilde{KO}^4 \oplus \widetilde{KO}^{-4}$ | $0$ | $0$ | $2\mathbb{Z}$ | $0$ | $2\mathbb{Z}_2$ | $2\mathbb{Z}_2$ | $2\mathbb{Z}$ | $0$ | $0$ | $0$ | $2\mathbb{Z}$ |
| $\widetilde{KO}^5 \oplus \widetilde{KO}^{-5}$ | $0$ | $\mathbb{Z}$ | $0$ | $\mathbb{Z} \oplus \mathbb{Z}_2$ | $\mathbb{Z}_2$ | $\mathbb{Z} \oplus \mathbb{Z}_2$ | $\mathbb{Z}_2$ | $\mathbb{Z}$ | $0$ | $\mathbb{Z}$ | $0$ |
| $\widetilde{KO}^6 \oplus \widetilde{KO}^{-6}$ | $2\mathbb{Z}$ | $0$ | $\mathbb{Z}_2$ | $\mathbb{Z}_2$ | $2\mathbb{Z}$ | $0$ | $\mathbb{Z}_2$ | $\mathbb{Z}_2$ | $2\mathbb{Z}$ | $0$ | $\mathbb{Z}_2$ |
| $\widetilde{KO}^7 \oplus \widetilde{KO}^{-7}$ | $\mathbb{Z}_2$ | $\mathbb{Z} \oplus \mathbb{Z}_2$ | $\mathbb{Z}_2$ | $\mathbb{Z}$ | $0$ | $\mathbb{Z}$ | $0$ | $\mathbb{Z} \oplus \mathbb{Z}_2$ | $\mathbb{Z}_2$ | $\mathbb{Z} \oplus \mathbb{Z}_2$ | $\mathbb{Z}_2$ |

# 4  D-brane spectra via boundary fermions

In this section, we demonstrate the $KO^n \oplus KO^{-n}$ classification of stable branes for the Pin$^-$ Type 0 theory with $n$ copies of ABK. This is done by analyzing the Clifford modules carried by open string endpoints. After doing so, we also study the spectra of non-stable branes in these theories, including the gauge groups supported on their worldvolumes and the representations of their open string tachyons.

## 4.1  (0+1)d Majorana fermions and their anomalies

### 4.1.1  Fermions and Clifford algebras

We begin by considering systems of (0+1)d Majorana fermions, which appear on the boundary of $n$ copies of the ABK theory. Let us say we have $r + s$ hermitian fermion operators $\xi_a = (\xi_a)^\dagger$, $a = 1, \ldots, r + s$, satisfying

$$\xi_a^2 = +1 \qquad a = 1, \ldots, r + s \tag{4.1}$$

and

$$\mathsf{T}\xi_a \mathsf{T}^{-1} = \begin{cases} +\xi_a & a = 1, \ldots, r, \\ -\xi_a & a = r+1, \ldots, r+s. \end{cases} \tag{4.2}$$

In the mathematics literature it is more common to consider operators invariant under $\mathsf{T}$. This can be achieved by defining

$$
\begin{cases}
\gamma_a = \xi_a & i = 1, \ldots, r, \\
\gamma_a = i\xi_a & i = r+1, \ldots, r+s.
\end{cases}
\tag{4.3}
$$

The $\gamma_a$ are no longer hermitian in general, but satisfy $\mathsf{T}\gamma_a\mathsf{T}^{-1} = \gamma_a$. They generate the real Clifford algebra $\mathrm{Cl}(r,s)$. We often use the abbreviations $\mathrm{Cl}(+n) := \mathrm{Cl}(n,0)$ and $\mathrm{Cl}(-n) := \mathrm{Cl}(0,n)$.[8]

The system of $r+s$ fermions giving rise to $\mathrm{Cl}(r,s)$ can have anomalies in the realization of $\mathsf{T}$ and $(-1)^{\mathsf{f}}$, which will be the topic of the next subsection. Before proceeding, we now give a rough argument for why these anomalies depend only on $r-s$ modulo 8. First we argue that only $r-s$ is relevant for the anomaly. The reason is that a pair of fermions with opposite $\mathsf{T}$ transformations allow a $\mathsf{T}$-invariant mass term, so they cannot be anomalous. For example, when $(r,s) = (1,1)$, we can simply add a $\mathsf{T}$-invariant mass term $i\xi_1\xi_2$, which would trivialize the vacuum, precluding any anomaly.

We next argue that only $r-s \mod 8$ matters. This can be understood in two steps. As the first step, we consider $\mathrm{Cl}(4)$. For this we can introduce a $\mathsf{T}$-invariant quartic hermitian interaction term

$$
H = \xi_1\xi_2\xi_3\xi_4
\tag{4.4}
$$

to the system, for which the vacuum is two-dimensional, purely bosonic, and $\mathsf{T}^2 = -1$. This realizes a Kramers doublet, on which

$$
\sigma_x := i\xi_1\xi_2, \qquad \sigma_y := i\xi_1\xi_3, \qquad \sigma_z := i\xi_1\xi_4
\tag{4.5}
$$

act as Pauli matrices.

As the second step, we combine two such Kramers doublets obtained from two copies of $\mathrm{Cl}(4)$, and make a single bosonic system with $\mathsf{T}^2 = +1$ and a unique vacuum. This can be done by introducing a $\mathsf{T}$-invariant term

$$
H' := \sigma_x \otimes \sigma_x + \sigma_y \otimes \sigma_y + \sigma_z \otimes \sigma_z,
\tag{4.6}
$$

which we note can be realized as a four-fermi operator using (4.5).

In other words, we can introduce to $\mathrm{Cl}(8)$ a four-fermion term $c_{ijk\ell}\xi^i\xi^j\xi^k\xi^\ell$ which is hermitian and $\mathsf{T}$-invariant, such that its addition leads to a non-degenerate vacuum [35]. Therefore, eight Majorana fermions with the same time-reversal properties can be removed without affecting the anomaly.

### 4.1.2 More general (0+1)d systems and anomalies

We do not necessarily have to couple $n$ boundary Majorana fermions to $n$ copies of the ABK theory. We only have to couple a boundary system which has the *same* anomaly as $n$ Majorana fermions. We thus need to understand the possible anomalies concerning the realizations of $\mathsf{T}$ and $(-1)^{\mathsf{f}}$. In the mathematical literature this analysis was first done in [71], in which the following eight-fold classification in terms of three signs was given.

The first sign is the most subtle to define. We ask whether $(-1)^{\mathsf{f}}$ can be realized in an irreducible ungraded representation of the algebra. If this is possible, the representation is of type $+$, and if not, it is of type $-$. As an example, consider $\mathrm{Cl}(+1)$. There are two ungraded irreducible representations, which are real one-dimensional such that $\xi_1 = \pm 1$. Clearly there is no $(-1)^{\mathsf{f}}$ operator that anticommutes with $\xi_1$, meaning that $\mathrm{Cl}(+1)$ is of type $-$.

---

[8]Our convention is that $\mathrm{Cl}(-n) = C_n$ and $\mathrm{Cl}(+n) = C'_n$ in the notation of Atiyah-Bott-Shapiro [70].

Table 4: The properties of $\mathsf{T}$ and $(-1)^{\mathrm{f}}$ for the eight anomaly types $n = 0, \dots, 7$. The corresponding graded division algebras $\mathcal{A} = \mathcal{A}_0 \oplus \mathcal{A}_1$, $\mathcal{A}_1 = \mathcal{A}_1^+ \oplus \mathcal{A}_1^-$ are also given.

| $n$ | type | $\mathsf{T}$ and $(-1)^{\mathrm{f}}$ | $\mathsf{T}^2$ |
|---|---|---|---|
| 0 | $+$ | commute | $\mathbb{R}$ |
| 1 | $-$ | anticommute | $\mathbb{R}$ |
| 2 | $+$ | anticommute | $\mathbb{R}$ |
| 3 | $-$ | commute | $\mathbb{H}$ |
| 4 | $+$ | commute | $\mathbb{H}$ |
| 5 | $-$ | anticommute | $\mathbb{H}$ |
| 6 | $+$ | anticommute | $\mathbb{H}$ |
| 7 | $-$ | commute | $\mathbb{R}$ |

| $n$ | $\mathcal{A}$ | | $\mathcal{A}_0$ | $\mathcal{A}_1^+$ | $\mathcal{A}_1^-$ | $\sqrt{\dim_{\mathbb{R}}}$ |
|---|---|---|---|---|---|---|
| 0 | $\mathrm{Cl}(+0) =$ | $\mathbb{R}$ | $\mathbb{R}$ | | | 1 |
| 1 | $\mathrm{Cl}(+1) =$ | $\mathbb{R} \oplus \mathbb{R}$ | $\mathbb{R}$ | $\mathbb{R}$ | | $\sqrt{2}$ |
| 2 | $\mathrm{Cl}(+2) =$ | $\mathbb{R}[2]$ | $\mathbb{C}$ | $\mathbb{C}$ | | 2 |
| 3 | $\mathrm{Cl}(+3) =$ | $\mathbb{C}[2]$ | $\mathbb{H}$ | $\mathbb{R}^3$ | $\mathbb{R}^1$ | $2\sqrt{2}$ |
| 4 | | $\mathbb{H}$ | $\mathbb{H}$ | | | 2 |
| 5 | $\mathrm{Cl}(-3) =$ | $\mathbb{H} \oplus \mathbb{H}$ | $\mathbb{H}$ | $\mathbb{R}^1$ | $\mathbb{R}^3$ | $2\sqrt{2}$ |
| 6 | $\mathrm{Cl}(-2) =$ | $\mathbb{H}$ | $\mathbb{C}$ | | $\mathbb{C}$ | 2 |
| 7 | $\mathrm{Cl}(-1) =$ | $\mathbb{C}$ | $\mathbb{R}$ | | $\mathbb{R}$ | $\sqrt{2}$ |

In the following, for representations of type $-$, we adjoin $(-1)^{\mathrm{f}}$ to the algebra and consider the resulting irreducible graded representations. Again take $\mathrm{Cl}(+1)$ as an example. Then we consider a representation given by

$$\xi_1 = \begin{pmatrix} 0 & 1 \\ 1 & 0 \end{pmatrix}, \qquad (-1)^{\mathrm{f}} = \begin{pmatrix} 1 & 0 \\ 0 & -1 \end{pmatrix}. \tag{4.7}$$

This is not irreducible as an ungraded representation but is irreducible as a graded representation.

The other two signs specifying the anomaly type are easier to define. The second sign is the one appearing in

$$\mathsf{T}(-1)^{\mathrm{f}} = \pm(-1)^{\mathrm{f}}\mathsf{T}, \tag{4.8}$$

while the third sign is the one appearing in

$$(\mathsf{T})^2 = \pm 1. \tag{4.9}$$

The eight types, labeled by $n = 0, \dots, 7$, are summarized in the left portion of Table 4. There, we showed $\mathsf{T}^2 = \pm 1$ in terms of the corresponding division algebras $\mathbb{R}$ and $\mathbb{H}$. In [71] it was shown that the tensor product of a representation of type $n$ and another of type $n'$ has the type $n + n'$ modulo 8. A very explicit analysis of $\mathrm{Cl}(+n)$ was given in Sec. 2.5.1 and 2.5.2 of [72], from which one can find that $\mathrm{Cl}(+n)$ is indeed of type $n$. Similarly, $\mathrm{Cl}(-n)$ is of type $-n$.

In [71] it was also shown that any graded irreducible representation of type $n$ automatically contains an action of the minimal algebra $\mathcal{A}$ for that type. This information is shown in the right portion of Table 4. There, $\mathbb{F}[n]$ stands for the $n \times n$ matrix algebra over the field $\mathbb{F}$, $\mathcal{A}_0$ and $\mathcal{A}_1$ are the bosonic and fermionic parts of the algebra, and $\mathcal{A}_1^\pm$ are the subspaces of $\mathcal{A}_1$ which are hermitian and anti-hermitian, respectively.[9] These eight graded algebras are known to exhaust the graded division algebras over $\mathbb{R}$, i.e. graded algebras such that any homogeneous element has an inverse.

### 4.1.3 On the boundary Hilbert space

Consider $n$ copies of the Kitaev chain on a segment $\sigma \in [0, \pi]$. We call $\sigma = 0$ the left boundary and $\sigma = \pi$ the right boundary. The Majoranas $\xi$ on the left and $\xi'$ on the right must have opposite transformation properties under time-reversal [35,45]. This can be seen by imagining

---

[9]Note that the hermitian conjugate on a real algebra is simply an involution satisfying $(ab)^* = b^* a^*$.

a process in which the endpoints of the segment join to give a closed circle. Once the endpoints come together, the T-invariant mass term involving $\xi$ and $\xi'$ should be able to gap the system, so $\xi$ and $\xi'$ needs to have opposite T-transformation properties.

We work in the convention[10] that we have $n$ fermions $\xi_i$ with $T\xi_i T^{-1} = +\xi_i$ on the left and $n$ fermions $\xi'_i$ with $T\xi'_i T^{-1} = -\xi'_i$ on the right. They form $\mathrm{Cl}(n)$ and $\mathrm{Cl}(-n)$, respectively. The Hilbert space associated to the open string segment, including the bulk and two boundaries, can be identified with $\mathrm{Cl}(n)$ itself, on which $\mathrm{Cl}(n)$ acts from the left and $\mathrm{Cl}(-n)$ acts from the right. That the Hilbert space on the segment should naturally be equal to the boundary algebra $\mathrm{Cl}(n)$ itself is clear from the state-operator correspondence. If we have other degrees of freedom on the worldsheet, the Hilbert space on the segment is of the form

$$\mathcal{H}_{\mathrm{phys}} = \mathrm{Cl}(n) \otimes \mathcal{H}_{\mathrm{other\ dof}}. \tag{4.10}$$

Naively, one would like to say that this Hilbert space is the tensor product of the Hilbert spaces on the two boundaries. The square root $\sqrt{\dim_{\mathbb{R}} \mathcal{A}}$, listed in Table 4, is then what would be taken as the dimension of the boundary Hilbert spaces. Note that this is not always an integer. It is sometimes useful to have a well-defined, non-anomalous boundary Hilbert space with integer dimension. This can be done by introducing $n$ auxiliary boundary fermions $\Xi_i$ with $T\Xi_i T^{-1} = -\Xi_i$ on the left and $n$ auxiliary boundary fermions $\Xi'_i$ with $T\Xi'_i T^{-1} = +\Xi'_i$ on the right; this technique was used in e.g. [73] in the Type II setting. We note that auxiliary boundary fermions have opposite T-transformation rules as compared to the physical boundary fermions. The fermions on the left boundary now form $\mathrm{Cl}(n,n)$ and can be quantized without anomaly. This can be represented on a space $V$, thus providing Chan-Paton indices to the boundary. We can do the same on the right. The Hilbert space on the segment, including both the physical and auxiliary boundary fermions, is then of the form

$$V \otimes V^* \otimes \mathcal{H}_{\mathrm{other\ dof}} = \mathcal{H}_{\mathrm{aux}} \otimes \mathcal{H}_{\mathrm{phys}}, \tag{4.11}$$

where $\mathcal{H}_{\mathrm{aux}} = \mathrm{Cl}(-n)$ and $\mathcal{H}_{\mathrm{phys}}$ was defined in (4.10). Note that the elements of $\mathcal{H}_{\mathrm{phys}}$ can be found by finding operators (anti)commuting with all the auxiliary boundary fermions $\Xi$.

Below, when we say that "a boundary carries a representation of $\mathrm{Cl}(n)$," we mean that there are physical boundary fermions forming $\mathrm{Cl}(n)$, where $n$ is taken modulo 8. If we use the auxiliary boundary fermions, they form $\mathrm{Cl}(-n)$.

## 4.2 D-branes and boundary fermions

Let us now study the boundary fermions in the context of the worldsheet theory of Pin$^-$ Type 0 strings, with $n$ copies of the ABK theory. We first consider open strings ending on 9-branes. We then have $n$ boundary fermions on the left forming $\mathrm{Cl}(n)$ and $n$ boundary fermions on the right forming $\mathrm{Cl}(-n)$, as discussed above. The open-string Hilbert space, before GSO projection, is of the form

$$\mathrm{Cl}(n) \otimes \mathcal{H}_0, \tag{4.12}$$

where $\mathcal{H}_0$ is the open string Hilbert space of the massless worldsheet fields. Also as discussed above, we can replace $\mathrm{Cl}(n)$ with any graded algebra having the same anomaly.

The restriction to 9-branes above meant that the only boundary fermions required were those needed to cancel the anomaly of $n$ copies of ABK. However, for branes of higher codimension there will be additional anomalies from bulk fermion zero-modes, which should be accompanied by additional boundary fermions $\zeta_i$. Let us set $n = 0 \mod 8$ for the moment, since the ABK boundary fermions can be easily reinstated later. In order to understand the additional boundary Majoranas $\zeta_i$, we must first understand when zero-modes can appear in

---

[10]In Section 5, the choice of the convention here will correspond to a choice of definition of the O9-plane state.

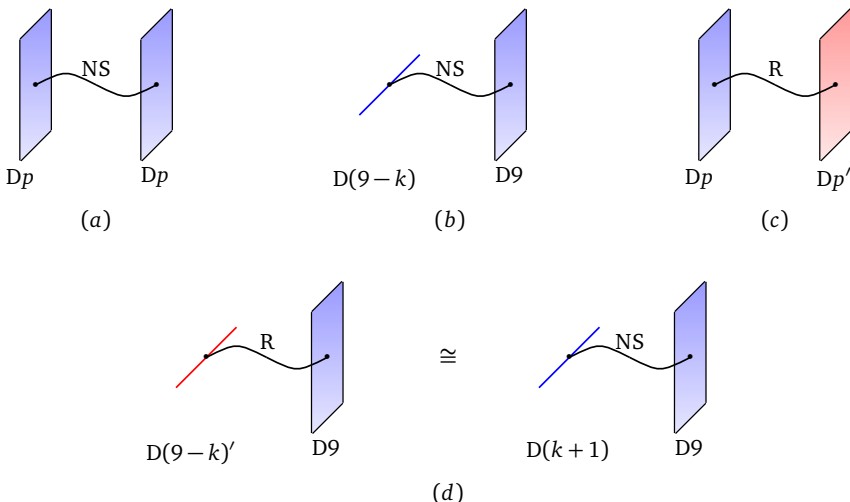

Figure 3: ($a$) A pair of D$p$-branes with an NS sector string stretched between them; ($b$) an NS sector string stretched between a D($9-k$)- and D9-brane; ($c$) an R sector string stretched between a D$p$- and D$p'$-brane; ($d$) an R sector string stretched between a D($9-k$)'- and D9-brane, which can be thought of as an NS sector string stretched between a D($k+1$)- and D9-brane, as far as the behavior of the fermion zero modes is concerned.

the bulk of the string. As reviewed in Appendix A, this depends on whether the ends of the string have Dirichlet (D) or Neumann (N) boundary conditions. The NN, DD, and ND strings satisfy the following boundary conditions,

$$
\begin{aligned}
\text{NN}: &\qquad \psi^\mu(t,0) = -\eta_1 \tilde{\psi}^\mu(t,0), &\qquad \psi^\mu(t,\pi) = +\eta_2 \tilde{\psi}^\mu(t,\pi), \\
\text{DD}: &\qquad \psi^\mu(t,0) = +\eta_1 \tilde{\psi}^\mu(t,0), &\qquad \psi^\mu(t,\pi) = -\eta_2 \tilde{\psi}^\mu(t,\pi), \\
\text{ND}: &\qquad \psi^\mu(t,0) = -\eta_1 \tilde{\psi}^\mu(t,0), &\qquad \psi^\mu(t,\pi) = -\eta_2 \tilde{\psi}^\mu(t,\pi), \qquad (4.13)
\end{aligned}
$$

where $\psi, \tilde{\psi}$ are left- and right-moving bulk fermions and $\eta_{1,2} = \pm 1$ specifies the boundary conditions at $\sigma = 0, \pi$. Though only the relative sign in these conditions is ultimately important, our current conventions are such that $\eta_{1,2} = +1$ represents a D$p$-brane at $\sigma = 0, \pi$, while $\eta_{1,2} = -1$ represents a D$p'$-brane at $\sigma = 0, \pi$. This is described in more detail in Appendices A and B.

The presence of zero-modes depends on the relative choice of $\eta_{1,2}$. This choice is in turn related to the choice of NS or R sectors on the open string; we declare that the string is in the NS sector for $\eta_1 = \eta_2$ and in the R sector for $\eta_1 = -\eta_2$; see Appendix A. From (4.13) we then conclude that NS sector fermions can have zero-modes along ND and DN directions, whereas R sector fermions can have zero-modes along NN and DD directions.

Consider now a coincident pair of 9-branes of type $\eta$ (Fig. 3(a)). In light-cone gauge, there are zero ND directions and a total of 8 NN+DD directions. Since open strings stretching between branes of the same type are in the NS sector, any potential zero-modes would be in ND directions, of which there are none. Hence there are no bulk zero-modes, and thus no boundary fermions. The same holds more generally for coincident pairs of $p$-branes of type $\eta$.

We now consider a 9-brane coincident with a $(9-k)$-brane, both of type $\eta$ (Fig. 3(b)). We consider an open string between them, which is again in the NS sector since the branes are still of the same type. However, there are now $k$ DN directions, and hence $k$ zero-modes. We denote the restriction of these zero modes to the boundary at the $(9-k)$-brane by $\psi_0^i$. As in (3.12), the boundary time-reversal operator $\mathsf{T}$ acts on these by $\mathsf{T}\psi_0^i\mathsf{T}^{-1} = \tilde{\psi}_0^i$. By (4.13),

if the $(9-k)$-brane is at $\sigma = 0$ then we have $\psi_0^i = \eta\tilde{\psi}_0^i$, whereas if it is at $\sigma = \pi$ we have $\psi_0^i = -\eta\tilde{\psi}_0^i$. Together we conclude that

$$\mathsf{T}\psi_0^i\mathsf{T}^{-1} = \begin{cases} +\eta\psi_0^i & \sigma = 0, \\ -\eta\psi_0^i & \sigma = \pi. \end{cases} \tag{4.14}$$

When $k \neq 0 \mod 8$, these zero-modes will lead to anomalies in $\mathsf{T}$ and $(-1)^f$, and in order to cancel them we must add extra boundary fermions. At $\sigma = 0$ there are two options: we may add $k$ boundary fermions $\zeta_i$ satisfying $\mathsf{T}\zeta_i\mathsf{T}^{-1} = -\eta\zeta_i$, or $8-k$ boundary fermions satisfying $\mathsf{T}\zeta_i\mathsf{T}^{-1} = +\eta\zeta_i$. These two choices are effectively identical, as they both behave as $\text{Cl}(-\eta k)$. On the boundary at $\sigma = \pi$, one likewise requires a representation of $\text{Cl}(+\eta k)$.

For full consistency, we must check that the same additional boundary fermions also cancel tentative anomalies in mixed strings stretching between D$p$- and D$q'$-branes. To begin, consider coincident D$p$- and D$p'$-branes (Fig. 3(c)). There are then zero ND directions and a total of 8 NN+DD directions. Since the branes are now of opposite types, the open strings stretched between them are in the R sector. Recall that in the R sector, there are zero-modes not from ND directions, but rather from NN and DD directions. Hence in this case there are a total of 8 bulk zero-modes. But because there are eight of them, there is no time-reversal anomaly, and no boundary fermions are necessary. This is consistent with the fact that the strings stretching from the D$p$-branes to themselves, or from the D$p'$-branes to themselves, were shown to not require any boundary fermions.

Another way to analyze this setup is to note that, insofar as counting zero-modes is concerned, switching $\eta \to -\eta$ is equivalent to switching Neumann and Dirichlet boundary conditions in all directions, as is evident from (4.13). Then the D$p'$-brane can be replaced by a D$(10-p)$-brane, and we now have a setup with 8 ND directions and zero NN and DD directions (in light-cone gauge). Since the branes are of the same type now, the open string stretched between them is in the NS sector, and we can again conclude that we have 8 zero-modes. This method of replacing the D$p'$-brane with a D$(10-p)$-brane is useful in analyzing e.g. the case of a D$(9-k)'$- and D9-brane, which can be replaced by a D$(k+1)$- and D9-brane (Fig. 3(d)). By our above analysis, this then has $8-k$ bulk zero-modes which satisfy $\mathsf{T}\psi_0\mathsf{T}^{-1} = -(-1)\psi_0 = +\psi_0$ at $\sigma = 0$. These can be cancelled by adding $8-k$ boundary fermions $\zeta_i$ satisfying $\mathsf{T}\zeta_i\mathsf{T}^{-1} = -\zeta_i$, or $k$ boundary fermions satisfying $\mathsf{T}\zeta_i\mathsf{T}^{-1} = \zeta_i$, both of which give a representation of $\text{Cl}(k)$. This matches the results found before for a D$(9-k)'$-brane. Using these techniques, one can easily check that the more general mixed strings between D$p$- and D$q'$-branes have vanishing anomaly when one makes the above assignments of boundary fermions.

Finally, we may reintroduce non-zero $n$. As discussed above, this gives a representation of $\text{Cl}(n)$ on the boundary at $\sigma = 0$ and of $\text{Cl}(-n)$ on the boundary at $\sigma = \pi$. Let us now consider an open string stretched between a $(9-k)$-brane of type $\eta$ at $\sigma = 0$ and a D$(9-k)$-brane at $\sigma = \pi$. In the theory with $n$ copies of ABK, the endpoint at $\sigma = 0$ carries a representation of $\text{Cl}(n - \eta k)$, while the endpoint at $\sigma = \pi$ carries a representation of $\text{Cl}(-n + \eta k)$.

## 4.3 K-theory classification of branes

We can now study the stability of D-branes in these theories. The analogous results in Type I were obtained in [73–75]. We first note that the tachyon vertex operator $\zeta$ is a fermionic, hermitian boundary operator such that the boundary interaction

$$\int dt\, i\psi_0^\mu \zeta D_\mu \mathcal{T}(X) \tag{4.15}$$

is T-invariant. Here $\psi_0'^{\mu}$ is the restriction of the bulk fermion field to the boundary and $\mathcal{T}(X)$ is the spacetime tachyon profile. In particular, the $\psi_0^{\mu}$ appearing here should be those associated to the Neumann directions, which have opposite T transformations as those given in (4.14). Thus for the coupling to be T-invariant, we need $\zeta$ to have the *same* T transformations as those given in (4.14),

$$\mathsf{T}\zeta\mathsf{T}^{-1} = \begin{cases} +\eta\zeta & \sigma = 0, \\ -\eta\zeta & \sigma = \pi. \end{cases} \tag{4.16}$$

Stable branes are those for which such $\zeta$ does not exist.

It is cumbersome to carry out the analysis below for the two cases $\eta = \pm 1$ separately. This can be circumvented by analyzing the $\sigma = 0$ end when $\eta = +1$ and the $\sigma = \pi$ end when $\eta = -1$. This means that we work with the Clifford algebra $\mathrm{Cl}(n - \eta k)$ for $\eta = +1$ and $\mathrm{Cl}(-n + \eta k)$ for $\eta = -1$, or in other words just $\mathrm{Cl}(\eta n - k)$ for both cases. Therefore, below we simply analyze $\mathrm{Cl}(\eta n - k)$ and look for a tachyon vertex operator satisfying

$$\mathsf{T}\zeta\mathsf{T}^{-1} = \zeta, \tag{4.17}$$

for both cases $\eta = \pm 1$. This means that the results to be obtained depend only on

$$\nu := \eta n - k. \tag{4.18}$$

When $\nu = 0 \mod 8$, the minimal choice of Chan-Paton algebra is simply $\mathbb{R}$, as can be seen from Table 4. We can enlarge the Chan-Paton algebra by introducing $N$ bosonic indices and $N'$ fermionic indices. Then the Chan-Paton algebra has the bosonic part $\mathbb{R}[N] \oplus \mathbb{R}[N']$ and the fermionic part $\mathbb{R}^N \otimes \mathbb{R}^{N'} \oplus \mathbb{R}^{N'} \otimes \mathbb{R}^N$, where we recall that the algebra $\mathbb{F}[N]$ stands for the $N \times N$ matrix algebra with entries in $\mathbb{F}$. The T-invariant fermionic part is then $\mathbb{R}^N \otimes \mathbb{R}^{N'}$. This means that the gauge group is $O(N) \times O(N')$ and the tachyon field is in the bifundamental. The stable branes are those such that the tachyon representation is empty, which occurs for $N \geq 0$ and $N' = 0$ or $N = 0$ and $N' \geq 0$. These can be labeled by $\mathbb{Z}$. The case $\nu = 4$ is similar; one simply replaces $\mathbb{R}$ by $\mathbb{H}$ and $O$ by $Sp$.

In the other six cases, the minimal choice of Chan-Paton algebra $\mathcal{A}$ already contains both the bosonic and the fermionic part. We can introduce additional Chan-Paton indices $i = 1, \dots, N$, thus enlarging the Chan-Paton algebra to $\mathcal{A}[N] := \mathcal{A} \otimes \mathbb{R}[N]$. From Table 4, we see that $\mathcal{A}[N]$ has the following structure:

$$
\begin{array}{c||c|c|c}
\nu & \mathcal{A}[N]_0 & \mathcal{A}[N]_1^+ & \mathcal{A}[N]_1^- \\
\hline
1 & \mathbb{R}[N] & \mathbb{R}[N]^{\mathrm{symm.}} & \mathbb{R}[N]^{\mathrm{anti.}} \\
2 & \mathbb{C}[N] & \mathbb{C}[N]^{\mathrm{symm.}} & \mathbb{C}[N]^{\mathrm{anti.}} \\
3 & \mathbb{H}[N] & \mathbb{H}[N]^{\mathrm{symm.}} & \mathbb{H}[N]^{\mathrm{anti.}} \\
5 & \mathbb{H}[N] & \mathbb{H}[N]^{\mathrm{anti.}} & \mathbb{H}[N]^{\mathrm{symm.}} \\
6 & \mathbb{C}[N] & \mathbb{C}[N]^{\mathrm{anti.}} & \mathbb{C}[N]^{\mathrm{symm.}} \\
7 & \mathbb{R}[N] & \mathbb{R}[N]^{\mathrm{anti.}} & \mathbb{R}[N]^{\mathrm{symm.}}
\end{array}
\tag{4.19}
$$

where $\mathcal{A}[N]_{0,1}$ denotes the bosonic and fermionic parts and $\mathcal{A}[N]_1^{\pm}$ denotes the hermitian and anti-hermitian parts. The symmetrization and antisymmetrization are defined as usual for $\mathbb{R}$ and $\mathbb{C}$, and as the symmetric and antisymmetric tensor square for the fundamental representation of $Sp(N)$ for $\mathbb{H}$. It is instructive to check that the table above reduces to Table 4 when $N = 1$. From this it is easy to read off the gauge group and the representation of the tachyons for the D-branes, which is given in Table 5. One may check that by setting $n = 0$ in Table 5 we get back the results of [73–75] for Type I.

The stable branes are those for which the tachyons are absent. For $\nu = 0, 4$ we already discussed that they are classified by $\mathbb{Z}$. For $\nu = 1, 2, 3, 5$ the tachyon field is always present.

Table 5: Gauge groups $G_\nu$ and tachyon representations $\rho_\zeta$ on the worldvolume of $N$ $(9-k)$-branes of type $\eta$ in the theory with $n$ copies of ABK. Note that $\nu = \eta n - k = \eta n + p - 1 \mod 8$. "Bifund." refers to the bifundamental representation, whereas "symm." ("anti.") refer to the symmetric (anti-symmetric) rank 2 tensor representations. We also listed the tensions of these branes, as well as their K-theory classifications, which can be found by considering when the tachyon field is empty.

| $\nu$ | 0 | 1 | 2 | 3 | 4 | 5 | 6 | 7 |
|---|---|---|---|---|---|---|---|---|
| $G_\nu$ | $O(N) \times O(N')$ | $O(N)$ | $U(N)$ | $Sp\left(\frac{N}{2}\right)$ | $Sp\left(\frac{N}{2}\right) \times Sp\left(\frac{N'}{2}\right)$ | $Sp\left(\frac{N}{2}\right)$ | $U(N)$ | $O(N)$ |
| $\rho_\zeta$ | bifund. | symm. | symm. | symm. | bifund. | anti. | anti. | anti. |
| $\lambda_\nu$ | 1 | $\sqrt{2}$ | 2 | $2\sqrt{2}$ | 2 | $2\sqrt{2}$ | 2 | $\sqrt{2}$ |
| $KO^\nu(pt)$ | $\mathbb{Z}$ | 0 | 0 | 0 | $\mathbb{Z}$ | 0 | $\mathbb{Z}_2$ | $\mathbb{Z}_2$ |

Finally, for $\nu = 6, 7$ the tachyon field is absent when $N = 1$ but appears when $N = 2$, and hence the classification is by $\mathbb{Z}_2$. These results match with $KO^\nu(pt)$.

Let us reanalyze this setup using the auxiliary boundary fermions discussed in Section 4.1.3. Our physical boundary fermions form an algebra $\mathcal{A}$ whose anomaly is of type $\nu$. We introduce auxiliary boundary fermions $\mathrm{Cl}(-\nu)$ to cancel the anomaly, so $\mathrm{Cl}(-\nu) \otimes \mathcal{A}$ is non-anomalous, acting on the Chan-Paton vector space $V$. The tachyon field corresponds to a hermitian fermionic element $\zeta$ in $\mathcal{A}$. This means that if such a tachyon field is present, the $\mathrm{Cl}(-\nu)$ action on $V$ is extended to a $\mathrm{Cl}(1-\nu)$ action on $V$. In other words, on the D-brane characterized by $\nu$, a Chan-Paton space $V$ with an action of $\mathrm{Cl}(-\nu)$ is necessary, and it is unstable if and only if this $\mathrm{Cl}(-\nu)$ action can be extended to an action of $\mathrm{Cl}(1-\nu)$. Using the result in [70], this can be directly connected to $KO^\nu(pt)$. To see this, recall that in [70] Atiyah, Bott, and Shapiro considered

$$M_{-n} := \text{free } \mathbb{Z}\text{-modules generated by graded irreducible representations of } \mathrm{Cl}(-n) \quad (4.20)$$

and considered the natural map $i^* : M_{-n-1} \to M_{-n}$ induced by $i : \mathrm{Cl}(-n) \to \mathrm{Cl}(-n-1)$. It was then shown that

$$KO^{-n}(pt) = M_{-n}/i^*(M_{-n-1}). \quad (4.21)$$

We note that by replacing $\Xi$ by $\hat{\Xi} = \Xi(-1)^{\mathrm{f}}$, a graded representation of $\mathrm{Cl}(n)$ can be converted to a graded representation of $\mathrm{Cl}(-n)$ and vice versa. Therefore we also have

$$KO^{-n}(pt) = M_n/j^*(M_{n+1}), \quad (4.22)$$

where $j^* : M_{n+1} \to M_n$ is now induced by $j : \mathrm{Cl}(n) \to \mathrm{Cl}(n+1)$. Then, Chan-Paton spaces with $\mathrm{Cl}(-\nu)$ action modulo those with $\mathrm{Cl}(-\nu+1)$ action are clearly classified by $KO^\nu(pt)$.

We note that this restatement also shows that the brane of type $\nu$ can naturally host an orthogonal bundle with an additional action of $\mathrm{Cl}(-\nu)$. It would be interesting to connect this observation to the definition of KO-theories in terms of Clifford bundles, which can be found e.g. in [76].

### 4.3.1 Brane tension from boundary fermions

Finally, let us comment on the tensions of the various branes, both stable and unstable, identified thus far. In order to obtain the tension one computes a disc path integral, with the

boundary of the disc anchored on the brane — this may be interpreted as the one-point function of the brane with a graviton in the closed string picture. If there are Majorana fermions present on the boundary of the disc, these will give a contribution to the path integral. In particular, as mentioned in the Introduction the contribution of a $(0+1)$d Majorana fermion to the path integral is an overall factor of $\sqrt{2}$. This holds in all cases, including Type II and Type 0 strings.

We begin by discussing the more familiar case of Type II, for which there are branes of only a single type $\eta = 1$. As we have discussed, the Type IIB and IIA theories are distinguished by the presence of respectively $n = 0, 1$ Majorana fermions on the boundary. For this reason, the non-BPS D9-branes in Type IIA have a tension which is $\sqrt{2}$ times that of their BPS counterparts in Type IIB. However, it is not the case that *all* branes in Type IIA have tensions which are larger by this factor. Indeed, let us consider branes of codimension $k$. By considering open strings stretched between such branes and D9-branes, we may conclude that strings ending on such branes must have $k$ bulk zero modes — this is argued for as above, by noting that a change in codimension leads to a change in the number of DN or ND directions for the open string endpoints. It is then possible for the anomalies of the invertible phase and bulk zero modes to cancel. In fact, since in this case the anomaly is only $\mathbb{Z}_2$ we can conclude that the number of boundary fermions is only $|n - k| \mod 2$. As such, we conclude that D$p$ branes in Type IIA/B have tensions $(\sqrt{2})^{|n-k| \mod 2} T_p^{\mathrm{II}}$, where $T_p^{\mathrm{II}}$ is the tension of the corresponding stable non-torsion $p$-brane. In particular, this tells us that for Type IIB, branes with $p$ odd have tensions $T_p^{\mathrm{II}}$ (essentially by definition) while for $p$ even the branes have tensions $\sqrt{2} T_p^{\mathrm{II}}$. Likewise for Type IIA, branes with $p$ even have tensions $T_p^{\mathrm{II}}$, while branes with $p$ odd have tensions $\sqrt{2} T_p^{\mathrm{II}}$.

We now generalize this to the case of Type 0 strings. As explained above, in this case the bulk theory carries the same anomaly as $\nu$ boundary Majorana fermions. It is natural to choose the minimal realization of the anomaly in the boundary system, which is a system of Majorana fermions in a representation of Cl($+\nu$) if $\nu < 4$ and of Cl($-\nu$) if $\nu > 4$. Then the tension of the corresponding branes is weighted by the path integral over the system of fermions, yielding the result $\lambda_\nu T_p^0$ where $\lambda_\nu := \sqrt{\dim_{\mathbb{R}} \mathcal{A}}$ as given in Table 4. For example, in the case of a D9- or D9$'$-brane in the theory with 1 or 7 copies of ABK, we find that the tension is $\sqrt{2}$ times that in the theory with no copies of ABK. The values $\lambda_\nu$ are tabulated in Table 5, and will be reproduced for stable branes via the boundary state formalism in Section 5.

### 4.4 Vacuum manifolds of tachyons as classifying spaces

We now reinterpret the results of this section in terms of tachyon condensation on the worldvolume of 9-branes. In the string theory literature, this is referred to as the Atiyah-Bott-Shapiro (ABS) construction [33, 34, 70].

We start from the 9-brane of type $\eta$. As before we use the trick of analyzing the $\sigma = 0$ boundary when $\eta = +1$ and the $\sigma = \pi$ boundary when $\eta = -1$. The boundary then carries an action of Cl($\eta n$). To construct a $(9-k)$-brane located at $X^1 = \cdots = X^k = 0$, we choose a set of time-reversal invariant Majorana fermion operators $\zeta_{i=1,\ldots,k}$ in Cl($\eta n$) (for this purpose one might need to replace $\eta n$ by $\eta n + 8m$ for some integer $m$) and use them to assemble the tachyon field

$$\mathcal{T}(X) = f(|X|) \sum_{i=1}^{k} X^i \zeta_i , \tag{4.23}$$

with $f(|X|)$ some convergence factor chosen such that $\mathcal{T}(X)$ obtains its vacuum value as $|X| \to \infty$ and such that $f(|X|)^2 \sum_{i=1}^{k} (X^i)^2 = 1$. Upon condensation this tachyon gives rise

to the desired $(9-k)$-brane. As we used up $k$ out of $\eta n$ Majorana fermions we originally had on the 9-brane, we end up with the residual action of $Cl(\eta n - k)$, as argued before in a different manner.

We now discuss the connection between tachyon profiles and KO-theory. Consider the usual 9-$\bar{9}$ stack, supporting an $O(N) \times O(N)$ gauge group for $n = 0 \mod 8$. We now follow Sen's construction and consider a domain wall configuration of the tachyon such that it condenses to produce a D8-brane [32]. This involves assignment of a vacuum value of $\zeta(X)$ to each point on the "sphere" $S^0$ at infinity. Thus such domain wall configurations are classified by $\pi_0(\mathcal{V}_9)$, where $\mathcal{V}_9$ is the vacuum manifold for the tachyon on D9-brane. The tachyon potential should be such that it breaks $O(N) \times O(N)$ to the $O(N)$ supported on the resulting D8. A minimal assumption is then

$$\mathcal{V}_9 = \frac{O(\infty) \times O(\infty)}{O(\infty)} \cong O(\infty)\,, \tag{4.24}$$

where we took the formal limit in which the number of original D9-branes is infinite. We can repeat the argument for lower D$p$-branes, and find that they would similarly be classified by $\pi_{8-p}(\mathcal{V}_9)$.

The D$p$-brane can also be obtained from tachyon condensation on D$q$-branes with $p < q < 9$. The vacuum manifold $\mathcal{V}_q$ for the tachyon field on the D$q$-brane can be determined from Table 5:

$$\mathcal{V}_2 = \frac{O(\infty)}{O(\infty) \times O(\infty)} \times \mathbb{Z}\,, \qquad \mathcal{V}_3 = \frac{U(\infty)}{O(\infty)}\,, \quad \mathcal{V}_4 = \frac{Sp(\infty)}{U(\infty)}\,, \quad \mathcal{V}_5 = \frac{Sp(\infty) \times Sp(\infty)}{Sp(\infty)}\,,$$

$$\mathcal{V}_6 = \frac{Sp(\infty)}{Sp(\infty) \times Sp(\infty)} \times \mathbb{Z}\,, \quad \mathcal{V}_7 = \frac{U(\infty)}{Sp(\infty)}\,, \quad \mathcal{V}_8 = \frac{O(\infty)}{U(\infty)}\,, \quad \mathcal{V}_9 = \frac{O(\infty) \times O(\infty)}{O(\infty)}\,, \tag{4.25}$$

where the subscript is defined modulo 8. One then expects that D$p$-branes can be classified by $\pi_{q-p-1}(\mathcal{V}_q)$.

All of this is compatible with the statement that the D$p$-branes are classified by $\widetilde{KO}^0(S^{9-p})$ thanks to the mathematical fact [76] that $KO_n := \mathcal{V}_{n+2}$ is the classifying space of KO-theory, in the sense that

$$\widetilde{KO}^n(X) = [X, KO_n]. \tag{4.26}$$

The $KO_n$ form an $\Omega$-spectrum, which entails the relation $KO_n \simeq \Omega KO_{n+1}$. We then have

$$\pi_{q-p-1}(\mathcal{V}_q) = [S^{q-p-1}, KO_{q-2}] = [pt, \Omega^{q-p-1} KO_{q-2}] = [pt, KO_{p-1}] = \widetilde{KO}^0(S^{9-p})\,. \tag{4.27}$$

# 5 D-brane spectra via boundary states

In the previous section, we were able to verify the K-theory classification of Pin$^-$ Type 0 strings from the open string perspective. In this section, we rederive these results from a closed string perspective. In addition to gaining more intuition about the behavior of invertible phases on worldsheets, we will use the closed string perspective to verify our earlier results on the tensions of stable branes appearing in each theory. These tools will also be used to address the issue of tadpole cancellation in Appendix C.

From the closed string point of view, D-branes correspond to states in the closed string Hilbert space. The basic idea is to define these states by imposing the open string boundary conditions (4.13) as gluing conditions. Upon appropriate rotation of the Euclidean worldsheet, one finds that the correct conditions to impose on boundary states $|Bp, \eta\rangle$ are

$$\begin{aligned} \text{N}: \quad & (\alpha_n^\mu - \tilde{\alpha}_{-n}^\mu)|Bp, \eta\rangle = (\psi_r^\mu - i\eta \tilde{\psi}_{-r}^\mu)|Bp, \eta\rangle = 0\,, \\ \text{D}: \quad & (\alpha_n^\mu + \tilde{\alpha}_{-n}^\mu)|Bp, \eta\rangle = (\psi_r^\mu + i\eta \tilde{\psi}_{-r}^\mu)|Bp, \eta\rangle = 0\,, \end{aligned} \tag{5.1}$$

as reviewed in Appendix B. Here $r$ is integer/half-integer for the R/NS sector. The solution to these equations ends up being the coherent state defined in (B.4). Furthermore, in Appendix B it is shown that the physical D-brane states in Type 0 theories are in fact linear combinations of these $|Bp, \eta\rangle$, of the form

$$|Dp, \eta\rangle = \frac{1}{\mathcal{N}_{Bp}} \frac{1}{\sqrt{2}} (\eta |Bp, \eta\rangle_{\text{NSNS}} + |Bp, \eta\rangle_{\text{RR}}), \qquad \eta = \pm 1, \qquad (5.2)$$

where $|Dp, +\rangle$ represents a D$p$-brane, $|Dp, -\rangle$ represents a D$p'$-brane, and $\mathcal{N}_{Bp} = 2^{\frac{5}{2}} (4\pi^2 \alpha')^{\frac{p-4}{2}}$ is a normalization factor obtained in Appendix B.3. This result will be the starting point for our analysis in this section.

## 5.1 Matching non-torsion brane spectra

In this subsection we begin by rederiving the stable non-torsion brane spectra of the eight Pin$^-$ Type 0 theories via the boundary state formalism. The analysis of the torsion brane spectra will be carried out in Section 5.2.

To understand the spectrum of branes in the theory with $n$ copies of ABK, we need to understand the action of the worldsheet parity operator $\Omega$ on D-brane states in the presence of the invertible phase. We begin by understanding the action of $\Omega$ on the constituent boundary states $|Bp, \eta\rangle$. In Section 3.2.2, we described the action of $\Omega$ on the ground states in the theory with $n$ copies of ABK. From this and (B.9) it easily follows that the action of $\Omega$ on the boundary states is given by

$$\Omega |Bp, \eta\rangle_{\text{NSNS}} = |Bp, \eta\rangle_{\text{NSNS}}, \qquad \Omega |Bp, \eta\rangle_{\text{RR}} = i^{\eta \nu} |Bp, \eta\rangle_{\text{RR}}, \qquad (5.3)$$

with the parameter $\nu = \eta n - k$ defined in Section 4.

The branes which survive the orientifolding are those which are left invariant by $\Omega$; i.e. those for which $\nu$ is zero modulo 4. Therefore, the non-torsion D$p$-branes are in one-to-one correspondence with non-torsion elements of $\widetilde{KO}^n(S^k)$, while the non-torsion D$p'$-branes are in one-to-one correspondence with non-torsion elements of $\widetilde{KO}^{-n}(S^k)$, as listed in Table 3.

Although $\nu$ is a mod 8 parameter, the presence of non-torsion branes depends on $\nu$ only mod 4. Indeed, this is to be expected since (5.3) was obtained by considering only the values of ABK on the Klein bottle $K_2$. However, as discussed in Section 3.2.2, $K_2$ is not the generating manifold for $\mathfrak{V}^2_{\text{Pin}^-}(pt) = \mathbb{Z}_8$. Rather, the generating manifold is $\mathbb{RP}^2$, and since $K_2 \cong \mathbb{RP}^2 \# \mathbb{RP}^2$ we currently have access to only a $\mathbb{Z}_4$ of the full $\mathbb{Z}_8$.

In contrast, the open string states *do* depend on $\nu$ mod 8. This is because the action of $\Omega$ on open string ground states comes from the Möbius strip amplitude. The Möbius strip $M_2$ contains a single $\mathbb{RP}^2$, whose amplitude is $e^{i\pi\text{ABK}(\mathbb{RP}^2)/4} = e^{\pm i\pi/4}$, as we saw in Section 2.2.3. Then we have the schematic action

$$\Omega |0; ij\rangle \sim e^{\pm i n\pi/4} |0; ji\rangle, \qquad (5.4)$$

where $i, j$ are Chan-Paton indices. Therefore shifting $n \to n+4$ changes the action of $\Omega$ on the Chan-Paton factors by a minus sign, between symmetric and anti-symmetric. This is precisely as expected when going between O9$^-$- and O9$^+$-planes, and in agreement with the fact that $\widetilde{KO}^{n+4}(X) \cong \widetilde{KSp}^n(X)$.

## 5.2 Matching torsion brane spectra

We now use the boundary state formalism to check that the torsion brane spectra predicted by $\widetilde{KO}^n(X) \oplus \widetilde{KO}^{-n}(X)$ are reproduced by the theory with $n$ copies of ABK. The study of torsion

branes [32, 77, 78] in the boundary state formalism for unoriented theories was first done in [79–81]. Here we will adapt these methods to the Pin⁻ Type 0 theories.

In contrast to the generic Type 0 non-torsion brane (5.2), the generic torsion branes of Type 0 consists of only the NSNS portion,

$$|\widetilde{Dp}, \eta\rangle = \frac{\lambda_\nu}{\mathcal{N}_{Bp}} \frac{\eta}{\sqrt{2}} |Bp, \eta\rangle_{\text{NSNS}} , \qquad\qquad \eta = \pm 1, \qquad\qquad (5.5)$$

where $\lambda_\nu > 0$ is a normalization factor related to the tension of the torsion brane, which will be shown to depend only on $\nu$. This state, though GSO invariant, is not stable in the oriented Type 0 theory since the tachyon is not projected out. This can be detected by computing the overlap of two boundary states in the closed string tree-channel, and then doing a modular transformation to the open string loop-channel, where a tachyon appears. However, in the unoriented theory the gauging of $\Omega$ can project out the tachyon. This is seen at the level of the amplitude by cancellation of the tachyon piece of the cylinder amplitude with the analogous piece of the Möbius strip amplitude.

To determine the spectrum of stable torsion branes, we begin by computing the closed string cylinder diagram in tree-channel. Using the results of (B.22), we have

$$\mathcal{A}_{C_2} = \int_0^\infty dl \, \langle\widetilde{Dp}, \eta| e^{-2\pi l H_{\text{cl}}} |\widetilde{Dp}, \eta\rangle = \frac{\lambda_\nu^2}{2^6} v_{p+1} \int_0^\infty \frac{d\ell}{\ell^{\frac{9-p}{2}}} \frac{f_3^8(2i\ell)}{f_1^8(2i\ell)} . \qquad (5.6)$$

The functions $f_i(\tau)$ are defined in (B.11) in terms of Jacobi theta functions and the Dedekind eta function. We can translate the tree-channel amplitude to loop-channel using the transformation $l = 1/2t$ and the modular $S$ transformations in (B.12), which yields

$$\mathcal{A}_{C_2} = \frac{1}{2} \lambda_\nu^2 v_{p+1} \int_0^\infty \frac{dt}{(2t)^{\frac{p+3}{2}}} \frac{f_3^8(it)}{f_1^8(it)} , \qquad (5.7)$$

with the answer independent of $\eta$ and $n$. Using the $q$-expansions of the $f_i(\tau)$ given in (B.11), we may isolate the tachyon contribution,

$$\mathcal{A}_{C_2}\big|_{\text{tachyon}} = \frac{1}{2} \lambda_\nu^2 v_{p+1} \int_0^\infty \frac{dt}{(2t)^{\frac{p+3}{2}}} e^{\pi t} . \qquad (5.8)$$

We must now check under which circumstances this can be cancelled by a contribution from the Möbius strip. In order to calculate the Möbius strip amplitude, we use the orientifold state $|Op\rangle$ introduced in (B.42) and calculate in the loop-channel

$$
\begin{aligned}
\mathcal{A}_{M_2} &= \int_0^\infty dl \left( \langle O9| e^{-2\pi\ell H_{\text{cl}}} |\widetilde{Dp}, \eta\rangle + \langle\widetilde{Dp}, \eta| e^{-2\pi\ell H_{\text{cl}}} |O9\rangle \right) \\
&= -\lambda_\nu v_{p+1} \int_0^\infty d\ell \left[ e^{i\frac{\pi n\eta}{4}} \left( \frac{f_3^8 f_4^{2(9-p)}}{f_1^{p-1}} \right) \left( 2i\ell + \frac{1}{2} \right) \right. \\
&\qquad\qquad\qquad\qquad \left. - e^{-i\frac{\pi n\eta}{4}} \left( \frac{f_4^8 f_3^{2(9-p)}}{f_1^{p-1}} \right) \left( 2i\ell + \frac{1}{2} \right) \right] \\
&= -\frac{\lambda_\nu}{2} v_{p+1} \int_0^\infty \frac{dt}{(2t)^{\frac{p+3}{2}}} \left[ e^{i\frac{\pi}{4}(9-p-n\eta)} \left( \frac{f_3^8 f_4^{2(9-p)}}{f_1^{p-1}} \right) \left( it + \frac{1}{2} \right) \right. \\
&\qquad\qquad\qquad\qquad \left. - e^{-i\frac{\pi}{4}(9-p-n\eta)} \left( \frac{f_4^8 f_3^{2(9-p)}}{f_1^{p-1}} \right) \left( it + \frac{1}{2} \right) \right] . (5.9)
\end{aligned}
$$

Isolating the tachyon contribution yields

$$\mathcal{A}_{M_2}\big|_{\text{tachyon}} = \lambda_\nu \sin\left[\frac{\pi}{4}\nu\right] v_{p+1} \int_0^\infty \frac{dt}{(2t)^{\frac{p+3}{2}}} e^{\pi t} \,. \tag{5.10}$$

Combining (5.8) and (5.10), we have

$$(\mathcal{A}_{C_2} + \mathcal{A}_{M_2})\big|_{\text{tachyon}} = \frac{1}{2}\lambda_\nu\left(\lambda_\nu + 2\sin\left[\frac{\pi}{4}\nu\right]\right) v_{p+1} \int_0^\infty \frac{dt}{(2t)^{\frac{p+3}{2}}} e^{\pi t} \,. \tag{5.11}$$

Since $\lambda_\nu$ is positive, the values of $p$ for which the tachyon can be cancelled are those such that $\sin\left[\frac{\pi}{4}\nu\right] < 0$. The tension of the corresponding torsion brane is then $\lambda_\nu T_p^0$, where $\lambda_\nu = -2\sin\left[\frac{\pi}{4}\nu\right]$ and $T_p^0$ is the tension of a stable non-torsion $p$-brane in Type 0. For example, when $n = 0$ we conclude that cancellation is possible if $p \in \{-1, 0, 7, 8\}$, regardless of $\eta = \pm 1$. The corresponding branes have tension $2T_p^0$ for $p = -1, 7$ and tension $\sqrt{2}T_p^0$ for $p = 0, 8$. These are in one-to-one correspondence with the torsion classes in $\widetilde{KO}^n(S^k) \oplus \widetilde{KO}^{-n}(S^k)$. The tensions determined here also match exactly with the ones listed in Table 5.

Note that in this example, it would naively seem that $p = 6$ yields an acceptable torsion brane as well. However, this is not the case. To understand this, recall that before orientifolding these branes are unstable due to a tachyon in the $p$-$p$ strings — these in particular are in the $(-1)^{\text{f}}$ odd part of the spectrum, and are built out of a certain vacuum $|0\rangle_{pp}^{odd}$. One then notes that, without the presence of Chan-Paton factors [80],

$$\Omega|0\rangle_{pp}^{odd} = e^{-i\frac{\pi}{4}p}|0\rangle_{pp}^{odd} \,. \tag{5.12}$$

Hence for $p = 2, 6$ we see that $\Omega^2 = -1$ on the vacuum. To compensate, we need to introduce at least a two-dimensional Chan-Paton index $i = 1, 2$, in the doublet representation of $Sp(1)$. Then the $\Omega$ projection keeps the antisymmetric combination $[ij]$ of the tachyon, and thus there is no stable torsion D6-brane. Indeed this class is absent from $\widetilde{KO}^n(S^k) \oplus \widetilde{KO}^{-n}(S^k)$.

The generalization of this condition for non-zero $n$ is as follows. The goal is to check that $\Omega^2 \neq -1$ on the relevant ground state, lest $\frac{1}{2}(1 + \Omega)$ not be a valid projector for removing the tachyon. We know that non-zero $n$ modifies the open string ground states such that the action of $\Omega$ is modified by (5.4). The cases with $n$ even descend from Type 0B, in which case we have $p$ even, whereas the cases with $n$ odd descend from Type 0A, in which case we have $p$ odd. Thus we conclude that $\Omega^2$ on the $|0; n\rangle_{pp}^{odd}$ ground state is given by $e^{-i\frac{\pi}{2}\eta n}e^{-i\frac{\pi}{2}p}$, and hence we must exclude cases for which $p + \eta n = \pm 2, 6, 10, 14, \ldots$.

To summarize, the conditions that must be satisfied by torsion $(9-k)$-branes of type $\eta$ in the theory with $n$ copies of ABK are the following,

$$\sin\left[\frac{\pi}{4}\nu\right] < 0 \,, \qquad\qquad 9 + \nu \neq \pm 2 \mod 8 \,. \tag{5.13}$$

Note that the combination of $n, k$ appearing here, namely $\nu = \eta n - k \mod 8$, is the same one appearing in Section 4 and Section 5.1. We have already checked above that the analysis via boundary states is in agreement with the previous analysis via boundary fermions when $n = 0$. Since the results (5.13) only depend on $\nu$, this agreement is simply extended to the general case.

## 5.3 Pin$^+$ Type 0 theories

We may now briefly turn to the analysis of the brane spectra in Pin$^+$ Type 0 theories. Using the action of $\Omega$ given in (5.3) (with $\nu \to -k$) and the action of $(-1)^{\text{f}_L}$ given in (B.8), one finds

$$\Omega_{\text{f}}|Bp, \eta\rangle_{\text{NSNS}} = -|Bp, -\eta\rangle_{\text{NSNS}} \,, \qquad \Omega_{\text{f}}|Bp, \eta\rangle_{\text{RR}} = i^{-\eta k}|Bp, -\eta\rangle_{\text{RR}} \,. \tag{5.14}$$

in the trivial phase. In the non-trivial phase, one gets an extra sign in the action of $\Omega_f$ on RR ground states, so more generally

$$\Omega_f|Bp,\eta\rangle_{\text{NSNS}} = -|Bp,-\eta\rangle_{\text{NSNS}}\,, \qquad \Omega_f|Bp,\eta\rangle_{\text{RR}} = i^{2n-\eta k}|Bp,-\eta\rangle_{\text{RR}}\,. \tag{5.15}$$

with $n = 0,1$ labelling the trivial or non-trivial phase. The stable D-brane states which are invariant under $\Omega_f$ then take the form

$$|Dp\rangle = \frac{1}{\mathcal{N}_{Bp}}\frac{1}{2}(|Bp,+\rangle_{\text{NSNS}} - |Bp,-\rangle_{\text{NSNS}} + |Bp,+\rangle_{\text{RR}} + i^{2n-k}|Bp,-\rangle_{\text{RR}})\,, \tag{5.16}$$

reminiscent of the Type II states obtained in (B.27). In fact, these states are invariant under $\Omega_f$ for *any* value of $k = 9-p$, both odd and even. On the other hand, the fully physical D-brane states must also be invariant under $(-1)^f$. It is easy to see that this requires $2n-k$ to be even. Thus for both $n = 0,1$, we keep all states with $p$ odd, reproducing the full spectrum of non-torsion branes in Type IIB.

We may now ask about torsion branes. These would take the same form as in Type I, given by

$$|\widetilde{Dp}\rangle = \frac{\lambda}{\mathcal{N}_{Bp}}\frac{1}{2}(|Bp,+\rangle_{\text{NSNS}} - |Bp,-\rangle_{\text{NSNS}})\,, \tag{5.17}$$

with $\lambda > 0$ a normalization factor. In order for such a brane to be stable, we require the open string tachyon contributions from the cylinder and Möbius strip amplitudes to cancel. In order to calculate the Möbius strip amplitude, one must make use of the appropriate orientifold plane state, which is obtained in Section B.4 and shown in (B.47). Importantly, note that the Pin$^+$ structure forces this state to be entirely in the RR sector. This means that the Möbius strip amplitude consists only of terms of the form $_{\text{NSNS}}\langle Bp|\ldots|Bp\rangle_{\text{RR}}$, which vanish. There is thus no Möbius strip contribution at all, and hence we cannot expect any cancellation of tachyons, and so no stable torsion branes. In conclusion, the spectrum of stable branes in these theories is precisely the same as for oriented Type IIB, and is classified by the complex K-group $K(X)$.

# 6 No new Type I theories

In this final section we classify unoriented Type II (i.e. Type I) strings. Unlike Type 0 strings for which one may consider Pin$^\pm$ theories separately, the unoriented Type II strings possess a more complicated spin extension of O($d$), which contains both Pin$^\pm(d)$ as subgroups. We refer to this as DPin structure, since it is a "doubled" Pin structure, and we define it precisely in Section 6.1. That such a thing is necessary is to be expected: as explained in Section 2.2, for Pin$^-$ structure the boundary circle of the Möbius strip is automatically in the NS sector, whereas for Pin$^+$ structure the boundary circle is automatically in the R sector. In contrast, we know that Type I strings allow both NS and R boundary conditions on the Möbius strip, so it is clear that these worldsheets must incorporate both Pin$^\pm$.

In order to understand possible anomalies and invertible phases on the worlsheet of unoriented Type II theories, it is necessary to calculate the groups $\mho_{\text{DPin}}^d(pt)$ for $d = 2,3$. This may be done using the Atiyah-Hirzebruch spectral sequence for twisted spin bordism, as will be recalled in Section 6.2. Details of the calculation are relegated to Appendix E. The final result is that $\mho_{\text{DPin}}^2(pt) = (\mathbb{Z}_2)^2$ and $\mho_{\text{DPin}}^3(pt) = \mathbb{Z}_8$. The latter implies that the unoriented Type II string is anomaly-free in ten dimensions. The former would naively suggest a quartet of string worldsheet theories, but as we discuss in Section 6.3 only two of these theories are physically distinct. The two distinct options are the traditional Type I and $\tilde{\text{I}}$ strings, corresponding to orientifoldings by orientifold O9$^\mp$-planes.

## 6.1 'Spin structure' on the Type I worldsheet

The oriented Type II worldsheet has separate spin structures for left- and right-movers, necessitating a $\mathrm{Spin} \times \mathbb{Z}_2$ structure. We would now like to understand the unoriented lift of this structure. We note that the $\mathbb{Z}_2$ part of the $\mathrm{Spin} \times \mathbb{Z}_2$ structure acts on $(\psi, \tilde{\psi})$ by $\mathrm{diag}(+1, -1)$, and should therefore be mapped to $\mathrm{diag}(-1, +1)$ under orientation reversal. To formalize, we then need an extension of $\mathrm{O}(d)$ by $\mathbb{Z}_2 \times \mathbb{Z}_2$,

$$0 \to \mathbb{Z}_2 \times \mathbb{Z}_2 \to G \to \mathrm{O}(d) \to 0 \tag{6.1}$$

such that the orientation reversal part of $\mathrm{O}(d)$ exchanges the two $\mathbb{Z}_2$ factors, and such that when restricted to $\mathrm{SO}(d)$ the extension class is given by $(w_2, w_2)$. Another way of saying this is that we would like a spin structure on the orientation double cover of the worldsheet. Note that for the worldsheet we have $d = 2$, but we will work more generally for the moment.

We now consider the effect of the homomorphism $s : \mathbb{Z}_2 \times \mathbb{Z}_2 \to \mathbb{Z}_2$, $(a, b) \mapsto ab$ in the extension. Over $\mathrm{SO}(d)$, the extension class of the image is $w_2 + w_2 = 0$. The image is also invariant under the orientation reversal part of $\mathrm{O}(d)$. Therefore the extension

$$0 \to s(\mathbb{Z}_2 \times \mathbb{Z}_2) \to s(G) \to \mathrm{O}(d) \to 0 \tag{6.2}$$

is trivial and can be split: $s(G) \simeq \mathrm{O}(d) \times \mathbb{Z}_2$. There are two natural splitting; once there is a splitting, we can compose it with

$$\begin{aligned} \mathrm{O}(d) \times \mathbb{Z}_2 &\to \mathrm{O}(d) \times \mathbb{Z}_2 \\ (g, c) &\mapsto (g, c \det g) \end{aligned} \tag{6.3}$$

to get another. So $G$ can also be put in the following sequence,

$$0 \to \mathbb{Z}_2 \to G \to \mathrm{O}(d) \times \mathbb{Z}_2 \to 0 \ . \tag{6.4}$$

Its extension class is a linear combination of $w_2$, $w_1^2$, $aw_1$, and $a^2$, where $a$ is the generator of $H^1(B\mathbb{Z}_2, \mathbb{Z}_2)$. To determine which linear combination, we may argue as follows. First, since $\mathbb{Z}_2$ is not extended to $\mathbb{Z}_4$ within $G$, the term $a^2$ is not involved. Second, since the entire group is not $\mathrm{Pin}^\pm \times \mathbb{Z}_2$, we need the term $aw_1$. Finally, since the extension is $\mathrm{Spin}(d) \times \mathbb{Z}_2$ over $\mathrm{SO}(d) \times \mathbb{Z}_2$, the class $w_2$ must be involved. This means that the extension class is either $w_2 + w_1 a$ or $w_2 + w_1^2 + w_1 a$. These two are exchanged by the change of the splitting, since (6.3) sends $a \mapsto a + w_1$. Let us pick $w_2 + w_1^2 + w_1 a$ for definiteness, and write $c : G \to \mathbb{Z}_2$ for the projection to the $\mathbb{Z}_2$ factor. The kernel of $c$ is $\mathrm{Pin}^-(d)$. The kernel of $\det g : G \to \mathrm{O}(d) \to \mathbb{Z}_2$ is $\mathrm{Spin}(d) \times \mathbb{Z}_2$, and the kernel of $c \det g$ is $\mathrm{Pin}^+(d)$. So $G$ is an interesting mixture of all three groups — as mentioned before, we refer to it as $G = \mathrm{DPin}(d)$.

For $d = 2$ we can construct the group $\mathrm{DPin}(2)$ more directly. We first let $\mathrm{Spin}(2)$ act on $(\psi, \tilde{\psi})$ via $\mathrm{diag}(e^{i\theta/2}, e^{-i\theta/2})$. Next we supplement this with a chiral $\mathbb{Z}_2$ acting via

$$Z = \mathrm{diag}(+1, -1) \ , \tag{6.5}$$

and then further include orientation-reversing elements of $\mathrm{Pin}^-(2)$, which can be chosen to be elements of the Clifford algebra $\mathrm{Cl}(-2)$,

$$\gamma_0 = \begin{pmatrix} 0 & i \\ i & 0 \end{pmatrix}, \qquad \gamma_1 = \begin{pmatrix} 0 & -1 \\ 1 & 0 \end{pmatrix}. \tag{6.6}$$

Note that we have

$$\gamma_0 \gamma_1 = \mathrm{diag}(i, -i), \tag{6.7}$$

which is a lift of a 180° rotation. This means that $\gamma_0 \gamma_1 Z = i\mathbb{1}$ is also a lift of a 180° rotation. Then we see that

$$\tilde{\gamma}_0 := i\gamma_1 , \qquad \tilde{\gamma}_1 := -i\gamma_0 \qquad (6.8)$$

are also in the group DPin(2). Together with Spin(2), these elements of Cl(+2) can be used to form $\text{Pin}^+(2)$. In this way, we see explicitly how DPin(2) contains all three of $\text{Spin}(2) \times \mathbb{Z}_2$, $\text{Pin}^-(2)$, and $\text{Pin}^+(2)$.

Now consider the Möbius strip $M_2$. As discussed above, the boundary circle of $M_2$ is automatically in the NS sector for $\text{Pin}^-$ since $\gamma_i^2 = -1$, whereas for $\text{Pin}^+$ structure the boundary circle of $M_2$ is automatically in the NS sector since $\tilde{\gamma}_i^2 = 1$. We may now contrast this with the case of DPin structure. In that case, we can use either $\gamma_i$ or $\tilde{\gamma}_i$ to construct $M_2$, and depending on this choice we can have NS or R on the boundary circle. From the boundary state point of view, this means that the orientifold plane states should have both NS and R sector contributions, which matches the well-known result (B.49).

## 6.2 The group $\mho^d_{\text{DPin}}(pt)$

To understand the anomalies and invertible phases present on unoriented Type II worldsheets, we must calculate $\mho^d_{\text{DPin}}(pt)$ for $d = 2, 3$. These groups may be calculated by using the Atiyah-Hirzebruch spectral sequence (AHSS) for twisted spin bordism groups.

We first recall twisted spin structures. We consider a space $X$ with a real vector bundle $V$ over it. Take a manifold $M$. A spin structure on $M$ twisted by $V$ is a pair

$$(f : M \to X , \text{ spin structure on } TM \oplus f^*(V)). \qquad (6.9)$$

We can then consider the corresponding bordism group $\Omega_d^{\text{Spin}}(X; V)$. Note that we have

$$w_1(TM) = f^*(w_1(V)), \qquad w_2(TM) = f^*(w_1(V)^2 + w_2(V)) \qquad (6.10)$$

since we have a spin structure on $TM \oplus f^*(V)$.

For example, when when $V$ is a zero-dimensional trivial bundle this reduces to an ordinary spin bordism of $X$. As another set of examples, take $L$ to be the real line bundle over $B\mathbb{Z}_2$ such that $w_1(L)$ is the generator of $H^1(B\mathbb{Z}_2, \mathbb{Z}_2) = \mathbb{Z}_2$. Then we have

$$\Omega_d^{\text{Spin}}(B\mathbb{Z}_2; L^{\oplus n}) = \begin{cases} \Omega_d^{\text{Spin}} \ (B\mathbb{Z}_2) & n = 0, \\ \Omega_d^{\text{Pin}^-} \ (pt) & n = 1, \\ \Omega_d^{\text{Spin}^{\mathbb{Z}_4}} \ (pt) & n = 2, \\ \Omega_d^{\text{Pin}^+} \ (pt) & n = 3, \end{cases} \qquad (6.11)$$

where $\text{Spin}^{\mathbb{Z}_4} = (\text{Spin} \times \mathbb{Z}_4)/\mathbb{Z}_2$.

The group DPin described above corresponds to taking $X = B\mathbb{Z}_2 \times B\mathbb{Z}_2$ and using $V = (L_1 \otimes L_2) \oplus L_2^{\oplus 3}$. Here $L_1$ and $L_2$ are real line bundles such that $w_1(L_1) = w$ and $w_1(L_2) = a$, where we denote the generators of $H^1(B\mathbb{Z}_2 \times B\mathbb{Z}_2, \mathbb{Z}_2) = \mathbb{Z}_2 \times \mathbb{Z}_2$ by $w$ and $a$. Then

$$w_1(V) = w, \qquad w_2(V) = wa. \qquad (6.12)$$

Let us now recall the basics of the AHSS. This review will not be comprehensive — for more thorough introductions to the AHSS, the reader can consult e.g. [18, 21, 25, 82]. The basic ingredients in the AHSS are the $E_2$ page and a set of differentials. For twisted spin bordism, the $E_2$ page consists of $E_2^{p,q} = H^p(X, \underline{\mho^q_{\text{Spin}}(pt)})$. The underline in the group $\underline{\mho^q_{\text{Spin}}(*)}$ denotes the fact that the coefficient system is twisted by $w_1(V) \in H^1(X, \mathbb{Z}_2)$. The differentials on the $E_2$ page are as follows:

- $d_2^2 : E_2^{p,2} = H^p(X, \mathbb{Z}_2) \to E_2^{p+2,1} = H^{p+2}(X, \mathbb{Z}_2)$ is given by

$$d_2^2(x) = \mathrm{Sq}^2 x + w_1(V)\,\mathrm{Sq}^1 x + w_2(V)x\,, \tag{6.13}$$

- $d_2^1 : E_2^{p,1} = H^p(X, \mathbb{Z}_2) \to E_2^{p+2,0} = H^{p+2}(X, \underline{U(1)})$ is given by

$$d_2^1(x) = \iota\left(\mathrm{Sq}^2 x + w_1(V)\,\mathrm{Sq}^1 x + w_2(V)x\right) = \iota\left(\mathrm{Sq}^2 x + w_2(V)x\right)\,, \tag{6.14}$$

where $\iota$ is the inclusion $\mathbb{Z}_2 \xhookrightarrow{\iota} U(1)$.[11]

These differentials, together with $d_2^3$, were determined in [83]. They were also deduced previously in [84, 85] when $w_1(V)$ is trivial. Alternatively, they can be deduced using the identity

$$\Omega_d^{\mathrm{Spin}}(X; V) = \tilde\Omega_{d+\dim V}^{\mathrm{Spin}}(\mathrm{Thom}(V))\,, \tag{6.15}$$

where $\mathrm{Thom}(V)$ is the Thom space of $V$. Indeed, denoting the Thom class by $U$, this equality implies

$$d_2^2(x)U = \mathrm{Sq}^2(xU), \qquad d_2^1(x)U = \iota(\mathrm{Sq}^2(xU)). \tag{6.16}$$

We then simply use the Cartan formula for the action of the Steenrod square, and the definition of the Stiefel-Whitney classes as $\mathrm{Sq}^d U = w_d(V)U$.

Details on our calculation will be given in Appendix E, where we demonstrate how one computes $\mho_X^{2,3}(pt)$ not only for $X = \mathrm{DPin}$ but also for $X = \mathrm{Spin} \times \mathbb{Z}_2$ and $\mathrm{Pin}^\pm$ to illustrate the methods involved. The results of the computation are that $\mho_{\mathrm{DPin}}^2(pt) = (\mathbb{Z}_2)^2$ and $\mho_{\mathrm{DPin}}^3(pt) = \mathbb{Z}_8$, precisely as for the oriented Type II strings.

## 6.3 Invertible phases for DPin structure

As in the case of oriented Type II strings, the fact that $\mho_{\mathrm{DPin}}^3(pt) = \mathbb{Z}_8$ means that worldsheet anomalies conveniently cancel in ten dimensions. Also as for oriented Type II strings, the result $\mho_{\mathrm{DPin}}^2(pt) = (\mathbb{Z}_2)^2$ implies four worldsheet theories, though two of these will be physically indistinct from the others. The generators of $\mho_{\mathrm{DPin}}^2(pt)$ can be taken to be $\{(-1)^{\int w_1^2}, (-1)^{\mathrm{Arf}(\hat\Sigma)}\}$, where $\hat\Sigma$ is the orientation double cover of $\Sigma$. The generator $(-1)^{\int w_1^2}$ is a bosonic invertible phase, which was discussed in detail in Section 2.2.1. The generator $(-1)^{\mathrm{Arf}(\hat\Sigma)}$ was discussed in the context of $\mathrm{Pin}^+$ in Section 2.2.4.

The effects of these phases on the Type I theory are easy to read off. We know that the presence of the phase $(-1)^{\int w_1^2}$, which has the effect of assigning $-1$ to Möbius strip amplitudes and $+1$ to cylinder and Klein bottle amplitudes, corresponds to the choice of $\mathrm{O}9^\pm$-planes, i.e. it distinguishes between Type I and Type Ĩ theories. On the other hand, adding $(-1)^{\mathrm{Arf}(\hat\Sigma)}$ leads to physically indistinct theories. To see this, note that on an oriented worldsheet $\Sigma$, the partition function contribution $(-1)^{\mathrm{Arf}(\hat\Sigma, \sigma)}$ can be interpreted as

$$(-1)^{\mathrm{Arf}(\Sigma, \sigma_L)} \times (-1)^{\mathrm{Arf}(\Sigma, \sigma_R)} \tag{6.17}$$

and can thus be absorbed for example by flipping $\psi^9$ and $\tilde\psi^9$ at the same time, i.e. by a spacetime parity flip in one direction, as explained in Section 3.1.2. So in fact the theory obtained from the non-trivial invertible phase $\mathrm{Arf}(\hat\Sigma)$ is not physically distinct from the one with the trivial phase, and the two are instead related in the same way that Type IIA/B and Type IIA/B$'$ were related.

---

[11]The second term in $d_2^1(x)$ can be dropped since the twisted Bockstein associated to $0 \to \mathbb{Z}_2 \to \underline{U(1)} \to \underline{U(1)} \to 0$ is $\mathrm{Sq}^1 + w(V)$, which implies that $\iota \circ (\mathrm{Sq}^1 + w(V)) = 0$. Therefore $\iota(w(V)\,\mathrm{Sq}^1 x) = \iota(\mathrm{Sq}^1\,\mathrm{Sq}^1 x) = 0$. The authors thank R. Thorngren for this point.

## Acknowledgments

The authors thank the TASI 2019 summer school at University of Colorado, Boulder, where JK and JPM were students and YT was a lecturer, for providing an ideal environment for collaboration; this paper grew out of a lunchtime conversation there. The authors are especially grateful to Edward Witten for many useful correspondences, and for providing much of the content of Appendix D. We also thank Arun Debray, Ryohei Kobayashi, and Ryan Thorngren for their help in computing bordism groups, as well as Oren Bergman, Jake McNamara, Max Metlitski, Matthew Heydeman, and Juven Wang for useful discussions. JPM thanks the Center for Mathematical Sciences and Applications at Harvard University for hospitality while this work was being completed. YT is in part supported by WPI Initiative, MEXT, Japan at IPMU, the University of Tokyo, and in part by JSPS KAKENHI Grant-in-Aid (Wakate-A), No.17H04837 and JSPS KAKENHI Grant-in-Aid (Kiban-S), No.16H06335. JPM is supported by the US Department of State through a Fulbright scholarship. JK and JPM thank the Mani L. Bhaumik Institute for generous support.

## A  NSR formalism

Throughout this paper we work in the NSR formalism, where the worldsheet fields for the closed string consist of scalars $X^\mu$ and left- and right-moving Majorana-Weyl fermions $\psi^\mu, \tilde{\psi}^\mu$, all of which are vectors in the ten-dimensional target space [8, 86].

**Closed strings:** We choose our conventions such that the worldsheet spatial coordinate $\sigma \in [0, 2\pi)$. For simplicity we work in light-cone gauge, with $\mu = 0, 1$ being the light-cone coordinates. In this gauge the worldsheet action is

$$\frac{1}{4\pi} \int dt\, d\sigma \left( \frac{1}{\alpha'} \left( \partial_t X^\mu \partial_t X_\mu - \partial_\sigma X^\mu \partial_\sigma X_\mu \right) + i\psi^\mu (\partial_t + \partial_\sigma)\psi_\mu + i\tilde{\psi}^\mu (\partial_t - \partial_\sigma)\tilde{\psi}_\mu \right), \quad (A.1)$$

with $\mu = 2, \cdots, 9$. The oscillator expansions for these fields are

$$X^\mu(t, \sigma) = x^\mu + \alpha' t\, p^\mu + i\sqrt{\frac{\alpha'}{2}} \sum_n \frac{1}{n} \left( \alpha_n^\mu e^{-in(t-\sigma)} + \tilde{\alpha}_n^\mu e^{-in(t+\sigma)} \right), \quad (A.2)$$

$$\psi^\mu(t, \sigma) = \sum_r \psi_r^\mu e^{-ir(t-\sigma)}, \qquad \tilde{\psi}^\mu(t, \sigma) = \sum_r \tilde{\psi}_r^\mu e^{-ir(t+\sigma)}, \quad (A.3)$$

where $r \in \mathbb{Z}$ or $\mathbb{Z} + \frac{1}{2}$ for R or NS boundary conditions, respectively.

We are mainly interested in the fermionic sector, and in particular in its zero-modes. There are no zero-modes for NS boundary conditions, and hence the left- and right-moving ground states, $|0\rangle_{NS}$ and $|\tilde{0}\rangle_{NS}$, are unique in this sector. It is easy to write fermion number operators in terms of the operators that count the number of modes, $N = \sum_{r,\mu} r\, \psi_{-r}^\mu \psi_r^\mu$ and $\tilde{N} = \sum_{r,\mu} r\, \tilde{\psi}_{-r}^\mu \tilde{\psi}_r^\mu$, as $(-1)^{f_L} = (-1)^{N-1}$ and $(-1)^{f_R} = (-1)^{\tilde{N}-1}$. The ground states are odd, i.e.

$$(-1)^{f_L}|0\rangle_{NS} = -|0\rangle_{NS}, \qquad (-1)^{f_R}|\tilde{0}\rangle_{NS} = -|\tilde{0}\rangle_{NS}. \quad (A.4)$$

On the other hand, fermions with R boundary conditions do allow zero-modes, which satisfy the anticommutation relations

$$\{\psi_0^\mu, \psi_0^\nu\} = \delta^{\mu\nu}, \qquad \{\tilde{\psi}_0^\mu, \tilde{\psi}_0^\nu\} = \delta^{\mu\nu}, \qquad \{\psi_0^\mu, \tilde{\psi}_0^\nu\} = 0. \quad (A.5)$$

Left- and right-moving zero-modes then separately furnish representations of the Clifford algebra Cl(8). These representations act on the degenerate ground states, which can be spinors $|0\rangle_R^a, |\tilde{0}\rangle_R^a \in \mathbf{8}_s$ or conjugate spinors $|0\rangle_R^{\dot{a}}, |\tilde{0}\rangle_R^{\dot{a}} \in \mathbf{8}_c$ of the little group $SO(8)$. The two irreducible representations are distinguished by the eigenvalue of the left- and right-moving fermion numbers

$$(-1)_0^{\mathsf{f_L}} = \prod_\mu \sqrt{2}\,\psi_0^\mu, \qquad (-1)_0^{\mathsf{f_R}} = \prod_\mu \sqrt{2}\,\tilde{\psi}_0^\mu. \tag{A.6}$$

The full fermion number operators are obtained by combining these with the operators counting the number of massive modes,

$$(-1)^{\mathsf{f_L}} = (-1)_0^{\mathsf{f_L}}(-1)^{\mathsf{N}}, \qquad (-1)^{\mathsf{f_R}} = (-1)_0^{\mathsf{f_R}}(-1)^{\tilde{\mathsf{N}}}. \tag{A.7}$$

Worldsheet parity $\Omega$ acts on the worldsheet fields as in (3.13), from which it follows that the action on the oscillator modes in (A.2) and (A.3) is

$$\Omega\alpha_n\Omega = \tilde{\alpha}_n, \qquad \Omega\psi_r\Omega = e^{2\pi r}\tilde{\psi}_r, \qquad \Omega\tilde{\psi}_r\Omega = -e^{2\pi r}\psi_r. \tag{A.8}$$

Next we review the low-lying spectra of the closed superstring. We will use the usual notation [8] and denote each of the low-lying states by $(A\pm, B\pm)$, where $A, B$ can be R or NS and denote respectively the left- and right-moving sectors, while the signs indicate the fermion parity. The NSNS sector ground state is

$$(\text{NS}-, \text{NS}-): \quad |0\rangle_{\text{NSNS}} = |0\rangle_{\text{NS}} \otimes |\tilde{0}\rangle_{\text{NS}}, \tag{A.9}$$

and is a scalar tachyon. The massless NSNS spectrum is given by the level one states,

$$(\text{NS}+, \text{NS}+): \quad \tilde{\psi}_{-1/2}^\mu \tilde{\psi}_{-1/2}^\nu |0\rangle_{\text{NSNS}} \in \mathbf{8}_v \otimes \mathbf{8}_v = \mathbf{1} \oplus \mathbf{28} \oplus \mathbf{35}_v \tag{A.10}$$

and consist of a dilaton, $\mathbf{1}$, 2-form, $\mathbf{28}$, and graviton, $\mathbf{35}_v$. The RR sector ground states transform in the product of spinor representations of the little group $SO(8)$,

$$
\begin{aligned}
(\text{R}+, \text{R}+): \quad & |0\rangle_{\text{RR}}^{ab} = |0\rangle_R^a \otimes |\tilde{0}\rangle_R^b \in \mathbf{8}_s \otimes \mathbf{8}_s = \mathbf{1} \oplus \mathbf{28} \oplus \mathbf{35}_-, \\
(\text{R}-, \text{R}-): \quad & |0\rangle_{\text{RR}}^{\dot{a}\dot{b}} = |0\rangle_R^{\dot{a}} \otimes |\tilde{0}\rangle_R^{\dot{b}} \in \mathbf{8}_c \otimes \mathbf{8}_c = \mathbf{1} \oplus \mathbf{28} \oplus \mathbf{35}_+, \\
(\text{R}+, \text{R}-): \quad & |0\rangle_{\text{RR}}^{a\dot{b}} = |0\rangle_R^a \otimes |\tilde{0}\rangle_R^{\dot{b}} \in \mathbf{8}_s \otimes \mathbf{8}_c = \mathbf{8}_v \oplus \mathbf{56}, \\
(\text{R}-, \text{R}+): \quad & |0\rangle_{\text{RR}}^{\dot{a}b} = |0\rangle_R^{\dot{a}} \otimes |\tilde{0}\rangle_R^b \in \mathbf{8}_c \otimes \mathbf{8}_s = \mathbf{8}_v \oplus \mathbf{56}.
\end{aligned} \tag{A.11}
$$

All of these states are massless and can be identified as the RR scalar $\mathbf{1}$, vector $\mathbf{8}_v$, two-form $\mathbf{28}$, three-form $\mathbf{56}$, and (anti)self-dual four-form $\mathbf{35}_{(-)+}$. Finally, the NSR states are

$$
\begin{aligned}
(\text{NS}+, \text{R}+): \quad & \psi_{-1/2}^\mu |0\rangle_{\text{NS}} \otimes |\tilde{0}\rangle_R^a \in \mathbf{8}_v \otimes \mathbf{8}_s = \mathbf{8}_c \oplus \mathbf{56}_s, \\
(\text{NS}+, \text{R}-): \quad & \psi_{-1/2}^\mu |0\rangle_{\text{NS}} \otimes |\tilde{0}\rangle_R^{\dot{a}} \in \mathbf{8}_v \otimes \mathbf{8}_c = \mathbf{8}_s \oplus \mathbf{56}_c,
\end{aligned} \tag{A.12}
$$

which include the dilatinos, $\mathbf{8}_s$ and $\mathbf{8}_c$, and gravitinos, $\mathbf{56}_s$ and $\mathbf{56}_c$. The RNS states are conjugate to these.

**Open strings:** When considering open strings, one can have Dirichlet (D) or Neumann (N) boundary conditions on each end, which relate the left- and right-moving oscillators. We define the NN, DD, and ND directions of the open string as in (4.13). Though only the relative sign in those conditions is important, the conventions shown there are such that $\eta_{1,2} = +1$ represents

a D$p$-brane at $\sigma = 0, \pi$, while $\eta_{1,2} = -1$ represents a D$p'$-brane at $\sigma = 0, \pi$.[12] That this is so will be discussed in the beginning of Appendix B.

Note that the relative choice of $\eta_{1,2}$ is related to the choice of NS or R sectors on the open string. In particular, the string is in the NS sector for $\eta_1 = \eta_2$ and in the R sector for $\eta_1 = -\eta_2$. This may be seen as follows. First, we may replace the left- and right-moving fermions on $\sigma \in [0, \pi]$ with a single chiral fermion on $\sigma \in [0, 2\pi]$ by defining

$$\psi(t, \sigma) = \eta_2 \, \tilde{\psi}(t, 2\pi - \sigma), \qquad \pi \leq \sigma \leq 2\pi . \tag{A.13}$$

The question of NS vs. R is then a question about the (anti-)periodicity of this extended fermion. In particular, say that $\tilde{\psi}(t, 2\pi) = \eta_3 \, \tilde{\psi}(t, 0)$ where $\eta_3 = +1$ for R and $-1$ for NS. Then we have

$$\psi(t, 0) = \eta_2 \, \tilde{\psi}(t, 2\pi) = \eta_2 \eta_3 \, \tilde{\psi}(t, 0) . \tag{A.14}$$

But from (4.13), we also have $\psi(t, 0) = -\eta_1 \, \tilde{\psi}(t, 0)$ and hence we conclude that $\eta_1 \eta_2 = -\eta_3$. The anti-periodic NS case then corresponds to $\eta_1 = \eta_2$, while the periodic R case corresponds to $\eta_1 = -\eta_2$. From (4.13) we then conclude that NS sector fermions can have zero modes along ND and DN directions, whereas R sector fermions can have zero modes along NN and DD directions. In constructing the open string fermion number operator $(-1)^{\text{f}}$ one must take these zero modes into account, as was done above for the RR sector of the closed string.

As for the open string spectrum, let us just mention that the ground state in the NS sector, $|0\rangle_{\text{NS}}$, is an open string tachyon with $(-1)^{\text{f}} |0\rangle_{\text{NS}} = -|0\rangle_{\text{NS}}$. We will not review the massless spectrum here.

# B  Boundary state formalism

Here we review the boundary state formalism — for more details, the reader may consult [63, 86, 87].

## B.1  Basics

From the closed string point of view, D-branes correspond to states in the closed string Hilbert space. Beginning with the boundary conditions (4.13), one can transition to Euclidean signature $t \rightarrow -i t_E$, and then do a rotation of the worldsheet to interchange the $t_E$ and $\sigma$ directions, thereby going from the open string to the closed string picture. In particular, rotation by $\frac{\pi}{2}$ takes $(t_E, \sigma) \rightarrow (\sigma, -t_E)$. Under this transformation, the fermions transform as

$$\psi^\mu(t_E + i\sigma) \rightarrow e^{i\pi/4} \psi^\mu(t_E + i\sigma), \qquad \tilde{\psi}^\mu(t_E - i\sigma) \rightarrow e^{-i\pi/4} \tilde{\psi}^\mu(t_E - i\sigma) . \tag{B.1}$$

The boundary conditions at fixed $\sigma$ then become conditions at fixed time — for example, on the slice $t_E = 0$ the initial conditions are

$$\begin{aligned} \text{N}: \quad \psi^\mu(0, \sigma) &= +i\eta_1 \, \tilde{\psi}^\mu(0, \sigma), \\ \text{D}: \quad \psi^\mu(0, \sigma) &= -i\eta_1 \, \tilde{\psi}^\mu(0, \sigma), \qquad \sigma \in [0, 2\pi] . \end{aligned} \tag{B.2}$$

The conditions at $t_E = \pi$ look the same, but with $\eta_1$ replaced by $\eta_2$ — note that we no longer have the relative minus sign for the same condition on the two boundaries, c.f. footnote 12.

---

[12]Note that the extra sign in (4.13) is needed to encode the *same* physical boundary condition at $\sigma = \pi$ as at $\sigma = 0$, as explained e.g. in footnote 69 of [63]. To summarize, because the boundaries at $\sigma = 0, \pi$ have opposite orientation, imposing the same boundary condition on the two boundaries involves a reflection of one of them $t \rightarrow -t$, under which $\psi^\mu \rightarrow e^{i\pi/2} \psi^\mu$ and $\tilde{\psi}^\mu \rightarrow e^{-i\pi/2} \tilde{\psi}^\mu$.

Let us now focus on the slice at $t_E = 0$. We will denote $\eta_1 = \eta$ for simplicity. We define the boundary states $|\mathrm{B}p, \eta\rangle$ to be the operator statement of the boundary conditions (B.2) on the closed string Hilbert space. Writing things in terms of Fourier modes and including bosonic constraints as well, these boundary states are then defined by

$$
\begin{aligned}
\text{N}: \quad & (\alpha_n^\mu - \tilde{\alpha}_{-n}^\mu)|\mathrm{B}p, \eta\rangle = (\psi_r^\mu - i\eta\tilde{\psi}_{-r}^\mu)|\mathrm{B}p, \eta\rangle = 0 \, , \\
\text{D}: \quad & (\alpha_n^\mu + \tilde{\alpha}_{-n}^\mu)|\mathrm{B}p, \eta\rangle = (\psi_r^\mu + i\eta\tilde{\psi}_{-r}^\mu)|\mathrm{B}p, \eta\rangle = 0 \, ,
\end{aligned}
\tag{B.3}
$$

where $r$ is integer/half-integer for the R/NS sector. The general solution to these conditions can be written as a coherent state,

$$
\begin{aligned}
|\mathrm{B}p, \eta\rangle \propto \exp\Bigg\{ &\sum_{n=1}^\infty \left[ -\frac{1}{n}\sum_{\mu=2}^{p+2} \alpha_{-n}^\mu \tilde{\alpha}_{-n}^\mu + \frac{1}{n}\sum_{\mu=p+3}^{9} \alpha_{-n}^\mu \tilde{\alpha}_{-n}^\mu \right] \\
&+ i\eta\sum_{r>0}^\infty \left[ -\sum_{\mu=2}^{p+2} \psi_{-r}^\mu \tilde{\psi}_{-r}^\mu + \sum_{\mu=p+3}^{9} \psi_{-r}^\mu \tilde{\psi}_{-r}^\mu \right] \Bigg\} |\mathrm{B}p, \eta\rangle^{(0)}
\end{aligned}
\tag{B.4}
$$

up to a normalization factor which we discuss in Appendix B.3. Here $|\mathrm{B}p, \eta\rangle^{(0)}$ is the ground state, which depends on the sector. In the NSNS sector the ground state is just the usual one

$$
|\mathrm{B}p, \eta\rangle_{\mathrm{NSNS}}^{(0)} = |0\rangle_{\mathrm{NSNS}} \, .
\tag{B.5}
$$

In the RR sector there is an extra subtlety because we need to solve the gluing conditions for the zero-modes. This is easily done by noticing that $|\mathrm{B}7, \eta\rangle_{\mathrm{RR}}^{(0)}$ need only satisfy conditions of the kind

$$
(\psi_0^\mu - i\eta\tilde{\psi}_0^\mu)|\mathrm{B}7, \eta\rangle_{\mathrm{RR}}^{(0)} = 0 \, .
\tag{B.6}
$$

We can then build the rest of the boundary ground states as

$$
|\mathrm{B}p, \eta\rangle_{\mathrm{RR}}^{(0)} = \prod_{\mu=p+3}^{9} (\psi_0^\mu + i\eta\tilde{\psi}_0^\mu)|\mathrm{B}7, \eta\rangle_{\mathrm{RR}}^{(0)} \, .
\tag{B.7}
$$

It is not entirely trivial to see the relation between $|\mathrm{B}7, \eta\rangle_{\mathrm{RR}}^{(0)}$ and the usual RR vacuum. This is explained, for instance, in Appendix B of [87]. The key feature is that the relation involves an even number of RR zero-mode operators. With this in mind it follows that

$$
\begin{aligned}
(-1)^{\mathsf{f}_\mathsf{L}}|\mathrm{B}p, \eta\rangle_{\mathrm{NSNS}} = -|\mathrm{B}p, -\eta\rangle_{\mathrm{NSNS}} \, , \qquad & (-1)^{\mathsf{f}_\mathsf{L}}|\mathrm{B}p, \eta\rangle_{\mathrm{RR}} = (-1)^{7-p}|\mathrm{B}p, -\eta\rangle_{\mathrm{RR}} \, , \\
(-1)^{\mathsf{f}_\mathsf{R}}|\mathrm{B}p, \eta\rangle_{\mathrm{NSNS}} = -|\mathrm{B}p, -\eta\rangle_{\mathrm{NSNS}} \, , \qquad & (-1)^{\mathsf{f}_\mathsf{R}}|\mathrm{B}p, \eta\rangle_{\mathrm{RR}} = |\mathrm{B}p, -\eta\rangle_{\mathrm{RR}} \, ,
\end{aligned}
\tag{B.8}
$$

where $(-1)^{\mathsf{f}_\mathsf{L}}$ and $(-1)^{\mathsf{f}_\mathsf{R}}$ are the left- and right-moving worldsheet fermion numbers. Similarly using (A.8) and (B.7) one can check that

$$
\Omega|\mathrm{B}p, \eta\rangle_{\mathrm{NSNS}} = |\mathrm{B}p, \eta\rangle_{\mathrm{NSNS}} \, , \qquad \Omega|\mathrm{B}p, \eta\rangle_{\mathrm{RR}} = -(-i\eta)^{7-p}|\mathrm{B}p, \eta\rangle_{\mathrm{RR}} = i^{-\eta k}|\mathrm{B}p, \eta\rangle_{\mathrm{RR}} \, .
\tag{B.9}
$$

where $k = 9 - p$.

## B.2 Theta functions, partition functions and boundary state amplitudes

In the remainder of this appendix, we will be calculating amplitudes for closed strings propagating between boundary states. In order to do so, some preliminary data will be needed. We now review our conventions for the different theta functions that appear in one-loop string

partition functions, and give a few useful formulas for partition functions and boundary state amplitudes.

First, note that we will use the usual shorthand for theta functions with characteristic,

$$\vartheta_1(z|\tau) = \vartheta\begin{bmatrix}\frac{1}{2}\\\frac{1}{2}\end{bmatrix}(z|\tau), \qquad \vartheta_2(z|\tau) = \vartheta\begin{bmatrix}\frac{1}{2}\\0\end{bmatrix}(z|\tau),$$

$$\vartheta_3(z|\tau) = \vartheta\begin{bmatrix}0\\0\end{bmatrix}(z|\tau), \qquad \vartheta_4(z|\tau) = \vartheta\begin{bmatrix}0\\\frac{1}{2}\end{bmatrix}(z|\tau), \tag{B.10}$$

and the notation $\vartheta_i(\tau) = \vartheta_i(0|\tau)$, where $\tau$ is the modular parameter of a torus. Recall that $\vartheta_1$ is odd and vanishes at the origin, i.e. $\vartheta_1(\tau) = 0$. For convenience we define the following combinations,

$$f_1(\tau) = \eta(\tau) = q^{1/12}\prod_{n=1}^{\infty}(1 - q^{2n}), \qquad f_2(\tau) = \sqrt{\frac{\vartheta_2(\tau)}{\eta(\tau)}} = \sqrt{2}q^{1/12}\prod_{n=1}^{\infty}(1 + q^{2n}),$$

$$f_3(\tau) = \sqrt{\frac{\vartheta_3(\tau)}{\eta(\tau)}} = q^{-1/24}\prod_{n=1}^{\infty}(1 + q^{2n-1}), \quad f_4(\tau) = \sqrt{\frac{\vartheta_4(\tau)}{\eta(\tau)}} = q^{-1/24}\prod_{n=1}^{\infty}(1 - q^{2n-1}), \tag{B.11}$$

where $q = e^{i\pi\tau}$ and $\eta(\tau) = \left(\vartheta_1'(0|\tau)/2\pi\right)^{1/3}$ is the Dedekind eta function. It will be useful to know the modular $S$ transformations,

$$\begin{aligned} f_1(i/t) &= \sqrt{t}f_1(it), & f_2(i/t) &= f_4(it), \\ f_3(i/t) &= f_3(it), & f_4(i/t) &= f_2(it), \end{aligned} \tag{B.12}$$

and the $T$ transformations,

$$\begin{aligned} f_1(it+1) &= e^{i\frac{\pi}{12}}f_1(it), & f_2(it+1) &= e^{i\frac{\pi}{12}}f_2(it), \\ f_3(it+1) &= e^{-i\frac{\pi}{24}}f_4(it), & f_4(it+1) &= e^{-i\frac{\pi}{24}}f_3(it), \end{aligned} \tag{B.13}$$

as well as the more unfamiliar $P = T^{\frac{1}{2}}ST^2ST^{\frac{1}{2}}$ transformation,

$$\begin{aligned} f_1\left(\frac{i}{4t}+\frac{1}{2}\right) &= \sqrt{2t}f_1\left(it+\frac{1}{2}\right), & f_2\left(\frac{i}{4t}+\frac{1}{2}\right) &= f_2\left(it+\frac{1}{2}\right), \\ f_3\left(\frac{i}{4t}+\frac{1}{2}\right) &= e^{i\frac{\pi}{8}}f_4\left(it+\frac{1}{2}\right), & f_4\left(\frac{i}{4t}+\frac{1}{2}\right) &= e^{-i\frac{\pi}{8}}f_3\left(it+\frac{1}{2}\right), \end{aligned} \tag{B.14}$$

where $t \in \mathbb{R}$. The Jacobi "abstruse" and "triple product" identities

$$f_2(\tau)^8 - f_3(\tau)^8 + f_4(\tau)^8 = 0, \qquad\qquad f_2(\tau)f_3(\tau)f_4(\tau) = \sqrt{2} \tag{B.15}$$

will be used to simplify results.

The functions $f_i(\tau)$ introduced above are useful since the open and closed string sector traces are written naturally in terms of them. Denote the boundary state amplitudes in the tree-channel in sector S as follows

$$\tilde{Z}_S^{\mathrm{LR}} = {}_{S,L}\langle B|e^{-2\pi lH_{\mathrm{cl}}}|B\rangle_{S,R}, \tag{B.16}$$

where for fermionic sectors $|B\rangle = |B, \eta\rangle$ and L, R denote the boundary conditions — either Neumann (N) or Dirichlet (D) — on each boundary. The amplitudes with opposite $\eta$ on either

side are given by exchanging N $\leftrightarrow$ D on one boundary state, as can be seen in (B.3). In terms of the $f_i(\tau)$, the bosonic contributions are found to be [86]

$$\tilde{Z}_B^{NN} = \frac{1}{f_1(2il)}, \qquad \tilde{Z}_B^{ND} = \frac{\sqrt{2}}{f_2(2il)}, \qquad \tilde{Z}_B^{DD} = \frac{1}{\sqrt{4\pi^2\alpha'l}}\frac{1}{f_1(2il)}, \tag{B.17}$$

the fermionic contributions in the NSNS sector are

$$\tilde{Z}_{NSNS}^{NN} = f_3(2il), \qquad \tilde{Z}_{NSNS}^{ND} = f_4(2il), \qquad \tilde{Z}_{NSNS}^{DD} = f_3(2il), \tag{B.18}$$

and in the RR sector

$$\tilde{Z}_{RR}^{NN} = -f_2(2il), \qquad \tilde{Z}_{RR}^{ND} = 0, \qquad \tilde{Z}_{RR}^{DD} = -f_2(2il). \tag{B.19}$$

Consider parallel B$p$ and B$q$ boundary states with $q > p$. Note that there are $p-1$ NN, $9-q$ DD, and $q - p$ ND directions. Also recall that changing $\eta \to -\eta$ is equivalent to exchanging the boundary conditions N $\leftrightarrow$ D. The amplitudes for exchanging closed strings between these states then take the simple form

$$\begin{aligned}
_{NSNS}\langle Bp, \eta | e^{-2\pi lH_{cl}} | Bq, \eta\rangle_{NSNS} &= \frac{V_{p+1}}{(4\pi^2\alpha'l)^{\frac{9-q}{2}}} \frac{f_3(2il)^8 f_4(2il)^{2(q-p)}}{f_1(2il)^{8-q+p}}, \\
_{NSNS}\langle Bp, \eta | e^{-2\pi lH_{cl}} | Bq, -\eta\rangle_{NSNS} &= \frac{V_{p+1}}{(4\pi^2\alpha'l)^{\frac{9-q}{2}}} \frac{f_4(2il)^8 f_3(2i\ell)^{2(q-p)}}{f_1(2il)^{8-q+p}}, \\
_{RR}\langle Bp, \eta | e^{-2\pi lH_{cl}} | Bq, \eta\rangle_{RR} &= -\frac{V_{p+1}}{(4\pi^2\alpha'l)^{\frac{9-q}{2}}} \frac{f_2(2il)^8}{f_1(2il)^8}, \\
_{RR}\langle Bp, \eta | e^{-2\pi lH_{cl}} | Bq, -\eta\rangle_{RR} &= 0,
\end{aligned} \tag{B.20}$$

where $V_{p+1}$ is the regularized volume of the $p$-brane, which comes from the zero-modes of the scalars parallel to its worldvolume. Note that we have used the Jacobi triple product identity (B.15) to remove some factors of $\sqrt{2}/f_2(2il)$ from these results. The last amplitude vanishes because $\eta_L = -\eta_R$ corresponds to Ramond boundary conditions in the direction orthogonal to the boundary, for which there is a fermion zero-mode.

## B.3 D-brane boundary states

### B.3.1 Boundary state normalization

In order to calculate tensions or tadpole contributions, we will want to find the proper normalization for the boundary states. The way to do this is to impose matching of the tree- and loop-channel cylinder amplitudes. That is, we want to impose the following identities

$$\begin{aligned}
\frac{1}{\mathcal{N}_{Bp}^2}\int_0^\infty dl\,_{NSNS}\langle Bp, \eta | e^{-2\pi lH_{cl}} | Bp, \eta\rangle_{NSNS} &= \int_0^\infty \frac{dt}{2t} \text{Tr}_{NS}\left[e^{-2\pi tH_{op}}\right], \\
\frac{1}{\mathcal{N}_{Bp}^2}\int_0^\infty dl\,_{NSNS}\langle Bp, \eta | e^{-2\pi lH_{cl}} | Bp, -\eta\rangle_{NSNS} &= \int_0^\infty \frac{dt}{2t} \text{Tr}_R\left[e^{-2\pi tH_{op}}\right], \\
\frac{1}{\mathcal{N}_{Bp}^2}\int_0^\infty dl\,_{RR}\langle Bp, \eta | e^{-2\pi lH_{cl}} | Bp, \eta\rangle_{RR} &= \int_0^\infty \frac{dt}{2t} \text{Tr}_{NS}\left[e^{-2\pi tH_{op}}(-1)^f\right], \\
\frac{1}{\mathcal{N}_{Bp}^2}\int_0^\infty dl\,_{RR}\langle Bp, \eta | e^{-2\pi lH_{cl}} | Bp, -\eta\rangle_{RR} &= \int_0^\infty \frac{dt}{2t} \text{Tr}_R\left[e^{-2\pi tH_{op}}(-1)^f\right] = 0.
\end{aligned}$$

We may for instance focus on the first one, which in loop-channel gives the result

$$\int_0^\infty \frac{dt}{2t} \text{Tr}_{\text{NS}}\left[e^{-2\pi t H_{\text{op}}}\right] = v_{p+1} \int_0^\infty \frac{dt}{(2t)^{\frac{p+3}{2}}} \frac{f_3(it)^8}{f_1(it)^8}, \tag{B.21}$$

where $v_{p+1} = V_{p+1}/(4\pi^2\alpha')^{\frac{p+1}{2}}$ is the regularized volume of the brane, in units of the string length. On the other hand, in tree-channel we have

$$\frac{1}{\mathcal{N}_{\text{B}p}^2}\int_0^\infty dl \,_{\text{NSNS}}\langle \text{B}p, \eta | e^{-2\pi l H_{\text{cl}}} | \text{B}p, \eta \rangle_{\text{NSNS}} = \frac{(4\pi^2\alpha')^{p-4}}{\mathcal{N}_{\text{B}p}^2} v_{p+1} \int_0^\infty \frac{dl}{l^{\frac{9-p}{2}}} \frac{f_3(2il)^8}{f_1(2il)^8}. \tag{B.22}$$

We can translate the tree-channel amplitude to loop-channel using the transformation $l = \frac{1}{2t}$ and the modular $S$ transformations in (B.12), which yields

$$\frac{(4\pi^2\alpha')^{p-4}}{\mathcal{N}_{\text{B}p}^2} v_{p+1} \int_0^\infty \frac{dt}{2t^2} \frac{1}{(2t)^{\frac{p-9}{2}}} \frac{f_3(it)^8}{(t)^{\frac{8}{2}} f_1(it)^8} \tag{B.23}$$

$$= \frac{2^5(4\pi^2\alpha')^{p-4}}{\mathcal{N}_{\text{B}p}^2} v_{p+1} \int_0^\infty \frac{dt}{(2t)^{\frac{p+3}{2}}} \frac{f_3(it)^8}{f_1(it)^8}. \tag{B.24}$$

Comparing (B.21) and (B.24), we find that the proper normalization for the boundary state is

$$\mathcal{N}_{\text{B}p} = 2^{\frac{5}{2}}(4\pi^2\alpha')^{\frac{p-4}{2}}. \tag{B.25}$$

One can check that imposing the other identities gives the same result.

### B.3.2 Type II and I

The states $|\text{B}p, \eta\rangle$ must be assembled into a D-brane state such that they give the right open string amplitudes. Let us begin by finding the D-brane state in Type II. Since the open string sector includes both NS and R strings we must have

$$\int_0^\infty dl \,\langle \text{D}p | e^{-2\pi l H_{\text{cl}}} | \text{D}p \rangle \tag{B.26}$$

$$= \int_0^\infty \frac{dt}{2t} \left(\text{Tr}_{\text{NS}}\left[e^{-2\pi t H_{\text{op}}}\frac{1}{2}(1+(-1)^{\text{f}})\right] - \text{Tr}_{\text{R}}\left[e^{-2\pi t H_{\text{op}}}\frac{1}{2}(1+(-1)^{\text{f}})\right]\right).$$

A brief calculation then shows that the proper normalization for the D-brane state is

$$|\text{D}p\rangle = \frac{1}{\mathcal{N}_{\text{B}p}} \frac{1}{2}\left(|\text{B}p, +\rangle_{\text{NSNS}} - |\text{B}p, -\rangle_{\text{NSNS}} + |\text{B}p, +\rangle_{\text{RR}} + |\text{B}p, -\rangle_{\text{RR}}\right), \tag{B.27}$$

with $\mathcal{N}_{\text{B}p}$ as defined in (B.25). The choice of relative sign of the NSNS and RR contributions differentiates branes and anti-branes.

Recall that for Type II theories we wish to gauge both $(-1)^{\text{f}_{\text{L,R}}}$, so we should keep only states invariant under projection by

$$P_{\text{NSNS}}^{\text{II}} = \frac{1}{4}\left(1+(-1)^{\text{f}_{\text{L}}}\right)\left(1+(-1)^{\text{f}_{\text{R}}}\right), \qquad P_{\text{RR}}^{\text{II}} = \frac{1}{4}\left(1+(-1)^{\text{f}_{\text{L}}}\right)\left(1\pm(-1)^{\text{f}_{\text{R}}}\right), \tag{B.28}$$

with the two choices of sign corresponding to Type IIB $(+)$ and Type IIA $(-)$. Using (B.8) it is easy to see that the boundary states in Eq. (B.27) are invariant when $p$ is odd for Type IIB and when $p$ is even for Type IIA.

In the presence of multiple branes the boundary state acquires an extra overall group theory factor $G$ that accounts for the trace over the Chan-Paton space,

$$|Dp\rangle \to G|Dp\rangle \quad \text{where} \quad G = \begin{cases} N & \text{for } U(N), \\ 2N & \text{for } Sp(N),\ SO(2N) \end{cases}. \tag{B.29}$$

The latter implies that in Type I the D-brane states for even a single brane are normalized with an extra factor of 2.

### B.3.3 Type 0

We now do the same analysis for D-branes in Type 0. In Type 0 the open strings stretching between two branes of the same (different) type are in the NS (R) sector, so we must have

$$\int_0^\infty dl\, \langle Dp, \eta | e^{-2\pi l H_{\text{cl}}} | Dp, \eta \rangle = \int_0^\infty \frac{dt}{2t} \text{Tr}_{\text{NS}} \left[ e^{-2\pi t H_{\text{op}}} \frac{1}{2}(1 + (-1)^{\text{f}}) \right], \tag{B.30}$$

$$\int_0^\infty dl\, \langle Dp, \eta | e^{-2\pi l H_{\text{cl}}} | Dp, -\eta \rangle = -\int_0^\infty \frac{dt}{2t} \text{Tr}_{\text{R}} \left[ e^{-2\pi t H_{\text{op}}} \frac{1}{2}(1 + (-1)^{\text{f}}) \right]. \tag{B.31}$$

From this and the relations above it follows that the properly normalized Type 0 D-brane state is

$$|Dp, \eta\rangle = \frac{1}{\mathcal{N}_{Bp}} \frac{1}{\sqrt{2}} (\eta | Bp, \eta\rangle_{\text{NSNS}} + | Bp, \eta\rangle_{\text{RR}}), \qquad \eta = \pm 1. \tag{B.32}$$

The factor of $\eta$ in front of $|Bp, \eta\rangle_{\text{NSNS}}$ is needed so that the force between the branes is attractive. To see this, consider the NSNS contribution to the amplitude,

$$\int_0^\infty dl\, {}_{NSNS}\langle Dp, \eta | e^{-2\pi l H_{\text{cl}}} | Dp, -\eta \rangle_{\text{NSNS}} = -\frac{v_{p+1}}{2^6} \int_0^\infty dl \frac{dl}{l^{\frac{9-p}{2}}} \frac{f_4(2il)^8}{f_1(2il)^8}. \tag{B.33}$$

The contribution from the massless states can be extracted from the constant term in the expansion

$$\frac{f_4(2il)^8}{f_1(2il)^8} = \frac{1}{q} - 8 + \mathcal{O}(q^1), \tag{B.34}$$

where now $q = e^{-2\pi l}$. The minus sign cancels with the overall sign in (B.33), yielding a positive contribution and hence an attractive force.

Recall that for Type 0 strings we gauge only a diagonal spin structure $(-1)^{f_L + f_R}$, and hence we keep only states invariant under projection by

$$P_{\text{NSNS}}^0 = \frac{1}{2}\left(1 + (-1)^{f_L + f_R}\right), \qquad P_{\text{RR}}^0 = \frac{1}{2}\left(1 \pm (-1)^{f_L + f_R}\right), \tag{B.35}$$

with the two choices of sign corresponding to Type 0B (+) and Type 0A (−). Using (B.8), we see that (B.32) are invariant for $p$ odd in Type 0B and $p$ even in Type 0A.

In contrast to (B.27) then, for each such $p$ there are now two boundary states for the Type 0 strings [87], which we will call D$p$ and D$p'$ for $|Dp, +\rangle$ and $|Dp, -\rangle$ respectively. Note that D$p'$-branes are *not* anti D$p$-branes.

Finally, note that the normalizations of Type 0 and Type II branes differ by a factor of $\sqrt{2}$. On the other hand, the amplitude for exchanging closed strings in Type II receives an extra contribution corresponding to R strings in the loop channel. This implies that the tensions of the Type 0 branes are smaller than those of Type II, in particular $T_p^0 = T_p^{\text{II}}/\sqrt{2}$ [88,89]. Finally, as stated before, when there are multiple branes the boundary state acquires an extra group theory factor (B.29).

## B.4 O-plane boundary states

### B.4.1 Crosscap state normalization

In analogy to the discussion above, we can find the correct normalization of the crosscap states that correspond to O-planes by requiring that the tree-channel amplitude for exchanging a closed string between a D-brane and a crosscap state matches the loop-channel Möbius strip amplitude. We know that the crosscap states are related to the usual boundary state by a $\pi/2$ translation in imaginary time, so we normalize them as

$$|Cq,\eta\rangle = -\frac{n_{Cq}}{\mathcal{N}_{Bq}}\, i^{H_{cl}}|Bq,\eta\rangle\,, \tag{B.36}$$

where $n_{Cq}$ is the normalization relative to the usual boundary state, and the minus sign is required to get negative tension. Then, the relations we must impose are

$$\frac{1}{\mathcal{N}_{Bp}}\int_0^\infty dl\ \left({}_{NSNS}\langle Cq,\eta|e^{-2\pi lH_{cl}}|Bp,\eta\rangle_{NSNS} - {}_{NSNS}\langle Bp,\eta|e^{-2\pi lH_{cl}}|Cq,-\eta\rangle_{NSNS}\right)$$

$$= \int_0^\infty \frac{dt}{2t}\, \mathrm{Tr}_{NS}\left[e^{-2\pi tH_{op}}\Omega\right]\,,$$

$$\frac{1}{\mathcal{N}_{Bp}}\int_0^\infty dl\ \left({}_{NSNS}\langle Bp,\eta|e^{-2\pi lH_{cl}}|Cq,\eta\rangle_{NSNS} - {}_{NSNS}\langle Cq,-\eta|e^{-2\pi lH_{cl}}|Bp,\eta\rangle_{NSNS}\right)$$

$$= \int_0^\infty \frac{dt}{2t}\, \mathrm{Tr}_{NS}\left[e^{-2\pi tH_{op}}(-1)^f\Omega\right]\,,$$

$$\frac{1}{\mathcal{N}_{Bp}}\int_0^\infty dl\ \ \left({}_{RR}\langle Bp,\eta|e^{-2\pi lH_{cl}}|Cq,\eta\rangle_{RR} - {}_{RR}\langle Cq,-\eta|e^{-2\pi lH_{cl}}|Bp,\eta\rangle_{RR}\right)$$

$$= \int_0^\infty \frac{dt}{2t}\, \mathrm{Tr}_{R}\left[e^{-2\pi tH_{op}}\Omega\right]\,,$$

$$\frac{1}{\mathcal{N}_{Bp}}\int_0^\infty dl\ \ \left({}_{RR}\langle Cq,\eta|e^{-2\pi lH_{cl}}|Bp,\eta\rangle_{RR} - {}_{RR}\langle Bp,\eta|e^{-2\pi lH_{cl}}|Cq,-\eta\rangle_{RR}\right)$$

$$= \int_0^\infty \frac{dt}{2t}\, \mathrm{Tr}_{R}\left[e^{-2\pi tH_{op}}(-1)^f\Omega\right]\,.$$

As before, we can fix the normalization using any of these relations by first writing the loop-channel Möbius strip amplitude

$$\int_0^\infty \frac{dt}{2t}\, \mathrm{Tr}_{NS}\left[e^{-2\pi tH_{op}}\Omega\right] = -v_{p+1}\int_0^\infty \frac{dt}{(2t)^{\frac{p+3}{2}}}\, e^{i\frac{\pi}{4}(q-p)}\left[\frac{f_3^8 f_4^{2(q-p)}}{f_1^{8-q+p}}\right]\left(it+\frac{1}{2}\right)\,, \tag{B.37}$$

and then calculating the corresponding tree-channel amplitude using the boundary states

$$\frac{1}{\mathcal{N}_{Bp}}\int_0^\infty dl\ \left({}_{NSNS}\langle Cq,\eta|e^{-2\pi lH_{cl}}|Bp,\eta\rangle_{NSNS} - {}_{NSNS}\langle Bp,\eta|e^{-2\pi lH_{cl}}|Cq,-\eta\rangle_{NSNS}\right)$$

$$= -\frac{n_{Cq}}{2^5}v_{p+1}\int_0^\infty \frac{dl}{l^{\frac{9-q}{2}}}\left[\left(\frac{f_3^8 f_4^{2(q-p)}}{f_1^{8-q+p}}\right)\left(2il-\frac{1}{2}\right) - \left(\frac{f_4^8 f_3^{2(q-p)}}{f_1^{8-q+p}}\right)\left(2il+\frac{1}{2}\right)\right]$$

$$= \frac{n_{Cq}}{2^4}v_{p+1}\int_0^\infty \frac{dl}{l^{\frac{9-q}{2}}}\left[\frac{f_4^8 f_3^{2(q-p)}}{f_1^{8-q+p}}\right]\left(2il+\frac{1}{2}\right)\,, \tag{B.38}$$

where in the second equality we used the modular $T$ transformations in (B.13). Next, we translate the tree-channel amplitude to the loop channel using the transformation $l = 1/8t$ and the modular $P$ transformations in (B.14) to get

$$\frac{n_{\text{O}q}}{2^4} v_{p+1} \int_0^\infty \frac{dt}{8t^2} \frac{1}{(8t)^{\frac{q-9}{2}}} \left[ \frac{-f_3^8 e^{i\frac{\pi}{4}(q-p)} f_4^{2(q-p)}}{(2t)^{\frac{8-q+p}{2}} f_1^{8-q+p}} \right] \left( it + \frac{1}{2} \right)$$

$$= -\frac{n_{\text{O}q}}{2^{q-4}} v_{p+1} \int_0^\infty \frac{dt}{(2t)^{\frac{p+3}{2}}} e^{i\frac{\pi}{4}(q-p)} \left[ \frac{f_3^8 f_4^{2(q-p)}}{f_1^{8-q+p}} \right] \left( it + \frac{1}{2} \right). \tag{B.39}$$

Comparing with the previous result, we find that the normalization of the $|\text{C}q, \eta\rangle$ crosscap state relative to the boundary state is

$$n_{\text{C}q} = 2^{q-4}. \tag{B.40}$$

### B.4.2 Pin⁻ Type 0

Finally, we must assemble the crosscap states into physical orientifold plane states. For the Pin⁻ theories, the Pin⁻ structure on the worldsheet requires the boundary of the Möbius strip to have NS boundary conditions. Thus we expect the O$q$-plane state to be purely in the NSNS sector. In addition, we know that the open strings on the orientifold are in the NS sector, so the O-plane state must give the following loop channel result

$$\int_0^\infty dl \left( \langle \text{D}p, \eta | e^{-2\pi l H_{\text{cl}}} | \text{O}q \rangle + \langle \text{O}q | e^{-2\pi l H_{\text{cl}}} | \text{D}p, \eta \rangle \right)$$

$$= \int_0^\infty \frac{dt}{2t} \text{Tr}_{\text{NS}} \left[ e^{-2\pi t H_{\text{op}}} \frac{1}{2}(1 + (-1)^{\text{f}})\Omega \right]. \tag{B.41}$$

Thus the physical orientifold boundary state is

$$|\text{O}q\rangle = \frac{1}{\sqrt{2}} \left( |\text{C}q, +\rangle_{\text{NSNS}} - |\text{C}q, -\rangle_{\text{NSNS}} \right). \tag{B.42}$$

Importantly, the crosscap state carries crucial information about the presence of $n$ copies of ABK. To see this, it is easiest to consider the Klein bottle amplitude. Requiring that the tree-channel amplitudes for exchanging closed strings between two crosscaps match the loop-channel Klein bottle amplitudes gives for example

$$\int_0^\infty dl \,_{\text{NSNS}}\langle \text{C}p, -\eta | e^{-2\pi l H_{\text{cl}}} | \text{C}p, \eta \rangle_{\text{NSNS}} = \int_0^\infty \frac{dt}{2t} \text{Tr}_{\text{RR}} \left[ e^{-2\pi t H_{\text{cl}}} \Omega \right], \tag{B.43}$$

$$\int_0^\infty dl \,_{\text{NSNS}}\langle \text{C}p, \eta | e^{-2\pi l H_{\text{cl}}} | \text{C}p, -\eta \rangle_{\text{NSNS}} = \int_0^\infty \frac{dt}{2t} \text{Tr}_{\text{RR}} \left[ e^{-2\pi t H_{\text{cl}}} (-1)^{\text{f}}\Omega \right]. \tag{B.44}$$

If we use these to calculate the normalization of the crosscap state as was done for the Möbius strip, then for $n = 0 \mod 8$ both (B.43) and (B.44) yield the same result (B.40).

For generic $n$, however, (B.43) and (B.44) are unequal complex conjugates and the result in (B.40) needs to be modified. This may be seen seen as follows. In the presence of $n$ copies of ABK, we know that the action of $\Omega$ on the closed string RR Hilbert space is modified by a factor of $i^n$ when the Pin⁻ structure is $q(a, b) = (2, 1)$ or $(2, 3)$, see (3.18). These Pin⁻ structures are exactly the ones captured by the right-hand sides of (B.43) and (B.44), and hence for non-zero $n$ the left-hand side must change by $i^n$. In other words, we should redefine $|\text{C}p, \eta\rangle$ by a phase

$e^{i\theta(n,\eta)}$ such that $e^{-i\theta(n,-\eta)}e^{i\theta(n,\eta)} = i^n$, a solution of which is $\theta(n,\eta) = \frac{\pi}{4}\eta n \mod 2\pi$. The correct crosscap states for the theory with $n \neq 0 \mod 8$ can then be taken to be

$$|Cq, \eta\rangle = -\frac{2^{q-4}}{\mathcal{N}_{Bq}} e^{\frac{i\pi n\eta}{4}} i^{H_{cl}} |Bq, \eta\rangle. \tag{B.45}$$

This is what we must insert into (B.42) to obtain the physical orientifold plane state.

### B.4.3  Pin$^+$ **Type 0**

Similarly, for unoriented Pin$^+$ Type 0 we know that the O$q$-plane state must give the following loop channel results

$$\int_0^\infty dl \left(\langle Dp, \eta|e^{-2\pi l H_{cl}}|Oq\rangle + \langle Oq|e^{-2\pi l H_{cl}}|Dp, \eta\rangle\right)$$
$$= \int_0^\infty \frac{dt}{2t} \operatorname{Tr}_R\left[e^{-2\pi t H_{op}}\frac{1}{2}(1 + (-1)^f)\Omega\right]. \tag{B.46}$$

The physical orientifold state is then found to be

$$|Oq\rangle = -\frac{1}{\sqrt{2}}\left(|Cq, +\rangle_{RR} + |Cq, -\rangle_{RR}\right). \tag{B.47}$$

The fact that this contains only RR sector contributions is the boundary state formulation of the fact that fermions on the boundary of the Pin$^+$ Möbius strip are automatically in the R sector.

### B.4.4  **Type I**

For completeness, we finally describe the physical orientifold plane states for Type I. These are obtained by requiring

$$\int_0^\infty dl \left(\langle Dp|e^{-2\pi l H_{cl}}|Oq\rangle + \langle Oq|e^{-2\pi l H_{cl}}|Dp\rangle\right) \tag{B.48}$$
$$= \int_0^\infty \frac{dt}{2t}\left(\operatorname{Tr}_{NS}\left[e^{-2\pi t H_{op}}\frac{1}{2}(1 + (-1)^f)\Omega\right] - \operatorname{Tr}_R\left[e^{-2\pi t H_{op}}\frac{1}{2}(1 + (-1)^f)\Omega\right]\right).$$

The correct combination is found to be

$$|Oq\rangle = \frac{1}{2}\left(|Cq, +1\rangle_{NSNS} - |Cq, -1\rangle_{NSNS} + |Cq, +1\rangle_{RR} + |Cq, -1\rangle_{RR}\right).$$

The fact that this contains both NSNS and RR contributions means that DPin structure on the worldsheet must allow the boundary of the Möbius strip to be in the NS or R sectors, and thus must contain both Pin$^\pm$ as subgroups.

As an aside, let us note that the normalization of the O9 state relative to a Type I D9 state has an extra factor of 32, as expected by the usual Type I tadpole cancellation.

## C  Tadpole Cancellation

In this appendix we discuss the issue of tadpole cancellation in the unoriented Type 0 theories.

We begin by considering the Pin$^-$ Type 0 theory with $n$ copies of ABK. Before beginning any calculations it is important to recall that in this case the orientifold state corresponding to the



Figure 4: The tadpole cancellation condition. The Möbius strip is represented by a cylinder with one crosscap (the "X" at the end) and we must include separate contributions from crosscaps on the left and right ends. The Klein bottle is represented by a cylinder with two crosscaps.

O9-plane does not have an RR contribution; see (B.42). This means that the orientifold does not carry RR charge, and hence we will only be encountering NSNS tadpoles. Such tadpoles are not fatal since they can be cancelled by the Fischler-Susskind mechanism [90, 91], but this introduces a spacetime dependent coupling. We thus ask in which cases these NSNS tadpoles can be cancelled without resorting to this mechanism.

The goal is to calculate the cylinder, Möbius strip, and Klein bottle amplitudes and check that the tadpole contributions cancel amongst them (Fig. 4). Furthermore, since the putative tadpoles are in the closed string sector we must focus on the amplitudes in the tree-channel. For the moment we will focus on the cases with $n$ even, which are orientifolds of Type 0B.

First we will consider the cylinder amplitude. We recall that in Type 0B we have two different kinds of nine-branes, with corresponding boundary states

$$|D9, \eta\rangle = \frac{G}{\mathcal{N}_{B9}} \frac{1}{\sqrt{2}} (\eta\,|B9, \eta\rangle_{NSNS} + |B9, \eta\rangle_{RR}) , \tag{C.1}$$

where $G$ is a group theory factor which equals $G = N$ for unitary gauge group and $G = 2N$ for orthogonal or symplectic gauge group; see (B.29). The corresponding antibranes are

$$|\overline{D9}, \eta\rangle = \frac{G}{\mathcal{N}_{B9}} \frac{1}{\sqrt{2}} (\eta\,|B9, \eta\rangle_{NSNS} - |B9, \eta\rangle_{RR}) . \tag{C.2}$$

In order to avoid introducing RR tadpoles, we must only introduce brane-antibrane pairs, with boundary state

$$|D\overline{D}9, \eta\rangle = |D9, \eta\rangle + |\overline{D9}, \eta\rangle = \frac{G}{\mathcal{N}_{B9}} \sqrt{2}\, \eta\,|B9, \eta\rangle_{NSNS} . \tag{C.3}$$

In terms of such boundary states the cylinder amplitude is given by

$$\mathcal{A}_{C_2} = \int_0^\infty dl\, \langle D\overline{D}9, \eta | e^{-2\pi l H_{cl}} | D\overline{D}9, \eta\rangle , \tag{C.4}$$

which can be evaluated using the results collected in Appendix B to give

$$\mathcal{A}_{C_2} = \frac{G^2}{16} v_{10} \int_0^\infty dl\, \frac{f_3^8(2il)}{f_1^8(2il)} . \tag{C.5}$$

Worldsheet parity does not affect this amplitude so the result does not depend on $n$. We can easily extract the massless NSNS tadpole contribution using the $q$-expansions in (B.11), giving

$$\mathcal{A}_{C_2}\big|_{\text{tadpole}} = \frac{G^2}{2} v_{10} \int_0^\infty dl . \tag{C.6}$$

Next we calculate the Möbius strip amplitude. The O9-plane state was given in (B.42); crucially, it was argued to be $n$-dependent. Using the result obtained there, the Möbius strip amplitude can be evaluated to give

$$\mathcal{A}_{M_2} = \int_0^\infty dl \left( \langle D\overline{D}9, \eta | e^{-2\pi l H_{cl}} | O9 \rangle + \langle O9 | e^{-2\pi l H_{cl}} | D\overline{D}9, \eta \rangle \right)$$

$$= -2 \, G \, v_{10} \int_0^\infty dl \left[ \frac{e^{i\eta \frac{\pi}{4} n} f_3^8 - e^{-i\eta \frac{\pi}{4} n} f_4^8}{f_1^8} \right] \left( 2il + \frac{1}{2} \right), \qquad \text{(C.7)}$$

with the tadpole being

$$\mathcal{A}_{M_2} \big|_{\text{tadpole}} = -2^5 \, G \cos\left[ \frac{\pi \eta n}{4} \right] v_{10} \int_0^\infty dl \,. \qquad \text{(C.8)}$$

Finally, the Klein bottle amplitude amplitude is

$$\mathcal{A}_{K_2} = \int_0^\infty dl \, \langle O9 | e^{-2\pi l H_{cl}} | O9 \rangle$$

$$= 16 \, v_{10} \int_0^\infty dl \left[ \frac{2 f_3^8 - \left( e^{-i\eta \frac{\pi}{2} n} + e^{i\eta \frac{\pi}{2} n} \right) f_4^8}{f_1^8} \right] (2il), \qquad \text{(C.9)}$$

with tadpole

$$\mathcal{A}_{K_2} \big|_{\text{tadpole}} = 16^2 \left( 1 + \cos\left[ \frac{\pi \eta n}{2} \right] \right) v_{10} \int_0^\infty dl$$

$$= 2^9 \cos\left[ \frac{\pi \eta n}{4} \right]^2 v_{10} \int_0^\infty dl \,. \qquad \text{(C.10)}$$

Putting all the contributions together we find that the total tadpole is

$$(\mathcal{A}_{C_2} + \mathcal{A}_{M_2} + \mathcal{A}_{K_2}) \big|_{\text{tadpole}} = \frac{1}{2} \left( G - 32 \cos\left[ \frac{\pi \eta n}{4} \right] \right)^2 v_{10} \int_0^\infty dl \,. \qquad \text{(C.11)}$$

We may now read off the tadpole cancellation conditions. For $n = 0$ one can cancel the NSNS tadpole by adding sixteen 9-$\overline{9}$ pairs. If we choose these to consist of $m$ D9-$\overline{D}9$ pairs and $16 - m$ D9$'$-$\overline{D}9'$ pairs, the resulting gauge group is $[SO(2m) \times SO(32 - 2m)]^2$. The cases with $m = 0, 16$ are purely bosonic and have gauge group $SO(32) \times SO(32)$. For $n = 4$, we have the symplectic version of $n = 0$ and the tadpole cannot be cancelled. For $n = 2, 6$ we have zero tadpole contribution, and would seemingly not require addition of any nine-branes.

Next we discuss cases with $n$ odd, which are orientifolds of Type 0A. In Type 0A there do not exist any stable 9-branes, but there are unstable ones. These unstable branes do not couple to RR fields, and are purely in the NSNS sector. Hence the corresponding states may be written as

$$|\widetilde{D9}, \eta \rangle = \frac{G}{\mathcal{N}_{B9}} \eta | B9, \eta \rangle_{\text{NSNS}}, \qquad \text{(C.12)}$$

with $\mathcal{N}_{B9}$ the usual normalization factor accompanying $|B9, \eta \rangle$ and $G$ the corresponding group theory factor. This result differs from (C.3) only by a factor of $\sqrt{2}$. Then by a similar calculation as above we conclude that tadpole cancellation requires $G = 32\sqrt{2} \cos\left[ \frac{\pi \eta n}{4} \right]$. The case of $n = 1$ allows the tadpole to be cancelled by the addition of sixteen 9-branes. If we choose these to consist of $m$ D9-branes and $16 - m$ D9$'$-branes, the resulting gauge group is

$SO(2m) \times SO(32 - 2m)$. Similar statements hold for $n = 7$. The $n = 5, 3$ cases are the corresponding symplectic cases, for which the NSNS tadpole cannot be cancelled by the addition of branes.

Finally, we discuss the issue of tadpole cancellation for Pin$^+$ strings. In contrast to the Pin$^-$ theories studied above, for these theories the orientifold only has contributions from the RR sector; see (B.47). Hence one has an RR tadpole which must be cancelled. A calculation analogous to the one above shows that the tadpole can be cancelled by adding 32 D9 and 32 D9′-branes, giving total gauge group $U(32)$. Though this introduces NSNS tadpoles [56, 60], these do not render the theory inconsistent and can be removed via the Fischler-Susskind mechanism [90, 91].

Note that it makes sense to talk about tadpoles in the Pin$^+$ theories despite $\mathbb{RP}^2$ not admitting a Pin$^+$ structure. The reason for this is that the tadpole is given by a one-point function on $\mathbb{RP}^2$, which corresponds to a punctured $\mathbb{RP}^2$. The latter manifold is conformally equivalent to the Möbius strip, which does in fact admit a Pin$^+$ structure.

As a final note, whenever tadpole cancellation requires the addition of D9-branes, the question of stability of D$p$-branes must be revisited to account for the possibility of tachyonic modes of the strings stretched between the D$p$- and D9-branes. In this case, the K-theory classification outlined in Sections 4 and 5 will be modified, and branes which were previously stable may become unstable.

# D   Arf **and** ABK **from index theory**

In this appendix we rephrase many of the results on the Arf and ABK invariants given in Section 2 in terms of index theory. The majority of this appendix is due to E. Witten [42]. The authors thank him for very kindly allowing them to reproduce the content here. Four-dimensional analogs of many of these results can be found in Appendix C of [45].

## D.1   $\eta$-invariants: generalities

Our normalization of the eta invariant is

$$\eta(\Sigma, \sigma) = \sum_E \text{sgn}(E), \tag{D.1}$$

where $E$ are the eigenvalues of the Dirac operator on $\Sigma$ with spin structure $\sigma$, and the sum is to be appropriately regularized. We work in the convention that $\text{sgn}(0) = 1$, so that $\eta(\Sigma, \sigma)$ also counts zero-modes. Often we will omit $\Sigma$ from the argument of $\eta(\Sigma, \sigma)$.

Because the Arf and ABK invariants can be expressed as ratios of massive fermion path integrals as in (2.3) and (2.20), they are examples of $\eta$-invariants. For example, for the Arf invariant we have

$$(-1)^{\text{Arf}(\Sigma, \sigma)} = \frac{Z_{\text{ferm}}(m \gg 0)}{Z_{\text{ferm}}(m \ll 0)} = \prod_E \frac{iE + m}{iE - m} = e^{i\frac{\pi}{2}\eta(\sigma)}. \tag{D.2}$$

An analogous result holds for the ABK invariant.

The $\eta$-invariant is not necessarily a bordism invariant, but in the case of two-dimensional theories the $\eta$-invariant modulo some integer is. This can be seen by appealing to the APS index theorem, which states that the index of the Dirac operator on a manifold $Y_{d+1}$ with boundary $\partial Y_{d+1} = X_d$ is given in terms of the $\eta$-invariant as[13]

$$\text{ind } iD_{Y_{d+1}} = -\frac{1}{2}\eta(X_d, \sigma) + \int_{Y_{d+1}} \hat{A}(R)\,\text{ch}(F)\,. \tag{D.3}$$

---

[13]In the original notation of APS [92], what we are calling $\eta$ is instead called $2\xi$.

Note that when $d$ is even, the local term on the right-hand side vanishes, and as a result the $\eta$-invariant can be a bordism invariant. Combined with the fact that the left-hand side is an integer, we see that the $\eta$-invariant modulo 2 is a bordism invariant. This can be refined further.

Assume that the fermion system whose Dirac operator is used in the definition of the $\eta$-invariant admits a mass term. This provides an invariant anti-symmetric bilinear form on the eigenfunctions, and therefore introduce a quaternionic structure. Therefore the index is in fact an even number, and $\eta$ modulo 4 is a bordism invariant.

Let us now consider a spin 2-manifold. Then there exists a globally well-defined chirality matrix $\Gamma$ satisfying $\Gamma^2 = 1$ and $\{iD, \Gamma\} = 0$, and hence for any state of non-zero eigenvalue $E$ there is also a state with eigenvalue $-E$. Then the contributions to the $\eta$-invariant from nonzero eigenvalues simply cancel out. Denoting the number of positive chirality zero-modes with spin structure $\sigma$ by $\zeta(\sigma)$, we have

$$\eta(\sigma) = 2\zeta(\sigma) \mod 4, \tag{D.4}$$

where the factor of 2 arises because $\eta(\sigma)$ counts both chiralities. This means that

$$(-1)^{\mathrm{Arf}(\Sigma, \sigma)} = e^{i\frac{\pi}{2}\eta(\sigma)} = (-1)^{\zeta(\sigma)} \tag{D.5}$$

generates at most a $\mathbb{Z}_2$, as expected by our previous definitions of the Arf invariant.

We next consider the Pin$^-$ case. As argued above, the $\eta$-invariant is a mod 4 bordism invariant. Let us now show that $\eta$ takes half-integer values, and thus provides us with a mod 8 invariant. The half-integrality is proven as follows. Given a Pin$^-$ structure $\sigma \in H^2(\Sigma, \mathbb{Z}_2)$, there exists a "complementary" Pin$^-$ structure $\sigma' := \sigma + w_1$ which is obtained by twisting by the orientation bundle. Then note that[14]

$$\eta(\sigma) + \eta(\sigma') = 0 \mod 4 . \tag{D.6}$$

We also have generally that

$$4\eta(\sigma + a) - 4\eta(\sigma) = 0 \mod 4 \tag{D.7}$$

for any $a \in H^1(\Sigma, \mathbb{Z}_2)$.[15] In the case that $a = w_1$ we have $\sigma + a = \sigma'$, and thus combining (D.6) and (D.7) we conclude that $\eta(\sigma)$ is generically half-integral. As a result, we have that $e^{i\pi\mathrm{ABK}(\Sigma, \sigma)} = e^{i\frac{\pi}{2}\eta(\Sigma, \sigma)}$ generates at most $\mathbb{Z}_8$, as expected.

## D.2 $\eta$-invariants: examples

We now offer some explicit calculations of the Arf and ABK invariants in terms of their definitions in this appendix.

$T^2$: A trivial example is that of the Arf invariant on $T^2$ with spin structure $\sigma$. In that case we know that for the NSNS, RNS, and NSR spin structures we have $\zeta(\sigma) = 0$, whereas for RR we have $\zeta(\sigma) = 1$. This together with (D.5) then reproduces the results of (2.10).

---

[14]This equality is true because the left-hand side is the $\eta$-invariant of the spin structure on the oriented double cover $\hat{\Sigma}$. Note that $\hat{\Sigma}$ is the boundary of the total space $X$ of the unit disk bundle of the orientation line bundle of $\Sigma$. That $\Sigma$ is Pin$^-$ is equivalent to $X$ being spin. These facts together imply that $\hat{\Sigma}$ is null-bordant, and so the right-hand side is 0 modulo 4.

[15]To see this, note that $4\eta(\sigma)$ is the $\eta$-invariant of the Dirac operator with Pin$^-$ structure $\sigma$ acting on a rank 4 trivial real vector bundle $V$, whereas $4\eta(\sigma + a)$ is the $\eta$-invariant of the Dirac operator with Pin$^-$ structure $\sigma$ acting on a rank 4 real vector bundle $V' = A^{\oplus 4}$, where $A$ has the property that $w_1(A) = a$. Because the Stiefel-Whitney classes of $V'$ all vanish, $V'$ is trivial and has the same mod 4 $\eta$-invariant as $V$, thereby giving (D.7).

$\mathbb{RP}^2$: A less trivial result is to reproduce the values of ABK on $\mathbb{RP}^2$. We compute it in two ways. The first is to consider an orbifold of the three-torus $T^3/\mathbb{Z}_2$ where $\mathbb{Z}_2$ acts as $x_i \to -x_i$ for $i = 1, 2, 3$. The resulting space has eight fixed points at $x_i \in \frac{1}{2}\mathbb{Z}$. We may remove a small ball around each of these points to obtain a smooth manifold, with the boundary of this manifold being eight copies of $\mathbb{RP}^2$. Then by the APS index theorem (D.3) for $d = 2$ we conclude that $\eta(\mathbb{RP}^2) = -\frac{1}{4}\text{ind}iD$, with $iD$ the Dirac operator on the $T^3/\mathbb{Z}_2$ with points removed. Using conformal invariance, it is possible to argue that the index of the Dirac operator on this manifold is the same as that on the original $T^3/\mathbb{Z}_2$, so we need only compute this quantity. Let us define $\mathcal{H}_{\pm}$ to be the spaces of spinors on $T^3$ which satisfy $\psi(-x) = \pm\psi(-x)$. The Dirac operator maps $\mathcal{H}_{\pm} \to \mathcal{H}_{\mp}$, and the index of the Dirac operator on $T^3/\mathbb{Z}_2$ is then just defined to be the number of zero-modes in $\mathcal{H}_+$ minus those in $\mathcal{H}_-$. These numbers are easily obtained: depending on the Pin$^-$ structure, the zero-modes are the 2-dimensional space of constant spinors in either $\mathcal{H}_+$ or $\mathcal{H}_-$, with no zero modes in the remaining space. Hence we have $\text{ind}iD = \pm 2$, and consequently $\eta(\mathbb{RP}^2) = \pm\frac{1}{2}$. We may finally calculate the ABK invariant to be $e^{i\pi\text{ABK}(\mathbb{RP}^2)} = e^{\pm i\frac{\pi}{4}}$, matching the previous results in (2.24).

The second derivation of this result is a direct computation from the spectrum of the Dirac operator. Instead of directly studying the Dirac equation on unoriented manifolds $\Sigma$ we will consider their orientable double covers $\hat{\Sigma}$. These are equipped with an orientation-reversing involution $\tau$ such that $\Sigma = \hat{\Sigma}/\tau$. We will make use of the following morphism

$$\text{Pin}^- \text{ structures on } \Sigma = \hat{\Sigma}/\tau \quad \longrightarrow \quad \tau\text{-invariant spin structures on } \hat{\Sigma}, \quad \text{(D.8)}$$

induced by the projection. This map is not injective, but rather two-to-one since given a Pin$^-$ structure $\sigma$, both $\sigma$ and its twist by the orientation bundle $\sigma'$ lift to the same spin structure on the orientable double cover. It is not surjective either, since there are spin structures on $\hat{\Sigma}$ which are not the lift of any Pin$^-$ structure. The $\tau$-invariance of the spin structures on $\hat{\Sigma}$ implies that $[\tau, iD] = 0$. Hence there is a basis of eigenspinors with a well-defined eigenvalue of $\tau$. The different eigenvalues of $\tau$ correspond to different Pin$^-$ structures $\sigma$ and $\sigma'$. In summary, we can extract the spectrum of the Dirac operator $iD$ on $\Sigma$ from that on the orientable double cover $\hat{\Sigma}$ by considering eigenspinors of $iD$ on the latter with a fixed eigenvalue of $\tau$.

Let us apply this strategy to $\mathbb{RP}^2$. Its orientable double cover is a two-sphere $S^2$, which has a single spin structure. The spectrum of the Dirac operator on the two-sphere is well known and is given by[16]

$$E = \pm(n+1) \quad \text{with multiplicity} \quad 2(n+1) \quad \text{and} \quad \tau = \mp(-1)^n, \quad \text{(D.9)}$$

with $n \geq 0$. For convenience we will regularize the sum over eigenvalues (D.1) as follows

$$\eta = \lim_{\epsilon \to 0^+} \sum_E \text{sgn}(E)\, e^{-\epsilon|E|}. \quad \text{(D.10)}$$

From this information we can readily calculate the $\eta$-invariant of $\mathbb{RP}^2$ for either Pin$^-$ structure,

$$\tau = \pm 1: \quad \eta = \lim_{\epsilon \to 0^+}\left(\mp \sum_{n \in 2\mathbb{N}} 2(n+1)\,e^{-\epsilon(n+1)} \pm \sum_{n \in 2\mathbb{N}+1} 2(n+1)\,e^{-\epsilon(n+1)}\right) = \mp\frac{1}{2} \quad \text{(D.11)}$$

reproducing our previous result.

---

[16]The eigenspace decomposition is simply the spinor spherical harmonics. One way to quickly derive the eigenvalues is to use the operator-state correspondence of a free massless Dirac fermion in dimension $d + 1$. There, the (absolute value of the) eigenvalues of the Dirac operator on $S^d$ are the dilatation eigenvalues of the single-particle operators of the form $\partial \cdots \partial \psi$, which are therefore given by $n + d/2$.

**The Klein bottle:** We now obtain the values for the $\eta$-invariant on the Klein bottle $K_2$. A triv-ial way to do so is to note that the $\eta$-invariant factorizes under connected sums, $\eta(\Sigma_1 \# \Sigma_2) = \eta(\Sigma_1) + \eta(\Sigma_2)$. Then recalling that $K_2 = \mathbb{RP}^2 \# \mathbb{RP}^2$, our previous results im-ply that $\eta(K_2) = 0, 0, \pm 1$ depending on the choice of $\text{Pin}^-$ structure. This reproduces the results of (2.25) for the ABK invariant.

A more fulfilling derivation of this result is to again consider the explicit Dirac spectrum. The orientable double cover in this case is the torus $T^2$, which we take to be rectangular with side lengths 1 and 2. That is, $T^2 = \mathbb{R}^2/\Gamma$ for the lattice $\Gamma = \mathbb{Z} \oplus 2\mathbb{Z}$. Taking $x^i = (x, y)$ to be the coordinates on the torus, we have $(x, y) = (x+1, y) = (x+1, y+2)$. The orientation-reversing involution is $\tau(x, y) = (-x, y+1)$. As was discussed in Section 2.2.4, of the four torus spin structures only those which are periodic in the $y$-direction descend in the quotient.

We first consider the spin structure periodic in $x$. We begin by finding the eigenspinors of the square of the Dirac operator, which is just the Laplacian, $(iD)^2 = -\Delta$. These can be easily constructed as

$$u_p(x^i) = f_p(x^i)\Psi, \quad f_p(x^i) = e^{2\pi i x^i p_i} \quad \text{with} \quad p_i \in \Gamma^* = \mathbb{Z} \oplus \frac{1}{2}\mathbb{Z}, \qquad (\text{D.12})$$

where $f_p(x^i)$ are the eigenfunctions of the Laplacian, with momenta taking values in the dual lattice $\Gamma^*$, and $\Psi$ a covariantly constant spinor. We can also construct eigenfunctions for the spin structure antiperiodic in $x$ by letting the momenta take values in $\tilde{\Gamma}^* = (\mathbb{Z} + \frac{1}{2}) \oplus \frac{1}{2}\mathbb{Z}$. In both cases it is easy to check that

$$(iD)^2 u(x^i) = 4\pi^2 p^2 u(x^i) . \qquad (\text{D.13})$$

In terms of the $u(x^i)$ we can construct the eigenspinors of the Dirac operator as

$$v^\pm(x^i) = \pm 2\pi|p| u(x^i) + iD u(x^i) , \quad \text{with} \quad iD v^\pm(x^i) = \pm 2\pi|p| v^\pm(x^i) . \qquad (\text{D.14})$$

This spectrum is clearly symmetric, and hence if there are no zero-modes the $\eta$-invariant van-ishes. The only case in which there are zero-modes is the case of periodic spin structure in both directions, and then the multiplicity of the zero-mode is two so that $\eta(T^2, \sigma_{\text{RR}}) = 2$, as we know.

To get the corresponding results for the Klein bottle, we now keep the portion of the spec-trum with fixed $\tau$ eigenvalue. To do so, it is useful to choose an explicit representation for the gamma-matrices, say as $\gamma_1 = \sigma_3$ and $\gamma_2 = \sigma_1$. Then in addition to acting on $(x, y)$ in the manner shown above, the involution $\tau$ acts as $\sigma_1$ on spinors. With this, it is easy to show that the eigenspinors with fixed eigenvalue under $\tau$ have $p_1 = 0$, and hence require periodic spin structure in the $x$-direction. Defining $n := 2p_2$, the remaining spectrum is

$$E = \pm \pi |n| \quad \text{with multiplicity} \quad 1 \quad \text{and} \quad \tau = \mp(-1)^n , \qquad (\text{D.15})$$

with the zero eigenvalue having multiplicity 2.

Two of the $\text{Pin}^-$ structures of $K_2$ lift to antiperiodic spin structure in the $x$-direction, and consequently have vanishing $\eta$-invariant. The remaining two $\text{Pin}^-$ structures lift to periodic spin structure in the $x$-direction, and correspond to the two different eigenvalues for $\tau$. The resulting $\eta$-invariants are

$$\tau = \pm 1: \qquad \eta(K_2) = \lim_{\epsilon \to 0^+} 2\left(\mp \sum_{n \in 2\mathbb{N}} e^{-\epsilon \pi |n|} \pm \sum_{n \in 2\mathbb{N}+1} e^{-\epsilon \pi |n|}\right) = \mp 1 , \qquad (\text{D.16})$$

reproducing earlier results.

### D.3  Quadratic forms and enhancements

Let us now make contact between index theory and the combinatoric definitions of Arf and ABK given in (2.7) and (2.21). In order to do so, we first rewrite the quadratic form $\tilde{q}(a)$ and enhancement $q(a)$ in terms of indices.

We start with the oriented case. We consider

$$\tilde{q}(a) := \zeta(\sigma + a) - \zeta(\sigma) \quad \text{mod } 2 \tag{D.17}$$

for a given spin structure $\sigma$, where $\zeta(\sigma)$ is the number of zero modes of the positive-chirality Dirac operator. We now verify that this is the quadratic refinement of the intersection form, i.e. the relation (2.6) is satisfied. To do so, we must check that

$$\zeta(\sigma + a + b) + \zeta(\sigma + a) + \zeta(\sigma + b) + \zeta(\sigma) = \int a \cup b \quad \text{mod } 2 \tag{D.18}$$

holds. We note that the left-hand side is $\zeta(V)$, the mod 2 index with spin structure $\sigma$ for the Dirac operator acting on a positive chirality spinor valued in a rank 4 real vector bundle $V = \epsilon + A + B + AB$, where $\epsilon$ is a trivial real line bundle and we have $w_1(A) = a$, $w_1(B) = b$, and $w_1(AB) = a + b$. From this definition, it also follows that $w_1(V) = 0$ and $w_2(V) = a \cup b$. Therefore, $V$ is topologically equivalent to $H \oplus L$, the direct sum of a rank 2 real trivial bundle $H$ and a complex line bundle $L$ with $c_1(L) = w_2(V) \mod 2$. This is because real vector bundles on a Riemann surface are classified topologically by their rank and Stiefel-Whitney classes. Clearly $\zeta(H) = 0$ modulo 0, so we have $\zeta(V) = \zeta(L)$. Under the $U(1)$ rotating $L$, the zero-modes of $L$ have charge $\pm 1$, with respective numbers $n_\pm$. We then have $\zeta(L) = n_+ + n_-$ mod 2. By complex conjugation, we can replace a charge $-1$ mode of positive chirality with a charge $+1$ mode of negative chirality. Let $m_\pm$ denote the number of positive/negative chirality modes of charge $+1$. Then we have $n_\pm = m_\pm$, and hence $\zeta(L) = m_+ - m_- \mod 2$. The right-hand side is now the index of the Dirac operator acting on $L$, which by the index theorem is $\int c_1(L) = \int w_2(V) = \int a \cup b \mod 2$. We then conclude that $\zeta(V) = \int a \cup b \mod 2$, thereby confirming (D.18).

With the definition (D.17), it is now simply to check that our combinatorial definition (2.7) is consistent with the definition (D.5). We have

$$
\begin{aligned}
(-1)^{\text{Arf}(\Sigma,\sigma)} &= \frac{1}{\sqrt{|H^1(\Sigma,\mathbb{Z}_2)|}} \sum_{a \in H^1(\Sigma,\mathbb{Z}_2)} (-1)^{\zeta(\sigma+a)-\zeta(\sigma)} \\
&= (-1)^{\zeta(\sigma)} \left( \frac{1}{\sqrt{|H^1(\Sigma,\mathbb{Z}_2)|}} \sum_{a \in H^1(\Sigma,\mathbb{Z}_2)} (-1)^{\zeta(a)} \right).
\end{aligned} \tag{D.19}
$$

The term in parenthesis can be shown to square to 1 using steps analogous to those used for the combinatorial definition, and one can then fix the result to +1 by checking an explicit example.

We now move on to the Pin$^-$ case. In that case we define the quadratic enhancement $q(a)$ as

$$q(a) = \zeta(\sigma + a) - \zeta(\sigma) \quad \text{mod } 4, \tag{D.20}$$

where $\sigma$ is now a Pin$^-$ structure. We must check that (2.22) is satisfied by this definition. To do so, let us first prove this in the special case of $\Sigma = \mathbb{RP}^2$. There is then only one non-trivial possibility for $a$ and $b$, namely $w_1$. The identity (2.22) is trivially satisfied unless $a = b = w_1$, so we focus on that case. Then noting that $q(0) = 0$, the identity we wish to prove is

$$q(w_1) = \int w_1^2 = 1 \quad \text{mod } 2. \tag{D.21}$$

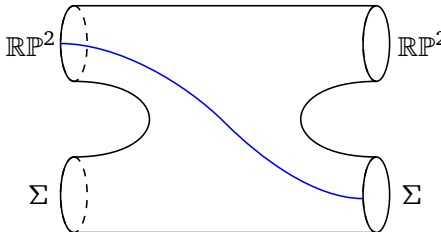

Figure 5: A bordism between $\mathbb{RP}^2 \times \Sigma$ and itself, with a real vector bundle $V'$ on it. The blue line represents the Poincaré dual of $w_2(V')$.

We now use the fact that $q(w_1) = \eta(\sigma') - \eta(\sigma) \mod 2$. As we showed in the previous subsection, for $\mathbb{RP}^2$ one of the two $\eta$-invariants is $+\frac{1}{2}$, while the other is $-\frac{1}{2}$. Either way, we conclude that $q(w_1) = 1 \mod 2$, thereby confirming the identity.

To prove the identity in generality, we now make use of bordism invariance. What we would like to prove is

$$\zeta(\sigma + a + b) - \zeta(\sigma + a) - \zeta(\sigma + b) + \zeta(\sigma) = 2 \int a \cup b \mod 4 . \tag{D.22}$$

Equivalently, this is

$$\eta(V') - \eta(V) = 2 \int w_2(V') \mod 4 , \tag{D.23}$$

where $V$ is a rank 8 trivial bundle and $V'$ is a rank 8 bundle with $w_1(V') = 0$ and $w_2(V') = a \cup b$. Recall that in two dimensions $w_2(V')$ is Poincaré dual to a point, while in three dimensions it is dual to a curve. Then consider a bordism from $\mathbb{RP}^2 \times \Sigma$ to itself by means of a connected sum, as shown in Fig. 5. In the figure, we have drawn a curve that starts at $\mathbb{RP}^2$ on the top left and goes down to $\Sigma$ on the bottom right. Consider a real vector bundle $V'$ such that $w_2(V')$ is Poincaré dual to this curve. This gives a bordism between $\mathbb{RP}^2 \times \Sigma$ with $w_2(V') = 1$ on $\mathbb{RP}^2$ and 0 on $\Sigma$, and $\mathbb{RP}^2 \times \Sigma$ with $w_2(V') = 0$ on $\mathbb{RP}^2$ and 1 on $\Sigma$. Because (D.23) is unchanged by this change in $V'$, it must hold for any $\Sigma$, thus proving the claim.

# E $\ \mho_{\mathrm{DPin}}^d(\mathrm{pt})$ via the Atiyah-Hirzebruch spectral sequence

In this appendix we analyze the Atiyah-Hirzebruch spectral sequence (AHSS) for $\mho_X^{d=2,3}(pt)$ for $X = \mathrm{Spin} \times \mathbb{Z}_2$, $\mathrm{Pin}^\pm$, and DPin. Except for the last case $X = \mathrm{DPin}$ the outcome is well-known; we include the computations here just to illustrate the method.

To write down the $E_2$ page, we will need the groups $H^p(X, \underline{\mho_{\mathrm{spin}}^q(pt)})$. More concretely, we need $H^*(B\mathbb{Z}_2 \times B\mathbb{Z}_2, \mathbb{Z}_2)$ and $H^*(B\mathbb{Z}_2 \times B\mathbb{Z}_2, \underline{U(1)})$, where the underline signifies that the first $\mathbb{Z}_2$ acts on $U(1)$ by complex conjugation and the second acts trivially. The first is standard: we have

$$H^*(B\mathbb{Z}_2 \times B\mathbb{Z}_2, \mathbb{Z}_2) = \mathbb{Z}_2[w, a] , \tag{E.1}$$

where $w$ and $a$ are the generators of $H^1(B\mathbb{Z}_2 \times B\mathbb{Z}_2, \mathbb{Z}_2) = H^1(B\mathbb{Z}_2, \mathbb{Z}_2) \oplus H^1(B\mathbb{Z}_2, \mathbb{Z}_2)$. As for $H^*(B\mathbb{Z}_2 \times B\mathbb{Z}_2, \underline{U(1)})$, they are determined as an abstract group in e.g. Appendix J.6 of [93]; in particular all elements are annihilated by 2. For our purposes we will need more detailed data. We note that the short exact sequence

$$0 \longrightarrow \mathbb{Z}_2 \overset{\iota}{\longrightarrow} \underline{U(1)} \overset{2\cdot}{\longrightarrow} \underline{U(1)} \to 0 \tag{E.2}$$

leads to the long exact sequence

$$\cdots \xrightarrow{2\cdot} H^{d-1}(B\mathbb{Z}_2 \times B\mathbb{Z}_2, \underline{U(1)}) \xrightarrow{\beta} H^d(B\mathbb{Z}_2 \times B\mathbb{Z}_2, \mathbb{Z}_2)$$

$$\xrightarrow{\iota} H^d(B\mathbb{Z}_2 \times B\mathbb{Z}_2, \underline{U(1)}) \xrightarrow{2\cdot} H^d(B\mathbb{Z}_2 \times B\mathbb{Z}_2, \underline{U(1)}) \xrightarrow{\beta} \cdots \quad \text{(E.3)}$$

Since 2· annihilates everything, we see that $H^d(B\mathbb{Z}_2 \times B\mathbb{Z}_2, \underline{U(1)})$ is a quotient of $H^d(B\mathbb{Z}_2 \times B\mathbb{Z}_2, \mathbb{Z}_2)$ by the image of the twisted Bockstein $\beta = \mathrm{Sq}^1 + w$. Therefore we find

$$H^0(B\mathbb{Z}_2 \times B\mathbb{Z}_2, \underline{U(1)}) = U(1) , \tag{E.4}$$

$$H^1(B\mathbb{Z}_2 \times B\mathbb{Z}_2, \underline{U(1)}) = \mathbb{Z}_2 = \frac{\langle w, a \rangle}{\langle w \rangle} , \tag{E.5}$$

$$H^2(B\mathbb{Z}_2 \times B\mathbb{Z}_2, \underline{U(1)}) = \mathbb{Z}_2^2 = \frac{\langle w^2, wa, a^2 \rangle}{\langle a(a+w) \rangle} , \tag{E.6}$$

$$H^3(B\mathbb{Z}_2 \times B\mathbb{Z}_2, \underline{U(1)}) = \mathbb{Z}_2^2 = \frac{\langle w^3, w^2 a, wa^2, a^3 \rangle}{\langle w^3, wa^2 \rangle} , \tag{E.7}$$

$$H^4(B\mathbb{Z}_2 \times B\mathbb{Z}_2, \underline{U(1)}) = \mathbb{Z}_2^3 = \frac{\langle w^4, w^3 a, w^2 a^2, wa^3, a^4 \rangle}{\langle w^2 a(a+w), a^3(a+w) \rangle} . \tag{E.8}$$

This data can be checked e.g. by noticing that in this low degree range $H^d(B\mathbb{Z}_2 \times BG, \underline{U(1)})$ with $T : \mathbb{Z}_2 \times G \to \mathbb{Z}_2$ given by $T = w$ equals $\mho^d_{\text{unoriented}}(BG)$. The generators of $\Omega_d^{\text{unoriented}}(B\mathbb{Z}_2)$ can be taken to be e.g. $S^1$ with nontrivial $\mathbb{Z}_2$ bundle for $d = 1$, $\mathbb{RP}^2$ with and without nontrivial $\mathbb{Z}_2$ bundle for $d = 2$, ($\mathbb{RP}^2$ with and without nontrivial $\mathbb{Z}_2$ bundle) $\times$ ($S^1$ with nontrivial $\mathbb{Z}_2$ bundle) for $d = 3$, and $\mathbb{RP}^4$ with and without nontrivial $\mathbb{Z}_2$ bundle, and $\mathbb{RP}^2 \times \mathbb{RP}^2$ with nontrivial $\mathbb{Z}_2$ on the first factor for $d = 4$. We can then evaluate all elements of $\mathbb{Z}_2[w, a]$ on the generators with the identification that $w$ is $w_1$ of the manifold and $a$ is $w_1$ of the $\mathbb{Z}_2$ bundle.

With this information, we can now proceed to the calculation of the relevant groups. Before computing $\mho^d_{\text{DPin}}(pt)$, we illustrate the technique in the known examples of $\mho^d_{\text{Spin}}(B\mathbb{Z}_2)$ and $\mho^d_{\text{Pin}^{\pm}}(pt)$. Below, the image of $\iota : \mathbb{Z}_2 \hookrightarrow \underline{U(1)}$ is denoted by prefixing by $\frac{1}{2}$, since $\{0, \frac{1}{2}\} \subset U(1)$.

$\underline{\mho_{\text{Spin}}(B\mathbb{Z}_2)}$

The $E_2$ page needed for obtaining $\mho_{\text{Spin}}(B\mathbb{Z}_2)$ is

| $q$ | | | | | |
|---|---|---|---|---|---|
| 3 | | | | | |
| 2 | $\mathbb{Z}_2$ | $\mathbb{Z}_2$ | $\mathbb{Z}_2$ | $\mathbb{Z}_2$ | $\mathbb{Z}_2$ |
| 1 | $\mathbb{Z}_2$ | $\mathbb{Z}_2$ | $\mathbb{Z}_2$ | $\mathbb{Z}_2$ | $\mathbb{Z}_2$ |
| 0 | $U(1)$ | $\frac{1}{2}\mathbb{Z}_2$ | | $\frac{1}{2}\mathbb{Z}_2$ | |
| | 0 | 1 | 2 | 3 | 4 | $p$ |

(E.9)

This can be found from the data given above by forgetting the pieces involving $w$. The differential $d_2$ starting from $E_2^{p,q}$ with $p + q \leq 4$ turns out to be zero. The $E_3$ page is then

| $q$ | | | | | | |
|---|---|---|---|---|---|---|
| 3 | | | | | | |
| 2 | $\mathbb{Z}_2$ | $\mathbb{Z}_2$ | ? | ? | ? | |
| 1 | $\mathbb{Z}_2$ | $\mathbb{Z}_2$ | $\mathbb{Z}_2$ | ? | ? | |
| 0 | $U(1)$ | $\frac{1}{2}\mathbb{Z}_2$ | | $\frac{1}{2}\mathbb{Z}_2$ | | |
| | 0 | 1 | 2 | 3 | 4 | $p$ |

(E.10)

The only possibly nontrivial $d_3$ is $d_3 : E_3^{0,2} \to E_3^{3,0}$ but a special property of untwisted bordism says that every $d_n$ starting from $E^{0,q}$ is zero. (This fact is explained below Theorem 9.10 of [82].) Then this is also the $E_4$ page, and $E^{p,q}$ with $p + q \leq 3$ cannot change any further.

From this we read off that $\mho_{\mathrm{Spin}}^d(pt)$ for $d = 1, 2, 3$ contains $4, 4$, and $8$ elements, respectively. This agrees with known results.

### $\mho_{\mathrm{Pin}^-}(pt)$

The $E_2$ page in this case is

| $q$ | | | | | | |
|---|---|---|---|---|---|---|
| 3 | | | | | | |
| 2 | $\mathbb{Z}_2$ | $w$ | $w^2$ | $w^3$ | $w^4$ | |
| 1 | $\mathbb{Z}_2$ | $w$ | $w^2$ | $w^3$ | $w^4$ | |
| 0 | $U(1)$ | | $\frac{1}{2}w^2$ | | $\frac{1}{2}w^4$ | |
| | 0 | 1 | 2 | 3 | 4 | $p$ |

(E.11)

This can be found from the data given above by forgetting the part involving $a$. For $d_2$ starting from $q = 2$, one has $d_2^2 = \mathrm{Sq}^2 + w_1(V)\,\mathrm{Sq}^1 + w_2(V) = \mathrm{Sq}^2 + w\,\mathrm{Sq}^1$. Then since $\mathrm{Sq}^2(w^2) = (\mathrm{Sq}^1 w)(\mathrm{Sq}^1 w) = w^4$ and $\mathrm{Sq}^1(w) = w^2$, we find

$$d_2^2(1) = d_2^2(w^3) = 0 , \qquad d_2^2(w) = w^3 , \qquad d_2^2(w^2) = w^4 . \tag{E.12}$$

On the other hand we have $d_2^1 = \frac{1}{2}\,\mathrm{Sq}^2$, and hence

$$d_2^1(1) = d_2^1(w) = d_2^1(w^3) = 0 , \qquad d_2^1(w^2) = \frac{1}{2}w^4 . \tag{E.13}$$

Then the $E_3$ page is

| $q$ | | | | | | |
|---|---|---|---|---|---|---|
| 3 | | | | | | |
| 2 | $\mathbb{Z}_2$ | | | ? | ? | |
| 1 | $\mathbb{Z}_2$ | $w$ | | | ? | |
| 0 | $U(1)$ | | $\frac{1}{2}w^2$ | | | |
| | 0 | 1 | 2 | 3 | 4 | $p$ |

(E.14)

This predicts $|\mho_{\mathrm{Pin}^-}^d(pt)| = 2, 8, 0$ for $d = 1, 2, 3$, in agreement with known results.

$\mho_{\mathrm{Pin}^+}(pt)$

The $E_2$ page in this case is

| $q$ | | | | | | |
|---|---|---|---|---|---|---|
| 3 | | | | | | |
| 2 | $\mathbb{Z}_2$ | $w$ | $w^2$ | $w^3$ | $w^4$ | |
| 1 | $\mathbb{Z}_2$ | $w$ | $w^2$ | $w^3$ | $w^4$ | |
| 0 | $U(1)$ | | $\frac{1}{2}w^2$ | | $\frac{1}{2}w^4$ | |
| | 0 | 1 | 2 | 3 | 4 | $p$ |

(E.15)

This is obtained from the previous data by setting $w = a$. We then have $d_2^2 = \mathrm{Sq}^2 + w_1\mathrm{Sq}^1 + w_2 = \mathrm{Sq}^2 + w\mathrm{Sq}^1 + w^2$, and so

$$d_2^2(1) = w^2 , \qquad d_2^2(w) = d_2^2(w^2) = 0 , \qquad d_2^2(w^3) = w^5 . \tag{E.16}$$

On the other hand $d_2^1 = \frac{1}{2}\mathrm{Sq}^2 + \frac{1}{2}w^2$ and hence

$$d_2^1(1) = \frac{1}{2}w^2 , \qquad d_2^1(w) = \frac{1}{2}w^3 , \qquad d_2^1(w^2) = 0 , \qquad d_2^1(w^3) = \frac{1}{2}w^5 . \tag{E.17}$$

Then the $E_3$ page is

| $q$ | | | | | | |
|---|---|---|---|---|---|---|
| 3 | | | | | | |
| 2 | | $w$ | $w^2$ | ? | ? | |
| 1 | | $w$ | | $w^3$ | ? | |
| 0 | $U(1)$ | | | | $\frac{1}{2}w^4$ | |
| | 0 | 1 | 2 | 3 | 4 | $p$ |

(E.18)

This predicts $|\mho_{\mathrm{Pin}^+}^d(pt)| = 0, 2, 2$ for $d = 1, 2, 3$, in agreement with known results.

$\mho_{\mathrm{DPin}}(pt)$

We finally arrive at the case of interest. The $E_2$ page is

| $q$ | | | | | | |
|---|---|---|---|---|---|---|
| 3 | | | | | | |
| 2 | $\mathbb{Z}_2$ | $w, a$ | $w^2, wa, a^2$ | $w^3, w^2a, wa^2, a^3$ | $w^4, \ldots$ | |
| 1 | $\mathbb{Z}_2$ | $w, a$ | $w^2, wa, a^2$ | $w^3, w^2a, wa^2, a^3$ | $w^4, \ldots$ | |
| 0 | $U(1)$ | $\frac{1}{2}a$ | $\frac{1}{2}w^2, \frac{1}{2}wa = \frac{1}{2}a^2$ | $\frac{1}{2}w^2a, \frac{1}{2}a^3$ | $\frac{1}{2}w^4, \frac{1}{2}w^3a = \frac{1}{2}w^2a^2, \frac{1}{2}wa^3 = \frac{1}{2}a^4$ | |
| | 0 | 1 | 2 | 3 | 4 | $q$ |

(E.19)

We have $d_2^2 = \mathrm{Sq}^2 + w\mathrm{Sq}^1 + wa$ and $d_2^1 = \frac{1}{2}\mathrm{Sq}^2 + \frac{1}{2}wa$. Then the $E_3$ page is

| $q$ | | | | | | |
|---|---|---|---|---|---|---|
| 3 | | | | | | |
| 2 | | $a$ | ? | ? | ? | |
| 1 | | $a$ | $a^2$ | ? | ? | |
| 0 | $U(1)$ | $\frac{1}{2}a$ | $\frac{1}{2}w^2$ | $\frac{1}{2}a^3$ | $\frac{1}{2}w^4 = \frac{1}{2}w^3a = \frac{1}{2}w^2a^2, \frac{1}{2}wa^3 = \frac{1}{2}a^4$ | |
| | 0 | 1 | 2 | 3 | 4 | $q$ |

(E.20)

At this point we see that there can be at most four elements in $\mho_{\text{DPin}}^2(pt)$ and eight elements in $\mho_{\text{DPin}}^3(pt)$. We already know a subgroup $\mathbb{Z}_2 \times \mathbb{Z}_2$ of $\mho_{\text{DPin}}^2(pt)$, generated by $(-1)^{\int w_1^2}$ and $(-1)^{\text{Arf}(\hat{\Sigma})}$, and thus we conclude that $\mho_{\text{DPin}}^2(pt) = (\mathbb{Z}_2)^2$. We also know that the anomaly of Majorana fermion on unoriented surfaces form $\mathbb{Z}_8$, so we conclude that $\mho_{\text{DPin}}^3(pt) = \mathbb{Z}_8$.

# F $\quad\mho_{\text{DPin}}^d(\text{pt})$ **via the Adams spectral sequence**

In this appendix[17], we compute $\mho_{\text{DPin}}^d(\text{pt})$ for $d \le 6$ a different way, using the Adams spectral sequence. Though computations with the Adams spectral sequence are often difficult, the problem simplifies greatly when computing twisted spin bordism groups $\Omega_d^X$, thanks to a technique that first appears in Davis' thesis [94] and builds on work of Stong [95] and Anderson-Brown-Peterson [96].

We highly recommend Beaudry and Campbell's paper [97] for a detailed introduction to this method of computation and the ingredients that go into it, as well as several worked examples. We assume familiarity with the definitions and notation they give.

**Theorem F.1.** *The low-degree dpin bordism groups are:* $\Omega_0^{\text{DPin}} \cong \mathbb{Z}/2$, $\Omega_1^{\text{DPin}} \cong \mathbb{Z}/2$, $\Omega_2^{\text{DPin}} \cong \mathbb{Z}/2 \oplus \mathbb{Z}/2$, $\Omega_3^{\text{DPin}} \cong \mathbb{Z}/8$, $\Omega_4^{\text{DPin}} \cong \mathbb{Z}/2 \oplus \mathbb{Z}/2$, $\Omega_5^{\text{DPin}} \cong 0$, *and* $\Omega_6^{\text{DPin}} \cong \mathbb{Z}/2 \oplus \mathbb{Z}/2$.

For any finite abelian group $A$, there is a (noncanonical) isomorphism $A \cong \text{Hom}(A, \text{U}(1))$, so this also computes $\mho_{\text{DPin}}^d(\text{pt})$ for $0 \le d \le 6$, and agrees with the calculations made in Appendix E. Recall from Sec. 6.2 that a dpin structure is equivalent to a choice of two real line bundles $L_1, L_2 \to M$ and a spin structure on

$$TM \oplus (L_1 \otimes L_2) \oplus (L_2)^{\oplus 3}. \tag{F.2}$$

One consequence is that if *MTDPin* denotes the Thom spectrum for dpin structures, so that $\pi_k(MTDPin) \cong \Omega_k^{\text{DPin}}$, then

$$MTDPin \simeq MTSpin \wedge (B\mathbb{Z}/2 \times B\mathbb{Z}/2)^{L_1 L_2 + 3L_2 - 4}. \tag{F.3}$$

The second summand, $(B\mathbb{Z}/2 \times B\mathbb{Z}/2)^{L_1 L_2 + 3L_2 - 4}$, which we denote $X$ to tame the notation, is the Thom spectrum of the virtual vector bundle

$$V := (L_1 \otimes L_2) \oplus (L_2)^{\oplus 3} - \mathbb{R}^4 \longrightarrow B\mathbb{Z}/2 \times B\mathbb{Z}/2. \tag{F.4}$$

By (F.3), $\Omega_k^{\text{DPin}} \cong \widetilde{\Omega}_k^{\text{Spin}}(X)$.

We will compute $\widetilde{\Omega}_k^{\text{Spin}}(X)$ for $0 \le k \le 6$ for our $X$ using the Adams spectral sequence, employing a standard trick to work over $\mathcal{A}(1) := \langle \text{Sq}^1, \text{Sq}^2 \rangle$ rather than the entire Steenrod algebra. For details on how this works and many worked examples, see Beaudry-Campbell [97], who carefully explain and summarize how to use the Adams spectral sequence for these kinds of computations. The idea is that we must determine $\widetilde{H}^*(X; \mathbb{F}_2)$ as an $\mathcal{A}(1)$-module. Then, the $E_2$-page of this Adams spectral sequence is

$$E_2^{s,t} = \text{Ext}_{\mathcal{A}(1)}^{s,t}(\widetilde{H}^*(X; \mathbb{F}_2), \mathbb{F}_2). \tag{F.5}$$

(Definitions and notation are as in [97].) The spectral sequence converges to $\widetilde{ko}_{t-s}(X) \otimes \hat{\mathbb{Z}}_2$, where $ko$ denotes connective real $K$-theory and $\hat{\mathbb{Z}}_2$ denotes the 2-adic integers. Furthermore, when $t - s \le 7$, $\widetilde{ko}_{t-s}(X)$ is isomorphic to $\widetilde{\Omega}_{t-s}^{\text{Spin}}(X)$ [96]. We will show, for our particular

---

[17]This appendix was contributed by Arun Debray.

choice of $X$, $\widetilde{\Omega}_*^{\text{Spin}}(X)$ lacks torsion for odd primes. Therefore tensoring it with $\hat{\mathbb{Z}}_2$ does not lose any information. (In general, information can be lost when tensoring with $\hat{\mathbb{Z}}_2$, but that information can be computed by other means.) This allows us to use the spectral sequence above to compute $\widetilde{\Omega}_{t-s}^{\text{Spin}}(X)$ in the degrees of our interest.

*Proof of Theorem F.1.* First we argue $\widetilde{\Omega}_*^{\text{Spin}}(X)$ has no $p$-torsion for odd primes $p$. In fact, we will show that if $p$ is an odd prime, $\widetilde{\Omega}_*^{\text{Spin}}(X) \otimes \mathbb{F}_p = 0$. For any finitely generated abelian group $A$, the $p$-torsion subgroup of $A$ includes into the $p$-torsion subgroup of $A \otimes \mathbb{F}_p$, so this suffices.

By definition, $\widetilde{\Omega}_k^{\text{Spin}}(X) \cong \widetilde{H}_k(MT\text{Spin} \wedge X)$. Tensoring with $\mathbb{F}_p$, the map

$$\widetilde{H}_k(MT\text{Spin} \wedge X) \otimes \mathbb{F}_p \longrightarrow \widetilde{H}_k(MT\text{Spin} \wedge X; \mathbb{F}_p) \tag{F.6}$$

is injective, by the universal coefficient theorem. The Künneth theorem computes $\widetilde{H}_*(MT\text{Spin} \wedge X; \mathbb{F}_p)$ as a sum of tensor products of the form $\widetilde{H}_i(MT\text{Spin}; \mathbb{F}_p) \otimes \widetilde{H}_j(X; \mathbb{F}_p)$, so it suffices to show $\widetilde{H}_j(X; \mathbb{F}_p)$ vanishes for all $j$. The twisted-coefficients Thom isomorphism tells us there is a (in this case nontrivial) $\mathbb{Z}[\mathbb{Z}/2 \times \mathbb{Z}/2]$-module structure $\widetilde{\mathbb{F}}_p$ on $\mathbb{F}_p$ such that

$$\widetilde{H}_j(X; \mathbb{F}_p) \cong H_j(\mathbb{Z}/2 \times \mathbb{Z}/2; \widetilde{\mathbb{F}}_p). \tag{F.7}$$

Maschke's theorem implies that since $\#(\mathbb{Z}/2 \times \mathbb{Z}/2)$ and $p$ are coprime, and since $\widetilde{\mathbb{F}}_p$ is $p$-torsion, $H_j(\mathbb{Z}/2 \times \mathbb{Z}/2; \widetilde{\mathbb{F}}_p)$ vanishes in degrees $j > 0$. Using that $0^{\text{th}}$ group homology is the abelian group of coinvariants, one can check directly that $H_0(\mathbb{Z}/2 \times \mathbb{Z}/2; \widetilde{\mathbb{F}}_p) = 0$ as well. Thus $\widetilde{\Omega}_*^{\text{Spin}}(X)$ has no $p$-torsion.

On to the Adams spectral sequence. First we determine $\widetilde{H}^*(X; \mathbb{F}_2)$. As a graded abelian group, this is characterized by the Thom isomorphism: if $U \in \widetilde{H}^0(X; \mathbb{F}_2)$ denotes the Thom class, cup product with $U$ is an isomorphism

$$(U \cdot): H^k(B\mathbb{Z}/2 \times B\mathbb{Z}/2; \mathbb{F}_2) \xrightarrow{\cong} \widetilde{H}^k(X; \mathbb{F}_2). \tag{F.8}$$

There is no degree shift because the virtual vector bundle $V \to B\mathbb{Z}/2 \times B\mathbb{Z}/2$ (from (F.4)) has rank zero. Let $w := w_1(L_1)$ and $a := w_1(L_2)$ in $H^1(B\mathbb{Z}/2 \times B\mathbb{Z}/2; \mathbb{F}_2)$; then

$$H^*(B\mathbb{Z}/2 \times B\mathbb{Z}/2; \mathbb{F}_2) \cong \mathbb{F}_2[w, a]. \tag{F.9}$$

The $\mathcal{A}(1)$-module structure on $\widetilde{H}^*(X; \mathbb{F}_2)$ is determined by the following rules.

1. $\text{Sq}^i(U) = U w_i(V)$, where $w_i$ denotes the $i^{\text{th}}$ Stiefel-Whitney class. In this case, $w_1(V) = w$ and $w_2(V) = wa$.

2. The Cartan formula determines the Steenrod squares of a product. We only need $\text{Sq}^1$ and $\text{Sq}^2$, for which the Cartan formula specializes to

$$\text{Sq}^1(xy) = \text{Sq}^1(x)y + x\,\text{Sq}^1(y) \tag{F.10a}$$
$$\text{Sq}^2(xy) = \text{Sq}^2(x)y + \text{Sq}^1(x)\,\text{Sq}^1(y) + x\,\text{Sq}^2(y). \tag{F.10b}$$

3. From the axiomatic properties of Steenrod squares, $\text{Sq}^1(w) = w^2$, $\text{Sq}^1(a) = a^2$, and $\text{Sq}^2(w) = \text{Sq}^2(a) = 0$.

Using these three rules one can determine the action of $\text{Sq}^1$ and $\text{Sq}^2$ on any cohomology class of $X$, as it is a sum of products of $U$, $w$, and $a$. This is routine, and indeed we used a computer program to make these calculations. The answer is displayed in Figure 6.

From this figure, we see that, as an $\mathcal{A}(1)$-module, $\widetilde{H}^*(X; \mathbb{F}_2)$ splits into several summands. All summands pictured except the orange summand are isomorphic to shifts of $\mathcal{A}(1)$. The

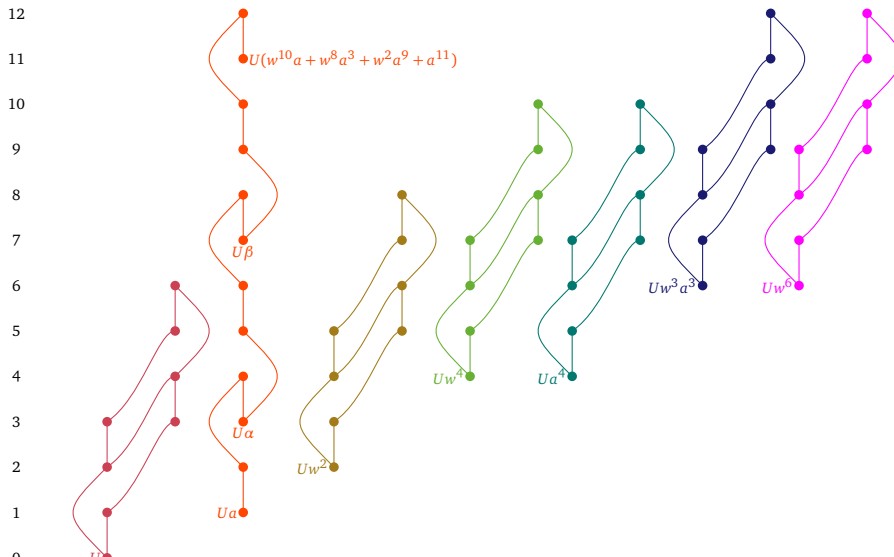

Figure 6: This $\mathcal{A}(1)$-submodule of $\widetilde{H}^*(X;\mathbb{F}_2)$ contains all elements of degree at most 7. Each dot represents an $\mathbb{F}_2$ summand, with its cohomological degree given by its height. The connecting lines, resp. curves, indicate an action by $\mathrm{Sq}^1$, resp. $\mathrm{Sq}^2$, carrying the lower dot to the upper dot. This $\mathcal{A}(1)$-module factors as several different summands; we give each summand a different color. In the generators of the orange summand, $\alpha := w^2 a + a^3$ and $\beta := w^6 a + w^4 a^3 + w^2 a^5 + a^7$.

orange summand, i.e. the one that contains $Ua$, continues above what we draw in Figure 6 and is isomorphic to the mod 2 cohomology of the spectrum $MO(1)$, the Thom spectrum of the tautological line bundle $\sigma \to BO(1)$ (see [97, Figure 4]); therefore we denote that summand by $\widetilde{H}^*(MO(1))$.[18] Specifically,

$$\widetilde{H}^*(X;\mathbb{F}_2) \cong \mathcal{A}(1) \oplus \widetilde{H}^*(MO(1)) \oplus \Sigma^2 \mathcal{A}(1) \oplus \Sigma^4 \mathcal{A}(1) \oplus \Sigma^4 \mathcal{A}(1) \oplus \Sigma^6 \mathcal{A}(1) \oplus \Sigma^6 \mathcal{A}(1) \oplus P, \quad \text{(F.11)}$$

where $P$ has no elements of degree less than 8. Hence, below degree $t - s = 8$, the $E_2$-page of the Adams spectral sequence (F.5) is the direct sum of the $E_2$-pages of the summands other than $P$, and these have all been calculated. For $\Sigma^k \mathcal{A}(1)$, there is a single $\mathbb{F}_2$ summand in bidegree $s = 0$, $t = k$; for $\widetilde{H}^*(MO(1))$, see [98, Example 6.3]. Putting these together, the $E_2$-page for this spectral sequence is

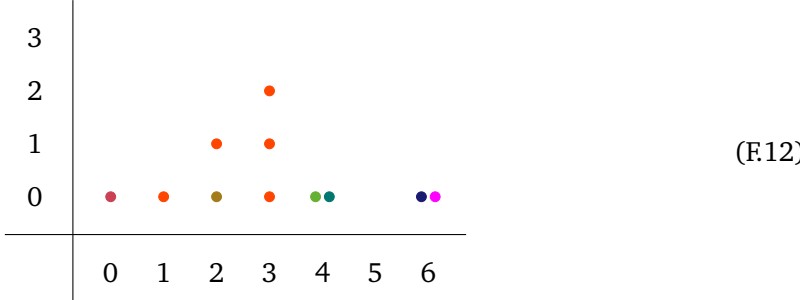

(F.12)

In this diagram, the $x$-axis is $t - s$ and the $y$-axis is $s$. Therefore a differential $d_r$ moves one degree to the left and $r$ degrees upwards. Each dot represents an $\mathbb{F}_2$ summand of the $E_2$-page; the different colors indicate which summands of $\widetilde{H}^*(X;\mathbb{F}_2)$ are responsible for which

---

[18]Strictly speaking, we have only calculated this summand up to degree 12, and it could differ from $\widetilde{H}^*(MO(1))$ in larger degrees. This would only affect the $E_2$-page in degrees larger than we use and display in (F.12), so the calculation is the same in either case.

data on the $E_2$-page. The $E_2$-page carries an action by $\mathrm{Ext}^{*,*}_{\mathcal{A}(1)}(\mathbb{F}_2, \mathbb{F}_2)$. The vertical lines indicate action by an element $h_0 \in \mathrm{Ext}^{1,1}_{\mathcal{A}(1)}(\mathbb{F}_2, \mathbb{F}_2)$, and the diagonal lines indicate action by $h_1 \in \mathrm{Ext}^{2,1}_{\mathcal{A}(1)}(\mathbb{F}_2, \mathbb{F}_2)$; see [97, Example 4.1.1] for more on $h_0$ and $h_1$. All differentials are $h_0$- and $h_1$-linear, i.e. $d_r(h_i x) = h_i d_r(x)$ ($i = 0, 1$). In this example, the only differential within the range displayed in (F.12) that could be nonzero is the $d_2$ from bidegree $(4, 0)$ to bidegree $(3, 2)$. Often, $h_0$- and $h_1$-linearity allow one to deduce that differentials vanish, but this does not provide any information about this $d_2$, so we have to do something different.

There will also be a question of extension problems: the line $t - s = k$ is the associated graded of a filtration, possibly nontrivial, on $\widetilde{\Omega}^{\mathrm{Spin}}_k(X)$. This in particular introduces an ambiguity in $\widetilde{\Omega}^{\mathrm{Spin}}_2(X)$: it could either be $\mathbb{Z}/2 \oplus \mathbb{Z}/2$ or $\mathbb{Z}/4$. Fortunately, when we show this $d_2$ vanishes in Corollary F.16, we will also be able to resolve this ambiguity.

Recall that a dpin structure on $M$ is data of two line bundles $L_1, L_2 \to M$ and a spin structure on $TM \oplus (L_1 \otimes L_2) \oplus (L_2)^{\oplus 3}$. Computing $w_1$ and $w_2$ of this bundle with the Whitney sum formula shows that if $(M, L_1, L_2)$ has a dpin structure, $w_1(M) = w_1(L_1)$ and $w_2(M) = w_1(M)(w_1(M) + w_1(L_2))$.

**Lemma F.13.** *The assignment from $(M, L_1, L_2)$ to a smooth representative of the Poincaré dual of $w_1(L_2) \in H^1(M; \mathbb{Z}/2)$ induces a map $D_{L_2} : \widetilde{\Omega}^{\mathrm{Spin}}_d(X) \to \Omega^{\mathrm{Pin}^-}_{d-1}$.*

*Proof.* These kinds of arguments are standard in bordism theory (e.g. [24, 25, 27]), so we will be succinct. Let $i : N \hookrightarrow M$ be a smooth representative for the Poincaré dual of $w_1(L_2)$ and $\nu \to N$ be the normal bundle; then $w(\nu) = 1 + w_1(L_2)$. Using the short exact sequence $0 \to TN \to \nu \to TM|_N \to 0$ and the Whitney sum formula, we get

$$w_1(N) = i^*(w_1(M) + w_1(L_2)) \tag{F.14a}$$

$$w_2(N) = i^*(w_2(M)) + w_1(N)w_1(\nu) \tag{F.14b}$$

$$= i^*(w_1(M)^2 + w_1(M)w_1(L_2) + (w_1(M) + w_1(L_2))w_1(L_2)) \tag{F.14c}$$

$$= i^*(w_1(M)^2 + w_1(L_2)^2) \tag{F.14d}$$

$$= w_1(N)^2, \tag{F.14e}$$

so $N$ admits a pin$^-$ structure; a choice of pin$^-$ structure amounts to the additional data of a nullhomotopy of the map $w_2 + w_1^2 : N \to K(\mathbb{Z}/2, 2)$. A choice of dpin structure on $(M, L_1, L_2)$ includes (up to a contractible choice) data of nullhomotopies of the maps $w_1(M) + w_1(L_1) : M \to K(\mathbb{Z}/2, 1)$ and $w_2(M) + w_1(M)(w_1(M) + w_1(L_2)) : M \to K(\mathbb{Z}/2, 2)$; via (F.14a), this induces a nullhomotopy of the map $w_1(N) + i^*(w_1(M) + w_1(L_2)) : N \to K(\mathbb{Z}/2, 1)$, which then induces a nullhomotopy of $w_2 + w_1^2 : N \to K(\mathbb{Z}/2, 2)$ via the rest of (F.14). The proof of bordism invariance of this construction is as usual. $\qquad\square$

$D_{L_2}$ is an example of a *Smith homomorphism*. For a general discussion of Smith homomorphisms, see e.g. [25, §4].

**Lemma F.15.** *The image of $D_{L_2} : \widetilde{\Omega}^{\mathrm{Spin}}_3(X) \to \Omega^{\mathrm{Pin}^-}_2 \cong \mathbb{Z}/8$ contains a generator of $\Omega^{\mathrm{Pin}^-}_2$.*

*Proof.* Let $a \in H^1(\mathbb{RP}^3; \mathbb{F}_2)$ be the generator. Let $L_1 \to \mathbb{RP}^3$ be trivial and $L_2 \to \mathbb{RP}^3$ be the tautological bundle, so $w_1(L_1) = 0$ and $w_1(L_2) = a$. Since $w(\mathbb{RP}^3) = (1 + a)^4 = 1 + a^4 = 0$, $w_1(\mathbb{RP}^3) = 0 = w_1(L_1)$ and $w_2(\mathbb{RP}^3) = 0 = w_1(\mathbb{RP}^3)(w_1(\mathbb{RP}^3)w_1(L_2))$. Hence $(\mathbb{RP}^3, L_1, L_2)$ admits a dpin structure; choose one.

The standard embedding $\mathbb{RP}^2 \hookrightarrow \mathbb{RP}^3$ represents the homology class Poincaré dual to $a$, so $D_{L_2}(\mathbb{RP}^3, L_1, L_2)$ is the pin$^-$ bordism class of $\mathbb{RP}^2$ with one of its two pin$^-$ structures. Kirby-Taylor [27, §3] describe how to show that $\mathbb{RP}^2$ with either choice of pin$^-$ structure generates $\Omega^{\mathrm{Pin}^-}_2$. $\qquad\square$

**Corollary F.16.**

1. $\widetilde{\Omega}_3^{\mathrm{Spin}}(X) \cong \mathbb{Z}/8$, so the $d_2$ noted above vanishes.

2. The extension
$$0 \longrightarrow \mathbb{Z}/2 \longrightarrow \widetilde{\Omega}_2^{\mathrm{Spin}}(X) \longrightarrow \mathbb{Z}/2 \longrightarrow 0, \tag{F.17}$$
which comes from the Adams filtration on $\widetilde{\Omega}_2^{\mathrm{Spin}}(X)$, splits.

*Proof.* For (1), let $\overline{x}, \overline{y}$ be elements on the $E_\infty$-page, i.e. elements of the associated graded of the Adams filtration. It is a general fact about the Adams spectral sequence that if $h_0\overline{x} = \overline{y}$, then there are preimages $x, y \in \widetilde{\Omega}_*^{\mathrm{Spin}}(X)$ of $\overline{x}$, resp. $\overline{y}$, such that $2x = y$. For example, supposing the $d_2$ of interest were nonzero, the line $t-s = 3$ on the $E_\infty$-page (i.e. the associated graded of $\widetilde{\Omega}_2^{\mathrm{Spin}}(X)$) would contain exactly two $\mathbb{Z}/2$ summands linked by an $h_0$; hence there would be nonzero $x_1, x_2 \in \widetilde{\Omega}_3^{\mathrm{Spin}}(X)$ with $x_1 = 2x_2$, so $\widetilde{\Omega}_3^{\mathrm{Spin}}(X) \cong \mathbb{Z}/4$. On the other hand, if $d_2 = 0$, there would be three $\mathbb{Z}/2$ summands linked by $h_0$s, so there would be nonzero $x_1, x_2 \in \widetilde{\Omega}_3^{\mathrm{Spin}}(X)$ with $x_1 = 4x_2$, and hence $\widetilde{\Omega}_3^{\mathrm{Spin}}(X)$ would be $\mathbb{Z}/8$. That is, $\widetilde{\Omega}_3^{\mathrm{Spin}}(X)$ is isomorphic to either $\mathbb{Z}/8$, if the $d_2$ in question vanishes, or $\mathbb{Z}/4$, if that $d_2$ does not vanish. Lemma F.15 says $\widetilde{\Omega}_3^{\mathrm{Spin}}(X)$ admits a surjective map to $\mathbb{Z}/8$, so $\mathbb{Z}/4$ does not work.

On to (2). Like in the above case with $h_0$, it is a general fact about the Adams spectral sequence that if $h_1\overline{x} = \overline{y}$, then one can choose preimages $x$ and $y$ in $\widetilde{\Omega}_*^{\mathrm{Spin}}(X)$ such that $\eta \cdot x = y$, where $\eta$ is the generator of $\pi_1\mathbb{S} \cong \mathbb{Z}/2$. (Concretely, if $x$ is the dpin bordism class of some manifold $M$, then $\eta \cdot x$ is the bordism class of $S^1 \times M$, where $S^1$ has the dpin structure induced from the nonbounding framing.)

If the extension in (F.17) did not split, then $\widetilde{\Omega}_2^{\mathrm{Spin}}(X)$ would be $\mathbb{Z}/4$ rather than $\mathbb{Z}/2 \oplus \mathbb{Z}/2$. However, we can rule this out: suppose it were $\mathbb{Z}/4$, and let $x$ be a generator. Then the image of $x$ in the associated graded of $\widetilde{\Omega}_2^{\mathrm{Spin}}(X)$ (i.e. the $t-s = 2$ line of the Adams $E_\infty$-page) is the nontrivial element of the yellow $\mathbb{Z}/2$ summand in bidegree $(2,0)$, and the image of $2x$ is the nonzero element of the orange $\mathbb{Z}/2$ summand in bidegree $(2,1)$. The $h_1$-action carries this to the nonzero element of the orange $\mathbb{Z}/2$ summand in bidegree $(3,2)$, so $\eta \cdot 2x \neq 0$. Since $2\eta = 0$, however, this is a contradiction, forcing $\widetilde{\Omega}_2^{\mathrm{Spin}}(X) \cong \mathbb{Z}/2 \oplus \mathbb{Z}/2$. $\square$

There can be no more nontrivial differentials or hidden extensions in the range shown in (F.12), so we are done. $\square$

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
