# Peer review of "Topological superconductors on superstring worldsheets"

_SciPost Physics, doi:SciPost Phys. 9, 010 (2020)_

## Round 1 · Referee Report · Anonymous (Referee 1) · 2020-4-24

Report

In this paper the authors provide a systematic understanding of GSO projection in
superstring theory by relating it to symmetry protected topological phases. Besides
providing us with a systematic classification of known string theories, this procedure
also gives new type 0 string theories. The authors' analysis also gives a systematic
way to classify the D-branes in these theories and derive their various
properties. They also give a natural description of the boundary degrees of freedom
of open strings on non-BPS branes, whose origin was somewhat obscure in the earlier
description of such D-branes. They also confirm their results using boundary state
construction of D-branes. This paper should definitely be published in SciPost.

I have one minor comment that the authors may want to take into account. For type I
string theory, the classification of stable D-branes using K-theory seems somewhat formal,
since it does not take into account possible tachyonic modes of the
open strings stretched between the D-brane under study and the
space-filling D9-branes that
must be present
for tadpole cancellation. For example type I D0-brane is stable but type I D8-brane
is expected to be unstable since the open strings connecting the D8-brane to the
D9-brane has $<4$ world-sheet fields satisfying ND boundary condition and we expect
their spectrum to contain a tachyon.
This issue will arise also in many of the unoriented type 0 theories, for
which the authors' analysis in appendix C shows that we need to add D9-branes to
cancel tadpoles.
This of course does not contradict any mathematical result since K-theory
gives classification
of stable D-branes in the presence of the orientifold plane but not in the presence of
D9-branes. As far as I can see, in the authors' analysis also the classification of stable
branes is done in absence of D9-branes. For example table 5 does not have a list of tachyons from the Dp-D9 strings. It will be better to state this explicitly to avoid confusion.

I also found one minor typo: the last term in the numerator of eq.(2.4) should be $-iE+m$.

---

## Round 1 · Referee Report · Anonymous (Referee 2) · 2020-6-7

Report

This paper has been a joy to read.
It offers a new approach to the systematic classification of closed string theories with N=(1,1) worldsheet supersymmetry, by relating them to topological fermion phases in two dimensions.
I highly recommend publication.

---

## Round 6 · Author Response

We thank the referees for their comments, which have been included in the revised version of the manuscript.

---

## Round 6 · List of Changes

- We have added additional paragraphs at the end of Section 3 and Appendix C, with comments about the possibility of branes which are stable according to K-theory becoming unstable in the presence of D9-branes required by tadpole cancellation.

- We have fixed the typo in Eq. (2.4).

- We have added a new Appendix F with a computation of the relevant bordism groups using the Adams spectral sequence, where agreement is found with the results of the AHSS. This appendix is by Arun Debray, who has been added to the list of authors.

You are currently on this page

Resubmission scipost_202002_00003v6 on 24 June 2020

---

## Editorial Decision

published